# Overlapping nuclear import and export paths unveiled by two-colour MINFLUX

Abhishek Sau[1], Sebastian Schnorrenberg[2], Ziqiang Huang[2], Debolina Bandyopadhyay[1], Ankith Sharma[1], Clara-Marie Gürth[3], Sandeep Dave[1] & Siegfried M. Musser[1✉]

The nuclear pore complex (NPC) mediates nucleocytoplasmic exchange, catalysing a massive flux of protein and nucleic acid material in both directions[1]. Distinct trafficking pathways for import and export would be an elegant solution to avoid unproductive collisions and opposing movements. However, the three-dimensional (3D) nanoscale spatiotemporal dynamics of macromolecules traversing the NPC remains challenging to visualize on the timescale of millisecond-scale transport events. Here we used 3D MINFLUX[2] to identify the nuclear pore scaffold and then to simultaneously monitor both nuclear import and nuclear export, thereby establishing that both transport processes occur in overlapping regions of the central pore. Whereas translocation-arrested import complexes bound at the pore periphery, tracks of translocating complexes within the central pore region revealed a preference for an approximately 40- to 50-nm diameter annulus with minimal circumferential movement, indicating activity-dependent confinement within the permeability barrier. Movement within the pore was approximately 1,000-fold slower than in solution and was interspersed with pauses, indicating a highly restricted environment with structural constraints and/or transient binding events during transport. These results demonstrate that high spatiotemporal precision with reduced photobleaching is a major advantage of MINFLUX tracking, and that the NPC permeability barrier is divided into annular rings with distinct functional properties.

The NPC embedded within the nuclear envelope of eukaryotic cells mediates the bidirectional transport of both small and large protein and nucleic acid cargos[1,3]. An intrinsically disordered polypeptide network generates the permeability barrier and provides binding sites for the cargo-carrying nuclear transport receptors (NTRs)[4,5]. This complex milieu occupies the approximately 50- to 70-nm diameter pore[6] where the transiting cargos are confined to specific regions during their millisecond-scale migration through the barrier[7–10]. Physically separate trafficking pathways for import and export would provide an elegant solution to alleviate congestion and prevent unproductive collisions between complexes moving in opposite directions[8–12]. Simultaneously monitoring both import and export with nanometre-scale precision would provide a rigorous test of this model. Although directly monitoring actively diffusing molecules with high spatiotemporal precision in cellular contexts is feasible with single-molecule fluorescence and super-resolution methods, the illumination intensities typically required result in rapid photobleaching, thus reducing sample observation time[13,14]. Consequently, although real-time transport trajectories obtained thus far have directly visualized transport through the NPC, the number of localizations within the pore itself have been low (typically less than three)[7,9,15], making experimental characterization of the permeability barrier properties largely inaccessible. The substantially higher illumination intensities needed by diffraction-limited approaches to increase time resolution[13] raise serious concerns about sample integrity and have not yielded substantially longer trajectories[9].

MINFLUX (minimal emission flux) is a powerful recently developed strategy to achieve exceptional spatiotemporal resolution on single molecules. It thus has the potential for delivering incredible new breakthroughs in biological imaging and cellular dynamics; nonetheless, its substantial promise for high-precision tracking of moving molecules remains at an early stage. Single-particle localization with MINFLUX is extremely photon efficient compared with diffraction-limited approaches, thus requiring substantially lower illumination intensities to achieve a similar precision[2,16]. MINFLUX localizes fluorophores with high precision and a low photon count by scanning an excitation donut in a pattern around a fluorophore and calculating the position of the molecule on the basis of the number of photons collected at the various positions[17]. For nanometre-scale tracking, the assumption is that diffusional steps are approximately less than 100–200 nm (depending on acquisition parameters), and therefore, that the scan pattern for each subsequent localization can be initiated where the molecule was last found[16]. For free diffusion in low-viscosity solutions, this assumption cannot currently be met due to diffusional steps in excess of multiple hundreds of nanometres per millisecond-scale timestep. However, in a diffusionally restricted environment, such as the NPC permeability barrier, MINFLUX tracking is feasible. We now demonstrate high-precision tracking of cargos moving in both directions

[1]Department of Cell Biology and Genetics, Texas A&M University, College Station, TX, USA. [2]EMBL Imaging Centre, European Molecular Biology Laboratory, Heidelberg, Germany. [3]Abberior Instruments GMBH, Göttingen, Germany. ✉e-mail: smusser@tamu.edu

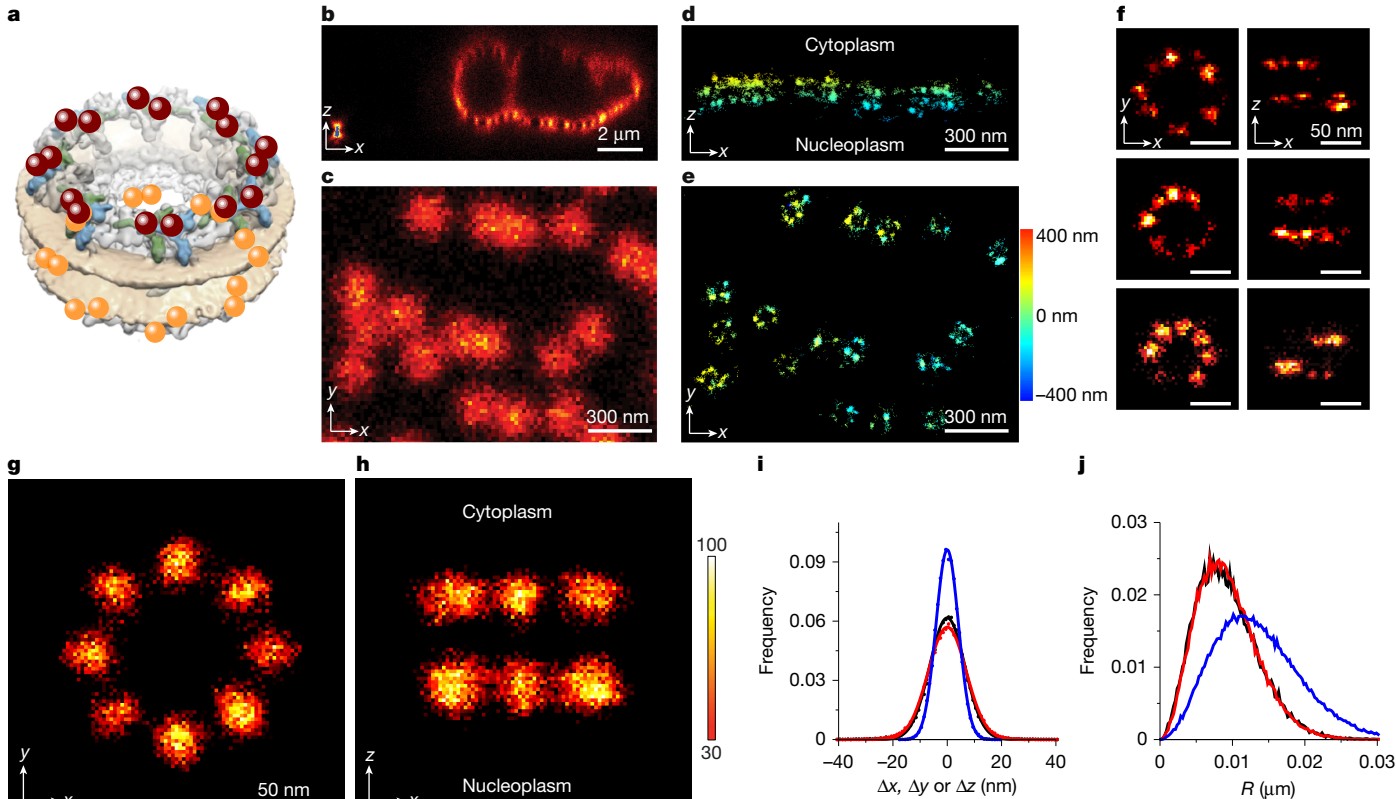

**Fig. 1 | MINFLUX imaging of transport-active nuclear pores. a**, Equal distribution of 32 NUP96 molecules between the cytoplasmic (maroon) and nucleoplasmic (orange) rings of human NPCs. Adapted from the electron microscopy density map EMD-2444 (refs. 28,40, Springer Nature, and ref. 41, Cell Press). **b**–**e**, MINFLUX imaging of NPCs in permeabilized U2OS cells containing NUP96–mEGFP. The confocal image of eGFP fluorescence identifies the outline of a cell nucleus and a gold bead (100 nm) used for image stabilization (lower left corner, **b**). A section of the bottom of the nucleus in **b** (**c**), and 3D MINFLUX imaging of NPCs (**d**,**e**) are also shown. Anti-GFP nanobodies (Nb[GFP])[42] modified with the HMSiR blinking dye[43] were used to visualize the NPCs within the region shown in **c** via the stochastic blinking of the dye. In **d**, the curvature of the nuclear envelope is apparent from the layers defined by the cytoplasmic and nucleoplasmic rings of the NPCs (see **a**). In **e**, all NPCs identified by confocal imaging in **c** were detected. Coloration shows the *z* scale. During data collection, the cytoplasm was on the bottom, but images throughout this article were flipped to place the cytoplasm on top for consistency with convention. **f**, MINFLUX images of single NPCs (more examples in Extended Data Fig. 3a). **g**,**h**, Composite 2D histogram images of an averaged NPC obtained by aligning individual pores on the basis of their centroids and rotated on the basis of their expected eightfold rotational symmetry (see Methods and Extended Data Fig. 3; 37 cells, 541 NPCs, *n* = 82,331 localizations). The scale is percent of maximum. **i**, Localization precision determined from centroid deviations within HMSiR 'trajectories' (20 points or more per trajectory, 37 cells, 269 clusters; *n* = 32,184 localizations; $\sigma_x = 6.5 \pm 0.1$ nm (black), $\sigma_y = 7.0 \pm 0.1$ nm (red) and $\sigma_z = 4.2 \pm 0.1$ nm (blue)). The values $\sigma_x/\sigma_y = 0.93$ and $\sigma_x/\sigma_z = 1.55$ were assumed throughout this article. **j**, Jump step histogram analysis of localization precision. The predicted distribution assuming the localization precision values determined in **i** (blue curve) fits the experimental data (black) poorly, thus indicating that the method in **i** overestimates the localization precision. A simulation model assuming diffusional drift (red; *n* = 96,000 jump steps, 25 localizations per trajectory; $\sigma_x = 4.1$ nm $= 0.93\sigma_y = 1.55\sigma_z$; $D_x, D_y$ and $D_z = 0.00072, 0.00083$ and $0.0003$ $\mu m^2 s^{-1}$, respectively) agrees with the data and yields the same centroid deviations as determined in **i** (see Extended Data Fig. 5i). See Methods and Extended Data Fig. 5 for a description of the analytical approach and a fit with no diffusional drift.

through NPCs, whose position and orientation were directly imaged within the same experiment.

## Imaging transport-active NPCs

NPCs have been a key biological structure for developing and demonstrating the resolving power of super-resolution approaches as they are structurally well characterized and have suitable dimensions, symmetry and physiological numbers for illustrating advanced imaging capabilities[2,18–20]. Although NPCs are well resolved by 3D MINFLUX, most studies have utilized fixed cells and GLOX buffer (deoxygenation + thiol) to induce blinking for dSTORM-type imaging[2,17,18], conditions that preclude real-time nucleocytoplasmic transport measurements. Although fixation ensures structural stability, we recently demonstrated high spatial stability (less than 10-nm fluctuations) of NPCs in an unfixed permeabilized cell system[7]. In this previous work, NPCs were localized using 3D astigmatism imaging with the spontaneous blinking dye HMSiR

affixed via an anti-GFP nanobody to NUP96–monomeric enhanced GFP (mEGFP), which yielded individual *xyz* localization precisions of 7–12 nm. The identical conditions were used here to image NPCs on the bottom of U2OS cell nuclei using 3D MINFLUX with up to 3.3-ms time resolution per HMSiR dye localization (Extended Data Fig. 1a,b and Supplementary Table 1). Within 15 min, more than 100 localizations per NPC were typically obtained, thereby identifying the locations of 30–50 NPCs per field. Gold beads near the cell surface were used to correct any translational fluctuations of the sample with less than 2-nm precision in *xyz* (Fig. 1a–e). Confocal imaging of eGFP fluorescence at the beginning and end of the MINFLUX acquisition revealed that the NPCs at the centre of the semi-flat nuclear envelope were the most positionally stable, which was quantitatively confirmed by splitting the MINFLUX dataset to generate early and late images (Extended Data Fig. 2). Recognizable single-pore images were obtained from HMSiR localizations (Fig. 1f and Extended Data Fig. 3a). A composite NPC image exhibited the expected double-ring structure and eightfold rotational

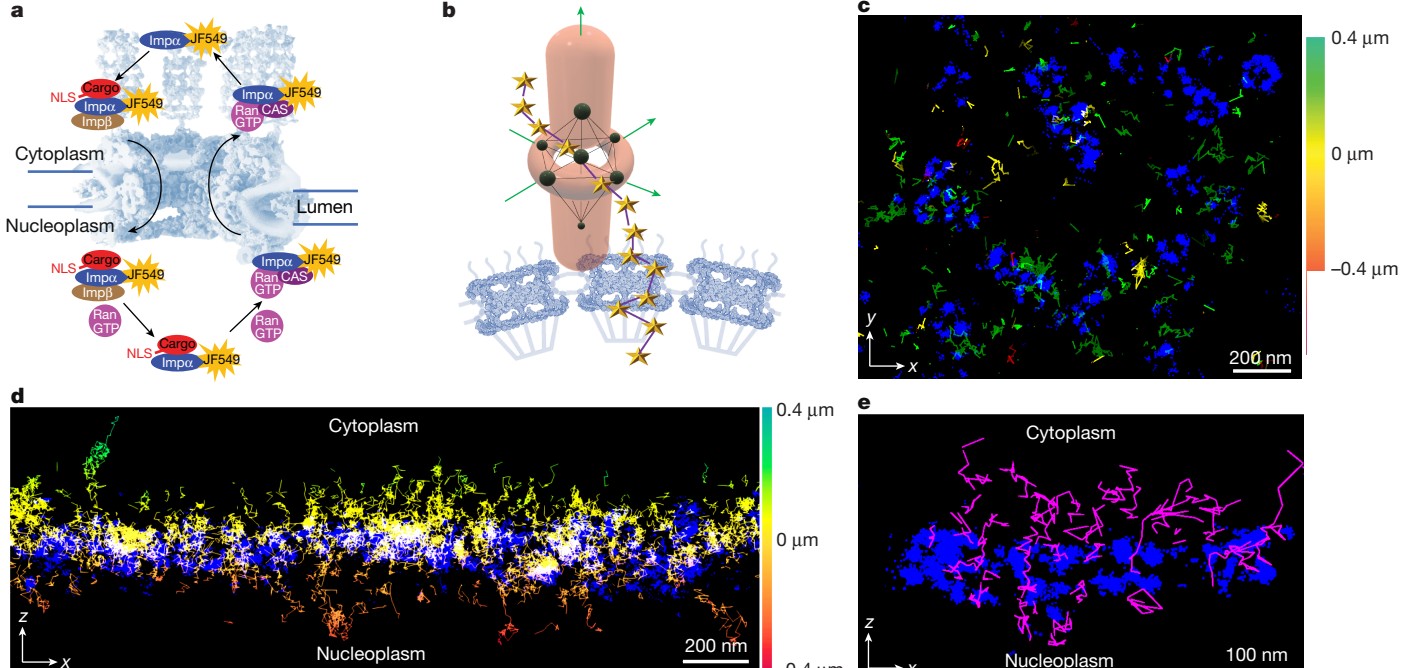

**Fig. 2 | MINFLUX tracking of the bidirectional transport of Imp α.**
**a**, Schematic of the concurrent import and export of Imp α labelled with JF549.
The NPC structure was adapted from refs. 44,45, AAAS. **b**, Target coordinate
pattern. The MINFLUX 3D donut was scanned in a seven-point octahedral
pattern (black dots) for the Imp α–JF549 tracking algorithm, yielding
successive localizations (gold stars). The NPC structure was adapted from
ref. 28, Springer Nature, ref. 46, AAAS and the RCSB Protein Data Bank[47].

**c**,**d**, Unfiltered two-colour MINFLUX localization data obtained in the presence
of transport mix. NPC localizations (blue; Nb[GFP]–HMSiR; see Fig. 1) were collected
for 20 min, and these were followed by tracking localizations (coloured z scale,
Imp α–JF549) collected for 20 min. Views from the cytoplasm (**c**; xy) and the
side (**d**; xz) for two different cells are shown. **e**, Tracks (magenta) that satisfied
MINFLUX filtering criteria (see Methods and Extended Data Fig. 6) overlaid
onto four NPC scaffolds (blue).

symmetry (Fig. 1f,g and Extended Data Fig. 3). Measurement of the
spacing between the cytoplasmic and nucleoplasmic rings by astig-
matism imaging was used to calibrate the z scale in MINFLUX images
(see Methods; Extended Data Fig. 4 and Supplementary Table 2). These
data confirmed the positional stability of the NPCs in unfixed permea-
bilized cells under MINFLUX imaging conditions.

As the HMSiR fluorophores were bound to an NPC scaffold during
imaging, successive localizations could therefore be used to estimate
localization precision. On the basis of centroid deviations, localization
precisions of $\sigma_x = 6.5$ nm, $\sigma_y = 7.0$ nm and $\sigma_z = 4.2$ nm were obtained,
yielding $\sigma_x/\sigma_y = 0.93$ and $\sigma_x/\sigma_z = 1.55$ (Fig. 1i). Jump step analysis revealed,
however, that these precision values were too large (Fig. 1j, blue curve).
The explanation is that the movement of particle centroids during the
measurements broadened the observed distributions of repeated
localizations. Whereas sample drift or multiple semi-stable dye posi-
tions around the attachment point can produce significant devia-
tions of the average centroid from the instantaneous centroid during
acquisition of a localization, jump step analysis only includes distances
between successive pairs of points and thus is much less sensitive to
centroid error. The jump step data were well fit by a simulation model
that included diffusional drift and 37% lower localization precisions
(Fig. 1j, red curve); these simulated data yielded centroid deviations
that matched the experimentally determined values (Extended Data
Fig. 5i). The 'diffusional drift' required to fit the data is not expected to
be sample drift that would generate large displacements over the total
imaging time, as this would make it impossible to assemble the NPC
scaffolds from localizations collected over a 20-min period. Rather, the
diffusional drift models the confined movement of the dye centroid
on the NPC scaffold during the measurement period, for example, to
favourable positions enabled by linkage error, or conformational shifts
or jiggles of the NPC scaffold. Considering that the nanobody-bound
HMSiR fluorophore is potentially up to 6 nm away from the NUP96

attachment point of the eGFP tag[7], a low nanometre-scale zone in which
the dye maintains distinct preferential positions is reasonable.

## Bidirectional transport of importin α

We next sought to track import and export complexes migrating
through the super-resolved NPCs (Fig. 2a,b). This required that the
MINFLUX excitation donut was at the right place at the right time for
the expected approximately 10-ms duration translocation events[21]. As
is typical for MINFLUX, we scanned a region of interest (for example,
Fig. 1c), and then, once a fluorescent molecule was found, a series of
reduced-size scan patterns were implemented for increased locali-
zation precision (see Methods and Supplementary Table 3). Thus, a
fluorophore had to be first detected through the scanning approach,
and then continuously tracked with the anticipation that the molecule
might go through an NPC (Fig. 2b), which turned out to be a rare event
(less than 1% of the time). We first identified the NPC positions and then
implemented the tracking routine.

Importin α (Imp α) binds to a protein cargo via a nuclear localiza-
tion signal (NLS) and it simultaneously binds to importin β1 (Imp β1),
an NTR that mediates interactions with the NPC[1]. After cargo release
in the nucleoplasm, Imp α is then returned to the cytoplasmic com-
partment by CAS in combination with RanGTP for another round of
cargo import[22] (Fig. 2a). Fluorescent Imp α was added to permeabilized
cells under conditions where both import (cargo translocation) and
export (Imp α recycling) occur. We positioned the nuclear envelope
near the focal plane, which yielded an effective observation window of
approximately ±400 nm in z when scanning in xy. Despite this range,
most tracking localizations were within approximately 150 nm of the
nuclear envelope, and some were near NPCs (Fig. 2c,d). Multiple filtra-
tion criteria were used to identify tracks as bona fide translocation
events (Fig. 2e and Extended Data Fig. 6). Although the low cellular

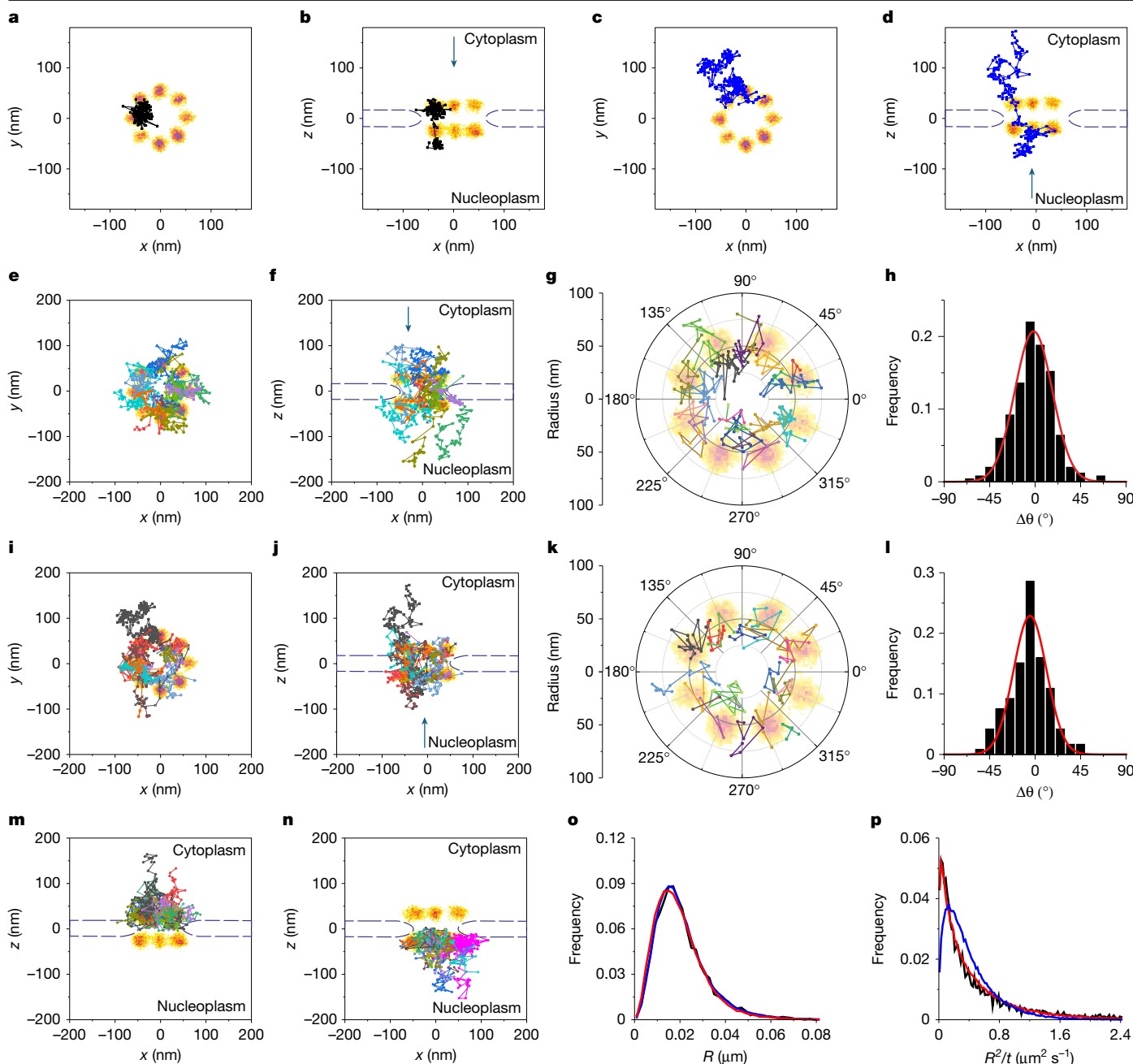

**Fig. 3 | Import and export trajectories.** Imp α–JF549 trajectories were overlaid on a composite NPC image (from Fig. 1g,h). The blue curved lines approximate the nuclear envelope. See Supplementary Table 4 for data summary and Extended Data Figs. 7 and 8 for additional trajectories. **a,b**, Representative import track observed from the cytoplasm and the side (see Supplementary Video 1). **c,d**, Representative export track observed from the cytoplasm and the side (see Supplementary Video 2). **e–h**, Import trajectories ($n = 13$ for **e,f**). The polar plot in **g** (20 tracks, $|z| \leq 25$ nm) reveals that the translocation paths are peripheral and largely confined to one of the eight lobes of the NPC scaffold, which is supported by the distribution of angular differences within the channel (**h**; Gaussian width = 19°). **i–l**, Export trajectories ($n = 17$ for **i,j**). The polar plot in **k** (20 tracks, $|z| \leq 25$ nm) reveals that the export paths are also peripheral and largely confined to one of the eight lobes of the NPC scaffold, which is

supported by the distribution of angular differences within the channel (**l**; Gaussian width = 17°). **m**, Abortive import trajectories ($n = 32$). **n**, Abortive export trajectories ($n = 35$). **o,p**, Localization error and diffusional behaviour from jump step and $R^2/t$ analysis of all 225 tracks that interacted with an NPC. The simplest simulation model (blue) assuming diffusive movement of a single species ($D = 0.049\ \mu m^2\ s^{-1}$; $\sigma_x = 4.45$ nm $= 0.93\sigma_y = 1.55\sigma_z$) does not simultaneously fit both representations of the experimental data (black) well. A three-species model (red; 7%, 56% and 37% of total, respectively) in which only species 3 undergoes diffusive movement ($D = 0.055\ \mu m^2\ s^{-1}$) fits the data much better ($\sigma_x = 4.1$ nm $= 0.93\sigma_y = 1.55\sigma_z$ for species 1, and $\sigma_x = 8.2$ nm $= 0.93\sigma_y = 1.55\sigma_z$ for species 2 and 3). Although species 1 and 2 do not have any prescribed diffusive movement, the large error for species 2 may subsume confined movements (see Extended Data Fig. 5j,k for additional models).

background yielded some false-positive tracks, these were largely eliminated with a low detector channel ratio filter (Extended Data Fig. 6c,d). From a total of 12,384 filtered tracks of 5 or more consecutive localizations from 37 cells (541 NPCs), 2,678 tracks had at least one localization within a 400-nm cube centred on an NPC. Of these,

225 tracks were deemed to have been caught transiting a pore (at least one point 25 nm or less from the midplane of an NPC) and were further analysed. With a maximal time resolution of 0.5–0.6 ms (Extended Data Fig. 1c,d), the median track length for those that entered the 400-nm NPC-centred cube was 18 points (longest = 238 points; data summary

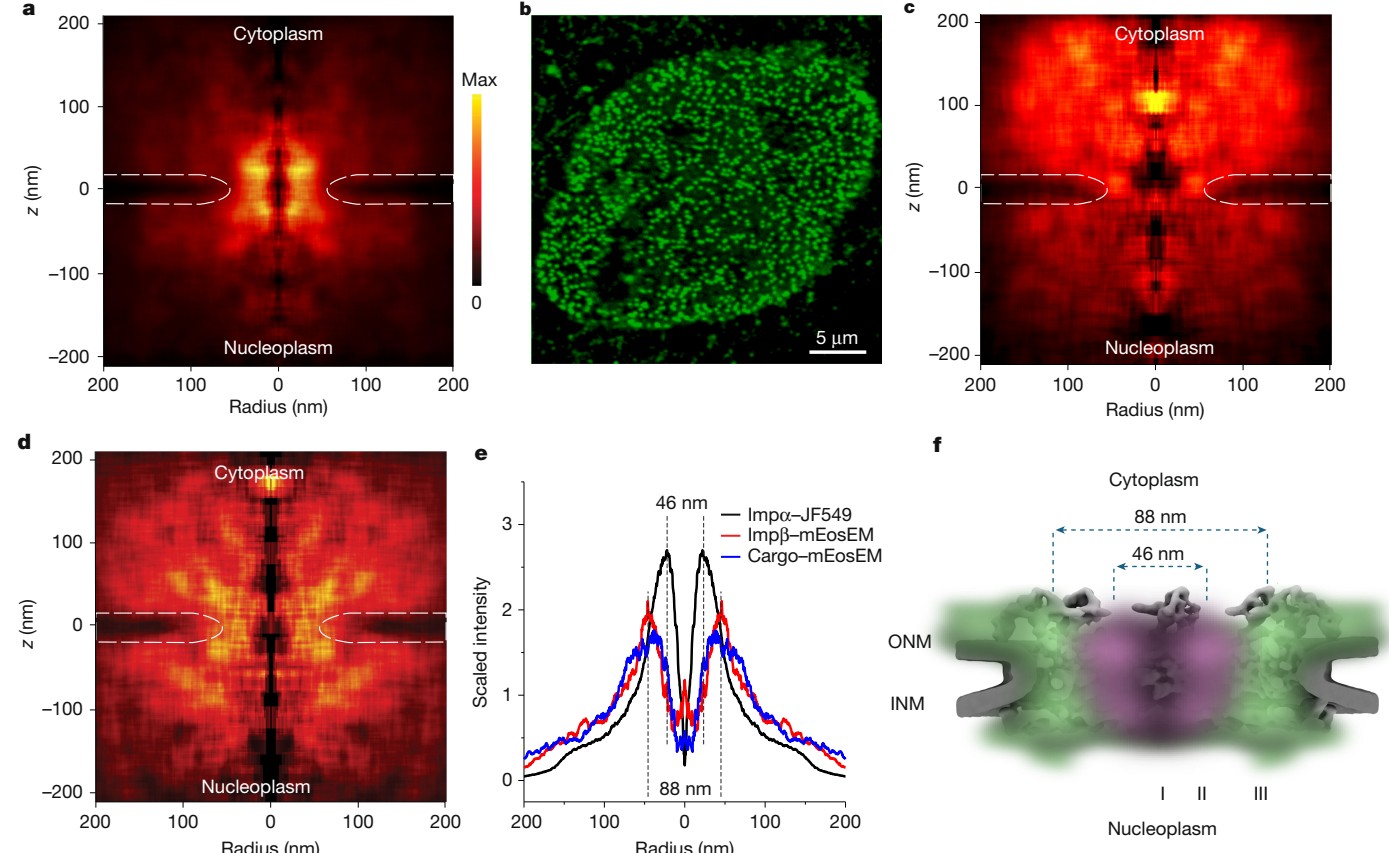

**Fig. 4 | Peripheral binding and translocation. a**, Volume-corrected radial density map for all MINFLUX localizations of Imp α–JF549 (2,678 tracks, $n$ = 48,603 localizations). **b**, Confocal image of a permeabilized U2OS cell nuclear envelope decorated with Imp β1–mEosEM (excitation = 478 nm). The punctate pattern indicates localization to NPCs. Typical result from $n$ = 41 cells. **c**, Volume-corrected radial density map of Imp β1–mEosEM showing peripheral binding within the pore. Reference NPC scaffolds were identified as described earlier[7], and mEosEM was localized by astigmatism imaging (excitation = 561 nm) after photoactivation (excitation = 408 nm; $n$ = 5,500 localizations, 48 cells, 412 NPCs; see Extended Data Fig. 9c,d and Supplementary Video 5). **d**, Volume-corrected radial density map for translocation-arrested import complexes (Imp β1/Imp α/NLS–BFP–mEosEM). Binding within the pore was largely at the periphery, yielding a largely vacant centre ($n$ = 4,361 localizations, 39 cells,

391 NPCs), similar to Imp β1 alone (**c**). The white curved lines in **a**,**c**,**d** approximate the locations of the nuclear envelope. The image in **a** is scaled from 0 (black) to maximum density (yellow); **c**,**d** are scaled to 70% and 60% of maximum, respectively, due to the hotspots at approximately 100 nm and approximately 170 nm above the pore centres (see Extended Data Fig. 9e,f for full scale). Average localization densities from **a**,**c**,**d** were calculated for $|z| \le 15$ nm. **e**, Localization densities for bound Imp β1, translocation-arrested import complexes and actively transiting Imp α. **f**, Annular zone model. Three distinct binding behaviours for Imp β1 were observed: (I) an empty centre (no binding); (II) a transport active zone (low-affinity binding of import complexes); and (III) a high-affinity binding zone for Imp β1 and import complexes. INM, inner nuclear membrane; ONM, outer nuclear membrane. The NPC structure was adapted from ref. 48, Cell Press.

in Supplementary Table 4). Although this time resolution is somewhat better than most previous studies, which utilized multiple fluorophores for increased brightness[7,15,23,24], the average number of points per trajectory was approximately fivefold larger than previously observed[7], yet required only a single fluorophore. This suggests that earlier results were significantly influenced by photobleaching and/or a high illumination flux. Of note, a median of 6 points for a single trajectory within the central pore was observed (Supplementary Table 4), compared with approximately 1 point per trajectory previously[7], allowing for the characterization of transport behaviour within the NPC transport channel, which was previously impossible.

The 3D transport trajectories identified by MINFLUX in the context of the double-ring structure of the NPC scaffold (Fig. 3) revealed multiple novel findings. Lengthy, confined interactions with the NPC permeability barrier (Fig. 3a,b, Extended Data Figs. 8 and 9 and Supplementary Video 1) suggest that transport is a discontinuous process. Slow movements both in and around the pore indicate a fairly large, highly restricted diffusional environment (Fig. 3c,d, Extended Data Figs. 8 and 9 and Supplementary Video 2). Both import and export trajectories were observed near the pore periphery, and neither were

observed near the central axis of the pore, suggesting that the opposing traffic pathways overlap (Fig. 3e–l). Within the central region of the pore ($|z| \le 25$ nm), both import and export trajectories were largely confined to an octant defined by the NPC scaffold, suggesting constraints that restrict angular movement (Fig. 3g,h,k,l). Such behaviour is consistent with defined translocation routes. Abortive import and abortive export trajectories virtually never crossed the midplane ($z$ = 0), suggesting that this was the 'point of no return' (Fig. 3m,n and Supplementary Videos 3 and 4), a conclusion also reported earlier[7].

Jump step analysis revealed that all movements of Imp α are consistent with a simulated model that includes a single diffusion coefficient ($D$ = 0.049 μm$^2$ s$^{-1}$) and $\sigma_x$ = 4.45 nm = 0.93$\sigma_y$ = 1.55$\sigma_z$ (Fig. 3o, blue curve), which is an unexpectedly good precision for molecules moving with more than 20-nm average steps. Such parameters are inconsistent with the experimental data, however, when replotted as an $R^2/t$ histogram (Fig. 3p, blue curve). A good fit to both jump step and $R^2/t$ histograms could not be obtained when assuming a two-species model (Extended Data Fig. 5j,k). Instead, a three-species model was required to simultaneously obtain a good fit to both the jump step and the $R^2/t$ histograms (Fig. 3o,p, red curves). This model assumes that

approximately 7% of molecules are tightly bound (the dye is highly immobilized), approximately 56% of molecules are highly confined (no diffusion coefficient) and yet poorly localized (probably due to localized translational movements and/or rotations of the complexes, conditions that can lead to dye displacements on the order of approximately 10 nm), and 37% of the molecules are moving with a diffusion coefficient of 0.055 $\mu m^2 s^{-1}$ (Fig. 3o,p). This model is consistent with visual inspection of the trajectories, where binding behaviour is seemingly observed as a series of localized small steps interspersed with movements that generated larger displacements (for example, Fig. 3b,d). This behaviour is consistent with a model in which NPC transport is mediated by a series of weak interactions between NTRs and the disordered polypeptides of the permeability barrier, but, in addition, translocation pauses occasionally occur, which may reflect staging sites for functionally important biochemical reactions (such as permeability barrier constraints, higher-affinity interactions, or transport complex assembly and disassembly reactions). Considering the relatively low number of trajectories recovered thus far, additional work is needed to establish and characterize translocation pause sites more thoroughly. The major findings of the jump step analysis are a direct estimate of the average diffusion coefficient, the localization precision for the moving particles (see Extended Data Fig. 5j,k for additional models) and the identification of multiple molecular species. The localization precision of approximately 7.2–8.2 nm during tracking (Fig. 3o,p and Extended Data Fig. 5j,k) was substantially worse than the approximately 4-nm precision obtained for the NPC scaffold dye (Fig. 1h), which is consistent with the worse localization precision expected due to movement.

## The vacant centre

The absence of localizations near the central transport axis (Fig. 3g,k) is consistent with our previous work[7]. To more firmly demonstrate the absence of MINFLUX localizations within the pore centre, a volume-corrected radial density map was constructed from all localizations in the vicinity of an NPC (Fig. 4a). One potential concern is that the experimental strategy might have missed fast events going through the central region of the pore. However, the fact that molecules were first localized outside the pore and then continuously tracked for tens of localizations going through the pore without crossing through the pore centre laterally suggests a barrier to access the NPC centre. To further probe accessibility, Imp β1 was tagged with the photoactivatable fluorescent protein mEosEM[25]. NPCs were saturated with Imp β1–mEosEM, and then single photoactivated mEosEM molecules were localized using 3D astigmatism microscopy at 70 ms per frame (Fig. 4b,c, Extended Data Fig. 9c,d and Supplementary Video 5). These conditions ensured that the imaged Imp β1–mEosEM was tightly bound and that it did not undergo movement within or through NPCs during imaging. Here too we observed a clear absence of molecules near the central transport axis within the pore scaffold region (Fig. 4c). Translocation-arrested import complexes were also preferentially localized to the periphery of the NPC (Fig. 4d). Both Imp β1 and import complexes were redistributed by RanGTPase activity, but still did not occupy the pore centre (Extended Data Fig. 9l,p). Cytoplasmic localizations of Imp β1 and import complexes up to approximately 200 nm away from the pore centre (Fig. 4c,d) indicate the presence of high-affinity binding sites, possibly on disordered FG-containing polypeptides of the permeability barrier that extend for long distances[5]. Lower-affinity binding sites are probably responsible for the migration observed in both import and export trajectories during movements to, from and through the pore. Although bound Imp β1 and translocation-arrested import complexes localized to the extreme periphery of the pore, active transport occurred at a radius of approximately 23 nm (current work) to approximately 30 nm (previous work[7]) from the central axis (Fig. 4e and Extended Data Fig. 9g,h). These data are consistent with a model in which the permeability barrier is divided into three distinct annular zones with respect to the binding behaviour of Imp β1: a non-binding zone (the centre), a transport active zone (approximately 50-nm diameter annulus) and a peripheral zone (strong binding of empty and cargo-loaded Imp β1; Fig. 4f).

## Discussion

This study illustrates the power of MINFLUX for high spatiotemporal precision 3D tracking of diffusing molecules in the context of a spatially resolved structure. Although millisecond-scale tracking of nuclear transport complexes has been previously described[7,9,23,24,26,27], the approximately fivefold longer trajectories reported here in 3D allowed us to uncover multiple critical properties of the NPC permeability barrier. First, transport complexes containing Imp α utilize the same or overlapping translocation conduits for both import and export, and this transport occurs in an annulus with a notable exclusion zone in the pore centre. Second, both import and export were largely confined to an octant within the pore scaffold, probably due to structural constraints established by the rotational symmetry of the NPC[20,28,29]. Third, transient pauses were observed; these may reflect staging sites for functionally important biochemical reactions, such as transport complex assembly and disassembly. Last, an apparent diffusion coefficient of approximately 0.055 $\mu m^2 s^{-1}$ (Fig. 3o,p) was obtained, which is over 40× slower than previously reported[7] and approximately 1,000× slower than free diffusion in buffer (Extended Data Fig. 10). Thus, movements within the NPC permeability barrier are comparable with those in an environment with an effective viscosity similar to glycerol[30]. Such an environment provides an effective diffusion barrier whose permeability is enhanced by selective access. The power of MINFLUX is evident by comparison with our previous work in which the same import complexes were tracked in 3D via astigmatism imaging using Imp α labelled with four dye molecules at 2-ms resolution. In this previous work, generally only one localization was observed within the NPC scaffold[7]. By contrast, the current study reports trajectories for the same import complex that have a median of seven localizations within this same region, corresponding to a median pore residence time of approximately 14 ms during import (Supplementary Table 4). One explanation is that the high illumination intensities and the greater number of dye molecules in the previous work created local photophysical effects (for example, heating, chemical modifications, among others) that influenced the transport properties of the pore. Alternatively, and more likely, is that rapid photobleaching artificially selected those molecules that transported faster, and local binding events were rejected.

An interesting feature of NPCs is their ability to conduct a massive flux of macromolecules in both directions[31], traffic that seemingly occurs simultaneously[11]. Distinct trafficking pathways for import and export would be an elegant solution to avoid unproductive collisions and opposing movements[8–12]. Two-dimensional data have been unclear in this regard[8,10,24]. The current work, however, argues against this model, as both import and export occurred in an approximately 46-nm diameter annulus, and, surprisingly, both import and export trajectories were largely confined to a single lobe of the octagonal structure, which complicates resolving encounters between traffic moving in opposing directions. Simulations indicate, however, that self-regulating mechanisms can influence binding interactions, and distributions of proteins within the permeability barrier can mitigate anticipated complications due to competition and crowding[32]. On the basis of structural propensities, it was postulated over a decade ago that regions of different FG-polypeptide densities exist within the pore[33], which could differentiate high-probability regions for translocation (for example, channels). More recently, such conduits were detected in an electron microscopy reconstruction of the permeability barrier[34]. However, we cannot rule out whether confinement during transport is influenced by direct interactions with the NPC

scaffold. The Kap-centric model predicted distinct binding regions for strongly bound (translocation arrested) and weakly bound (transiting) NTRs[35,36], which is supported by the reported data (Fig. 4), although we find that transport occurs in an annulus rather than through the centre of the pore. Of note, a recent cryo-electron microscopy study has found that large preribosomal subunits are exported near the NPC periphery[37], which undoubtedly must overlap with the transport pathways characterized here. The absence of any identified transport in this work near the central axis of the NPC remains an intriguing mystery as colloidal gold-labelled import and export cargos have been previously found to be centrally localized[11,12]. Although it is possible that the central region is used primarily for mRNA export[8,11], an alternate explanation is that a central plug[29,38] generally restricts access. Thus, the central permeability barrier is subdivided into at least three zones with respect to Imp β1: a central non-binding zone, a transport annulus ($r = 23\text{–}30$ nm from the centre) and a strongly bound annulus (pore periphery). Of note, the lamin-B receptor seemingly migrates around the pore during transport[39], suggesting a lack of confinement for membrane proteins and, hence, different structural constraints. Future studies are expected to resolve whether the permeability barrier exhibits different functional properties towards other transport receptors and transport pathways.

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

## Methods

### mEosEM fusion proteins

New plasmids were submitted to Addgene, and their construction is described in the history of the linked SnapGene files. Plasmid pET28–Imp β1–mEosEM encoding Imp β1–mEosEM with a C-terminal 6×His tag was created by attaching mEosEM to the C terminus of human Imp β1. The gene encoding Imp β1 was from pQE9-β1 (ref. 49) and the gene encoding mEosEM was from pRSET-mEosEM[25]. Plasmid pTrcHisA–NLS–BFP–mEosEM encoding NLS–BFP–mEosEM with a C-terminal 6×His tag was created by replacing the C-terminal blue fluorescent protein (BFP) domain in the import cargo NLS–2×BFP[7] with mEosEM[25]. Coding sequences were verified by DNA sequencing.

### Protein overproduction and purification

Protein overproduction and purification protocols are provided in the following sections or the indicated references. When used, antibiotics were 50 μg ml$^{-1}$ for ampicillin (Amp) and 30 μg ml$^{-1}$ for kanamycin (Kan). All purified proteins were aliquoted and stored at −80 °C until use.

**LaG-9(S151C), Imp α, Imp β, NLS–2×BFP, RanGDP and NTF2.** These proteins were overproduced in *Escherichia coli* and purified as previously described[7]. Plasmids pET21b–LaG9(S151C) and pNLS–2×BFP are available from Addgene (ID#172490 and ID#176151, respectively).

**GST–CAS.** The plasmid pGEX4T3–CAS[50] encodes GST fused to the N terminus of CAS (GST–CAS). JM109 cells[51] transformed with pGEX4T3–CAS were inoculated into 5 ml Luria-Bertani (LB) medium[52] + Amp, and then were incubated overnight at 30 °C. The following day, the starter culture was transferred to 0.5 l LB + 1% glucose + Amp and incubated at 30 °C. At an optical density at 600 nm ($OD_{600}$) of approximately 2, 2 mM isopropyl-β-D-1-thiogalactopyranoside (IPTG) was added, and the culture was incubated for another 2 h at 30 °C. The cells were harvested by centrifugation (at 5,000$g$ for 10 min at 4 °C). The cell pellet was resuspended in 1.5 ml of 10× PBS (1.37 M NaCl, 27 mM KCl, 100 mM Na$_2$HPO$_4$ and 18 mM KH$_2$PO$_4$) and 13.5 ml of H$_2$O, and then 5 mM dithiothreitol, 5 mM phenylmethane sulfonyl fluoride (PMSF) and 1 mg ml$^{-1}$ lysozyme were added. Cells were lysed by French Press (three times at 16,000 psi). The lysate was centrifuged (at 10,000$g$ for 15 min at 4 °C) and the supernatant was mixed with 0.6 ml of glutathione Sepharose beads (#GE17-0756-01, Millipore Sigma) that had been equilibrated with ice cold PBS. After rotary incubation (for 3 h at 4 °C), the suspension was transferred to a gravity column, and the resin was washed with 10 ml of PBS + 0.1 mM PMSF, 10 ml of PBS + 600 mM NaCl, and then 10 ml of PBS + 0.1 mM PMSF. The protein was eluted (1-ml fractions) with ice-cold 100 mM Tris-Cl, 120 mM NaCl and 20 mM reduced glutathione, pH 8.

**RanGAP and RanBP1.** Plasmids pQE60–RanGAP and pQE60–RanBP1 (gifts from D. Görlich)[49] were used to overproduce *Schizosaccharomyces pombe* RanGAP and mouse RanBP1, respectively, in JM109. The proteins had a C-terminal 6×His-tag and were purified identically. Cells were inoculated into 5 ml LB + Amp, and then were incubated overnight at 37 °C. The following day, the starter culture was transferred to 1 l LB + Amp and incubated at 37 °C. At an $OD_{600}$ of approximately 0.8, 1 mM IPTG was added, and the culture was incubated overnight at 25 °C. The cells were harvested by centrifugation (at 5,000$g$ for 20 min at 4 °C). The cell pellet was resuspended in 20 ml of 5 mM Tris, 200 mM NaCl, 5 mM MgCl$_2$, 10 mM imidazole, 4 mM β-mercaptoethanol (βME), pH 7.0, + protease inhibitors (1 mM PMSF, 100 μg ml$^{-1}$ trypsin inhibitor, 20 μg ml$^{-1}$ leupeptin and 100 μg ml$^{-1}$ pepstatin A), and then lysed by French Press (three times at 16,000 psi). The lysate was centrifuged (15,000$g$ for 20 min at 4 °C), and the supernatant was mixed with 0.5 ml Ni-NTA resin (rotated for 30 min at 4 °C). The suspension was transferred to a gravity column, and the resin was washed with 20 ml of 5 mM Tris, 500 mM NaCl, 5 mM MgCl$_2$, 0.1% Triton X-100, pH 8.0, + protease inhibitor and then 20 ml of 5 mM Tris, 50 mM NaCl, 5 mM MgCl$_2$, 20 mM imidazole, pH 8. The protein was eluted (1-ml fractions) with 5 mM Tris, 50 mM NaCl, 5 mM MgCl$_2$ and 250 mM imidazole, pH 8. The protein in the highest concentration fraction was purified by Enrich SEC 650 (7801650, Bio-Rad) size-exclusion chromatography using 20 mM HEPES, 110 mM potassium acetate (KOAc), 5 mM sodium acetate (NaOAc), 2 mM magnesium acetate (MgOAc$_2$) and 2 mM dithiothreitol, pH 7.4.

**Imp β1–mEosEM.** BL21(DE3) cells[53] transformed with plasmid pET28a–Imp β1–mEosEM were inoculated into 5 ml LB supplemented with 2% glucose + Kan, and then were incubated overnight at 37 °C. The following day, the starter culture was transferred to 1 l of LB medium with 2% glucose and Kan and incubated at 37 °C. At an OD of approximately 0.8, 0.7 mM IPTG was added, and the culture was incubated for another 3 h at 37 °C. The cells were harvested by centrifugation (at 5,000$g$ for 10 min at 4 °C). The cell pellet was resuspended in 6 ml of 5 mM Tris, 500 mM NaCl, 5 mM MgSO$_4$, 10 mM imidazole, 4 mM βME, pH 8.0, + protease inhibitors and then lysed by French Press (three times at 16,000 psi). The lysate was centrifuged (at 15,000$g$ for 20 min at 4 °C), and the supernatant was mixed with 0.5 ml Ni-NTA resin (rotated for 30 min at 4 °C). The suspension was transferred to a gravity column, and the resin was washed with 20 ml of 5 mM Tris, 500 mM NaCl, 5 mM MgSO$_4$, 10 mM imidazole, 4 mM βME, 0.1% Triton X-100, pH 8.0, + protease inhibitors and then 20 ml of 5 mM Tris, 100 mM NaCl and 10 mM imidazole, pH 8. The protein was eluted (500-μl fractions) using 5 mM Tris, 100 mM NaCl and 250 mM imidazole, pH 8.0. The highest concentration fractions were combined, and the protein was purified by Enrich SEC 300 size-exclusion chromatography using 20 mM HEPES, 110 mM KOAc, 5 mM NaOAc, 2 mM MgOAc$_2$ and 2 mM dithiothreitol, pH 7.4.

**NLS–BFP–mEosEM.** JM109 cells[51] transformed with plasmid pTrcHisA–NLS–BFP–mEosEM–6×His were inoculated into 5 ml LB + Amp, and then were incubated overnight at 37 °C. The following day, the starter culture was transferred to 1 l LB + Amp and incubated at 37 °C. At an OD of approximately 0.8, 0.7 mM IPTG was added, and the culture was incubated for 14 h at 25 °C. The remainder of the protein purification protocol was identical to that for Imp β1–mEosEM.

### Protein labelling

The spontaneously blinking dye HMSiR maleimide (SaraFluor 650B-maleimide; A209-01, Goryo Chemical) was attached to the C-terminal cysteine on the anti-GFP nanobody LaG-9(S151C) by incubating with a 15-fold molar excess at room temperature for 15 min to yield Nb$^{GFP}$–HMSiR. Imp α was under labelled with JF549 maleimide (Janelia Fluor 549 maleimide; 6500, Tocris) by incubating with a 10-fold molar excess at room temperature for 15 min in 20 mM HEPES, 150 mM NaCl, pH 7.4 (maximum reaction volume of 1 ml) to yield Imp α–JF549. On the basis of the concentration-dependent labelling efficiency, this molar excess produced approximately 25% labelling saturation (approximately 1 cysteine labelled, on average, out of the 4 available reactive cysteines). The reactions were quenched with 10 mM βME. Excess dye was removed by adding the dye–protein mixture to 0.1 ml Ni-NTA resin (30-min incubation), washing the resin-bound protein with 50 ml of 20 mM HEPES, 500 mM NaCl, 0.1% Triton X-100, pH 7.3, and 20 mM HEPES and 150 mM NaCl, pH 7.3, and then eluting the labelled proteins with 20 mM HEPES, 150 mM NaCl and 250 mM imidazole, pH 7.3.

### Protein concentrations and labelling purity

Protein concentrations were determined by densitometry using SDS–PAGE gels stained with Coomassie Blue R-250 with bovine serum albumin as a standard and a ChemiDoc MP imaging system (Bio-Rad Laboratories). The purity of dye-labelled proteins was more than 95%, as determined by in-gel fluorescence imaging using the same ChemiDoc imaging system.

## Cell culture

For MINFLUX imaging, U2OS-CRISPR–NUP96–mEGFP clone #195 (300174, CLS GmbH) cells were grown in Dulbecco's modified eagle medium (11880028, Thermo Fisher Scientific) supplemented with 1× MEM non-essential amino acids solution (11140050, Thermo Fisher Scientific), 1× GlutaMAX solution (35050061, Thermo Fisher Scientific), 1× ZellShield (13-0050, Minerva Biolabs) and 10% (v/v) fetal bovine serum (F7524, Sigma) in 5% (v/v) $CO_2$ enriched air at 37 °C. Cells were typically grown to approximately 80% confluency and split using TrypLE Express (12604013, Thermo Fisher Scientific) without phenol red.

For 3D astigmatism and Luminosa confocal imaging, U2OS-CRISPR–NUP96–mEGFP clone #195 and U2OS (300364, CLS GmbH) cells were grown in McCoy's 5A (modified) media (16600082, Thermo Fisher Scientific) supplemented with 100 U ml$^{-1}$ penicillin–streptomycin (15140148, Thermo Fisher Scientific), 1 mM sodium pyruvate (11360070, Thermo Fisher Scientific), 1× MEM non-essential amino acids solution (11140050, Thermo Fisher Scientific) and 10% (v/v) fetal bovine serum (A3160401, Thermo Fisher Scientific) in 5% (v/v) $CO_2$ enriched air at 37 °C. Cells were typically grown to approximately 95% confluency and were split using Accutase (A1110501, Thermo Fisher Scientific). Cells were grown from fresh stocks and used within 1 month; they were not tested for mycoplasma contamination or authenticated.

## 3D MINFLUX imaging

**Microscope system.** A MINFLUX 3D microscope (Abberior Instruments) was used for all MINFLUX localization and tracking experiments. A 100× oil immersion objective lens (UPL SAPO100XO/1.4, Olympus) and 488-nm, 561-nm and 642-nm CW excitation lasers were used for mEGFP confocal imaging, cargo tracking and NPC scaffold localization, respectively. Four avalanche photodiodes (SPCM-AQRH-13, Excelitas Technologies) with detection ranges of 500–550 nm, 580–630 nm, 650–685 nm and 685–760 nm were used with a pinhole size corresponding to 0.78 airy units. All hardware was controlled by Abberior Imspector software (v16.3.13924-m2112). Drift during tracking was minimized using the built-in stabilization system with typical drifts less than 1 nm in *xyz*. Scattering from 200 nm gold nanoparticles (A11–200-CIT-DIH-1-10, Nanopartz) pre-deposited on the coverslip surface and at a similar *z* height to the bottom of the nucleus were used as a positional reference for the sample stabilization and two-colour alignment registration.

**Sample preparation.** Six channel µ-Slide VI 0.5 glass bottom slides (80607, Ibidi) were pre-treated with 200 nm gold nanoparticles (used as a positional references) and poly-L-lysine (which reduced cell-detachment after permeabilization). Undiluted 200 nm gold nanoparticles (A11-200-CIT-DIH-1-10, Nanopartz) were added to each channel, and 15 min later were washed away with 1× PBS. Then, 50 µl of 0.01% poly-L-lysine (P4832, Sigma) was added to each lane. After 5 min, the lanes were washed with cell culture media (3 × 100 µl). Freshly split U2OS NUP96–mEGFP cells (to less than 60% confluence) were grown overnight on the pre-treated coverslips. The next day, the cells were washed with 50 µl of import buffer (20 mM HEPES, 110 mM KOAc, 5 mM NaOAc, 2 mM MgOAc$_2$ and 1 mM EGTA, pH 7.3) and then permeabilized by the addition of 2 × 50 µl of 40 µg ml$^{-1}$ digitonin in import buffer for 3 min. Permeabilized cells were washed once with 50 µl import buffer–polyvinylpyrrolidone (IB-PVP; import buffer containing 1.2% (w/v) PVP (360,000 g mol$^{-1}$; P5288, Sigma)). This permeabilization method was slightly modified from Yang et al.[26] to accommodate the different cell line and growth conditions. Nb$^{GFP}$–HMSiR in IB-PVP (40 µl, 150 nM) was incubated with the permeabilized cells for 6 min. The cells were washed twice (2 × 40 µl IB–PVP). 'Transport mix' (40 µl), consisting of 1.5 µM RanGDP, 1.5 µM NTF2, 1.0 µM RanGAP, 1.0 µM RanBP1, 1 mM GTP, 0.5 µM Imp β1, 0.5 µM NLS–2×BFP, 2 µM GST–CAS and 1 nM Imp α–JF549 in IB-PVP was added to the permeabilized nanobody-tagged cells, and MINFLUX imaging begun approximately 1 min after addition.

**Imaging and tracking.** A diagonal scanning approach with a grid spacing of 300 nm was used to find both fluorophores, and then an iterative strategy within $z = 0 ± 400$ nm was used to localize HMSiR on the NPC scaffold or to track Imp α–JF549 in 3D. Non-default measurement parameters for the individual MINFLUX scan iterations are summarized in Supplementary Tables 1 and 3. The pooled results reported here were acquired using slightly different measurement parameters to optimize signal acquisition over background contributions. The data obtained via these distinct imaging conditions are identified as datasets 1 and 2 (see Supplementary Tables 1 and 3). Approximately 15–20 min was used for Nb$^{GFP}$–HMSiR localizations, and approximately 15–20 min was used for tracking cargo transport. Confocal images of the mEGFP fluorescence (excitation = 488 nm) from NUP96–mEGFP in the region of interest were obtained at the beginning and end of MINFLUX imaging.

**Channel alignment.** Coordinates were transformed from the JF549 emission channel to the HMSiR emission channel, which was the reference. The *xyz* coordinates from the same 8–10 gold nanoparticles (200 nm) on each cell were measured every few seconds during the two independently recorded MINFLUX datasets (HMSiR imaging and cargo tracking). During post-processing, the gold nanoparticle coordinates from the two datasets were used to derive an *xy* alignment matrix incorporating rotational and translational corrections, as described earlier[7], which was used to transform the JF549 coordinates to the HMSiR coordinate system with a precision of approximately 2 nm. A correction factor of 0.67 as determined in Extended Data Fig. 4 was applied to all *z* values. The mean *z* position for the gold nanoparticles associated with each cell differed by 5–14 nm between the two channels, and the *z* coordinate was corrected by simple subtraction of this mean *z* deviation.

**Data filtering parameters.** Multiple parameters providing information about the photons collected in the MINFLUX scan patterns were used for data filtering. These are reported as frequencies (in kHz) or ratios. The frequencies are discrete variables, as they were obtained by dividing the number of photons collected (that is, a quantized variable) during a collection period:

**Effective frequency at centre.** The effective frequency at centre (EFC) is the emission frequency measured at the centre of the MINFLUX scan pattern.

**Effective frequency at offset.** The effective frequency at offset (EFO) is the averaged emission frequency measured over all points in the *xy* plane of the MINFLUX scan pattern except for the centre.

**Centre frequency ratio.** The centre frequency ratio (CFR) is the ratio of the EFC and EFO, that is, CFR = EFC/EFO. The CFR is a measure of the quality of a localization. As the fewest photons are collected when the centre of the excitation donut coincides with the fluorophore position, a high CFR can indicate that the centre of the scan pattern is not well localized to the position of the fluorophore or that a second fluorophore is close by. Thus, lower CFR values indicate good localizations.

**Detector channel ratio.** The detector channel ratio (DCR) is the fractional component of photons collected in one of two channels. It is used to distinguish fluorophores with different emission spectra present within the same sample, and it can be effective for eliminating some background signals. For example, if detector channel 1 collects an emission frequency in the 650- to 685-nm range (EF1), and detector channel 2 collects an emission frequency in the 580- to 630-nm range (EF2), the detector channel ratio = EF1/(EF1 + EF2).

**Identifying NPC scaffolds from HMSiR localizations.** 3D HMSiR localizations were obtained using an eight iteration MINFLUX sequence (Supplementary Table 1). Iterations 1–6 were used first to locate the molecule and then for progressively increased localization precision. The final output coordinates came from iteration 7 (*xy*) and iteration 8 (*z*). The CFR upper limit was set to 0.8 (see Extended Data Fig. 6a) and

was checked within iteration 7. This CFR check occurred at the level of data acquisition to help eliminate acquisitions where two nearby HMSiR molecules were simultaneously in the 'on' state. Successive localizations occurred by cycling between iterations 7 and 8 until the CFR check failed, or the fluorophore switched off for longer than 3 ms. To distinguish HMSiR localizations from background noise, an EFO lower limit of 25 kHz or 50 kHz (for datasets 1 and 2, respectively) was used during acquisition. An upper EFO threshold of 60 kHz or 100 kHz (for datasets 1 and 2, respectively) was used post-acquisition to eliminate signals from multiple dye molecules that were not eliminated by the CFR check during acquisition (see Extended Data Fig. 6b). HMSiR localizations were exported using 'MINFLUX-BASE Imspector 16.3.15620' and Paraview 5.8.1 software.

Individual NPCs were identified, and averaged NPC scaffolds were generated essentially as done previously using localizations from astigmatism imaging[7]. This approach is briefly outlined here and in Extended Data Fig. 3. The localizations in each of the individual NPC localization clusters were fit to a double-circle model, which reflected the double-ring structure of NUP96 within the NPC (Fig. 1a). Owing to the relatively flat nuclear envelope, we assumed that the two circles were both parallel to the $xy$ plane with their centres defining an axis parallel to the $z$ axis. Although the nuclear envelope was not perfectly flat (for example, Fig. 1d), the angular tilt of the NPCs used was less than 10°, consistent with our previous analysis[7]. The angles of the individual localizations relative to the centroid obtained from the double-circle fit for each NPC were binned (0–45°), assuming an eightfold periodicity, and fit to a sinusoidal function with a 45° period and a variable phase. The individual localization clusters were rotated in the $xy$ plane about their $xyz$ centroids using the determined phase angle, and then these clusters were aligned on the basis of their $xyz$ centroids to yield averaged NPC scaffolds (for example, Fig. 1g,h and Extended Data Fig. 3e). All the fitting routines were performed using MATLAB scripts available on GitHub (https://github.com/npctat2021/MINFLUX_NPC_Tracking.git).

**Measuring the z-scaling factor.** The $z$-axis data in 3D MINFLUX imaging required a correction to account for spherical aberrations caused by the difference in refractive index of the sample (approximately 1.33) and that of the immersion medium (1.51) used with the objective. The $z$-scaling factor of 0.7 recommended by the manufacturer of the MINFLUX microscope was calculated on the basis of simulations[54]. This $z$-scaling factor has been applied previously to 3D MINFLUX data[2,16,17] and has been directly measured as 0.69 (ref. 55). The $z$-scaling factor used here was estimated from independent measurements of NPC scaffold structures as determined by HMSiR astigmatism imaging (see the '3D astigmatism imaging' section). The advantage of this approach is that both MINFLUX and astigmatism measurements were made under identical conditions (the same buffer and added reagents, cell type, nanobodies, HMSiR labelling conditions and range of $z$ heights above the surface) and the astigmatism measurements were independently calibrated for each day's experiments by imaging beads while $z$ stepping a nanostage[7]. As the $z$ spacing between the two rings of the NPCs at the bottom of cell nuclei can be assumed to be identical for both imaging strategies, the astigmatism ring spacing of 51.5 ± 1.1 nm was used to correct the raw MINFLUX data, where the ring spacing was determined as 76.8 ± 0.8 nm (see Supplementary Table 2 and Extended Data Fig. 4). The $z$-scaling factor was therefore calculated as 51.5 nm/76.8 nm = 0.67. This value was used for all MINFLUX $z$-axis scalings. A direct comparison of the MINFLUX and astigmatism NPC images and the corresponding data is shown in Extended Data Fig. 4, and a summary of the associated parameters is given in Supplementary Table 2.

**Identifying Imp α trajectories from JF549 localizations.** After collecting the NPC scaffold data, the scanning and localization protocol for JF549 was implemented and continued for approximately 15–20 min.

For tracking Imp α–JF549, the five iteration MINFLUX sequence was designed to be as fast as possible with an octahedral scan pattern in the last iteration, which yielded an $xyz$ localization within a single step (Supplementary Table 3). The fifth iteration was repeated if insufficient photons were collected (20 or 25 minimum; see Supplementary Table 3) or to yield the next localization within the trajectory until the particle was lost. Unlike for HMSiR localizations in which the CFR check during imaging was set to a low value to select for high-quality localizations at the time of acquisition, for tracking, the CFR ratio was set to a large cut-off (more than 2.0) to avoid rejecting tracks that were temporarily interrupted. Instead, the CFR was checked during data analysis of the localization data. All detected tracks within the acquisition volume ($z = 0 \pm 400$ nm) were identified with Abberior Imspector 16.3.15620 and Paraview 5.8.1 software and exported to MATLAB format. Imp α– JF549 trajectories were converted into red channel (HMSiR) coordinates via the alignment procedure discussed earlier. The data were curated by eliminating those tracks that did not have any localizations within a 400-nm cube centred on an NPC. Each trajectory was then rotated by the same angle as the NPC that it was linked to. The alignment routine was performed using MATLAB scripts available on GitHub (https://github.com/npctat2021/MINFLUX_NPC_Tracking.git). For trajectories that entered an NPC scaffold ($|z| \le 25$ nm), the tracks were verified as authentic using the criteria summarized in Extended Data Fig. 6c–f. Although there was some background fluorescence and leakage from HMSiR fluorescence within the permeabilized cells, tracks that resulted from this background were effectively eliminated with a DCR filter (Extended Data Fig. 6c,d). On the basis of the earlier results[56], we used a CFR of less than 0.8 (Extended Data Fig. 6e). The EFO was used to eliminate background signals, but it also indicated that approximately 10–15% of transport trajectories had two JF549 dyes on Imp α instead of one (Extended Data Fig. 6f). This was an expected consequence of the under-labelling strategy. The reference angle in Fig. 3h,l was the localization nearest the pore midplane ($z = 0$), except for three cases of poor localization precision.

## 3D astigmatism imaging

**Microscope system.** The 3D astigmatism microscope system was described earlier[7] and was used here without modification except that a ×2 magnifying lens was removed[57] and either a Prime 95B or Kinetix22 CMOS camera (both from Teledyne Photometrics) was used for imaging, which yielded square pixels of 120 nm and 138 nm at camera plane, respectively. A TIRF-lock system provided a $z$ stability for the coverslip of less than 3 nm for the duration of the experiment. The astigmatism was set to 60-nm root mean square (rms) deviation using a deformable mirror to generate the $z$-dependent spot ellipticity needed for 3D information. Orange (mEosEM) and red (HMSiR) fluorescence emission were collected with a quad-bandpass filter set (ZT405/488/561/640/rpcv2-UF2, Chroma). Data collection on this Zeiss 200M microscope was acquired using Micro-Manager 2.0 (ref. 58). The Mirao 52-e deformable mirror system (Imagine Optic) used for wavefront correction and to create astigmatism utilized CasAO 1.0 and MiCAO 1.3. The Nano-LPS200 piezo nano-positioning stage controlling a TIRF-lock stabilization system (Mad City Labs) was controlled by LabVIEW 2015. ImageJ (Fiji 1.52P), Origin 8.5, Kaleidagraph 5.01 and Microsoft Excel 16.76 (23081101) were used for data analysis, data presentation and simulations.

**Sample preparation.** Freshly split U2OS NUP96–mEGFP cells were grown overnight at less than 60% confluence on #1.5 coverslips (24 × 60 mm; 16004-312, VWR), which were pretreated with 0.01% poly-L-lysine (P4832, Sigma) for 10 min at room temperature and air-dried overnight. The next day, flow chambers (approximately 10 µl) were constructed by inverting a small coverslip (10.5 × 35 mm; 72191-35, Electron Microscopy Sciences) with beads of high-vacuum grease parallel to its short edges over the cells[59]. Cells within the flow chambers were permeabilized by incubating with digitonin (40 µg ml⁻¹)

in import buffer for 3 min. Permeabilized cells were washed once with 10 μl IB–PVP. Then, 10 μl of 150 nM Nb$^{GFP}$–HMSiR in IB–PVP was flowed onto the permeabilized cells and incubated for 3 min. The cells were washed (2 × 10 μl IB–PVP) and then Imp β1–mEosEM (0.5 μM) or a mixture of Imp β1 (0.5 μM), Imp α (0.5 μM) and NLS–BFP–mEosEM (0.5 μM) was added. After 10 min, the permeabilized cells were washed twice (2 × 10 μl IB–PVP) to remove unbound proteins. For Extended Data Fig. 9j–l, cells with bound Imp β1–mEosEM were incubated with 'Ran mix' (2 × 10 μl; 1.5 μM RanGDP, 1.5 μM NTF2, 1 mM GTP, 1 μM RanBP1 and 1 μM RanGAP in IB–PVP) for 10 min, and then washed (2 × 10 μl IB–PVP). For Extended Data Fig. 9n–p, cells with bound Imp NLS–BFP–mEosEM (cargo complexes) were incubated with 'transport mix–high-α' (2 × 10 μl; 1.5 μM RanGDP, 1.5 μM NTF2, 1.0 μM RanGAP, 1.0 μM RanBP1, 1 mM GTP, 0.5 μM Imp β1, 0.5 μM Imp α, 0.5 μM NLS–2×BFP and 2 μM GST–CAS in IB–PVP) for 10 min, and then washed (2 × 10 μl IB–PVP). Note that 'transport mix–high-α' is the same composition of proteins used in the simultaneous import–export MINFLUX experiments ('transport mix'), except with a higher concentration of Imp α (non-fluorescent).

**3D localizations.** HMSiR localizations were acquired first (excitation = 641 nm; 50 ms per frame, twenty 500-frame videos with a 5-s gap between videos), and then the photoactivatable mEosEM was imaged (excitation = 561 nm; 70 ms per frame, thirty 1,000-frame videos with a 5-s gap between videos) in the presence of constant UV illumination (activation laser, 408 nm). Data analysis to identify and align NPC scaffolds and to then align mEosEM localizations with these scaffolds was performed as described earlier[7]. With minimum photon counts of 3,000 and 1,000 for HMSiR and mEosEM, respectively, the average precisions were 4.2–5.4 nm and 6–8.3 in $xy$, and 8.7–9.9 nm and 14.6–15.2 nm in $z$, determined as previously described[7,57]. Note that the precisions varied slightly for the two cameras used[57].

**Channel alignment.** The two channels were aligned by imaging five 0.1-μm TetraSpeck microspheres (T7279, Thermo Fisher Scientific) embedded in 2% agarose, as previously described[7], with a precision of 1–2 nm ($xy$) and 3–7 nm ($z$).

**Confocal imaging and fluorescence correlation spectroscopy**
Confocal imaging and fluorescence correlation spectroscopy (FCS) measurements were performed with a Luminosa single-photon counting confocal microscope (Picoquant). For confocal imaging of NPCs in U2OS cells decorated with mEosEM fusion proteins, samples were prepared as for astigmatism imaging, except that the final wash step before adding Nb$^{GFP}$–HMSiR and the mEosEM protein was 3 × 10 μl IB–PVP. Imaging of mEosEM was performed using 478-nm excitation in continuous-wave mode. FCS experiments of purified proteins were performed in buffers as indicated in Extended Data Fig. 10 using the FCS measurement function of the Luminosa system software (Luminosa 1.0.0.4067).

**Error estimation and dye movement analysis**

**Jump histograms.** Jump probability histograms summarize the measured distances between two successive localizations, either for static particles (for example, on the NPC scaffold) or for moving particles (for example, cargo movement around and through the NPC). The following jump probability distributions (equations (1)–(4)) were summarized earlier[7] and are provided here for completeness. These distributions assume anisotropic translational movement described by a diffusion coefficient, $D$. All distributions are normalized (that is, the sum of all probabilities = 1):
**One dimension:**

$$p(x; D, t)dx = \frac{2b}{\sqrt{4\pi Dt}} \exp\left(-\frac{x^2}{4Dt}\right)dx \tag{1}$$

where $t$ is the time between successive localizations, $b$ is the bin size of the jump distances, and $x$ is the measurement axis for the displacements. This is a Gaussian (normal) distribution, except that we have assumed that all the jump distances are positive for easier comparison with the 2D and 3D cases, which necessitates the factor of 2.
**Two dimensions:**

$$p(r; D, t)dr = \frac{br}{2Dt} \exp\left(-\frac{r^2}{4Dt}\right)dr \tag{2}$$

where $r^2 = x^2 + y^2$.
**Three dimensions:**

$$p(R; D, t)dR = \frac{bR^2}{\sqrt{4\pi}(Dt)^{\frac{3}{2}}} \exp\left(-\frac{R^2}{4Dt}\right)dR \tag{3}$$

where $R^2 = x^2 + y^2 + z^2$. If there are distinct molecular populations with different diffusion coefficients, a weighted sum can be generated. For example, for two species in 3D:

$$p(R; D_1, D_2, A, t)dR = [Ap(R; D_1, t) + (1 - A)p(R; D_2, t)]dR$$

or,

$$\begin{aligned} p(R; D_1, D_2, A, t)dR \\ = \frac{bR^2}{\sqrt{4\pi}}\left[\frac{A}{(D_1 t)^{\frac{3}{2}}}\exp\left(-\frac{R}{4D_1 t}\right) + \frac{1 - A}{(D_2 t)^{\frac{3}{2}}}\exp\left(-\frac{R}{4D_2 t}\right)\right]dR \end{aligned} \tag{4}$$

where $A$ is a weighting factor for the two distributions. In this case, there are three fitting parameters, $D_1$, $D_2$ and $A$ for a fixed time step $t$.

**Time-independent histograms.** The jump probability histograms described by equations (1)–(4) are valid for equal time steps ($t$ = constant). For larger $t$, the distributions become broader, a consequence of larger average jump distances. MINFLUX localizations are typically rejected unless a minimum number of photons are collected; this is typically rectified by repeating the localization process until the minimum photon limit is met, leading to unequal timesteps (see Extended Data Fig. 1). To correct for unequal timesteps, the probability distributions for isotropic diffusion can be converted to time-independent expressions with a change of variables. For completeness, this is demonstrated here for all three cases, although only the approach of the 3D case was used to analyse data (for example, Fig. 3p and Extended Data Fig. 5).
**One dimensions:**
Assume, $u = \frac{x^2}{t}$
Then: $du = \frac{2x}{t}dx$ and $\sqrt{u} = \frac{x}{\sqrt{t}}$
From equation (1),

$$\begin{aligned} p(x; D, t)dx &= \frac{2b}{\sqrt{4\pi Dt}} \exp\left(-\frac{x^2}{4Dt}\right)dx \\ &= \frac{2b}{\sqrt{4\pi Dt}} \frac{t}{2x} \exp\left(-\frac{x^2}{4Dt}\right)\frac{2x}{t}dx \\ &= \frac{b}{\sqrt{4\pi D}} \frac{\sqrt{t}}{x} \exp\left(-\frac{x^2}{4Dt}\right)\frac{2x}{t}dx \end{aligned}$$

or,

$$p(u; D)du = \frac{b}{\sqrt{4\pi Du}} \exp\left(-\frac{u}{4D}\right)du \tag{5}$$

## Two dimensions:

Assume, $u = \dfrac{r^2}{t}$

Then: $du = \dfrac{2r}{t}dr$

From equation (2),

$$p(r;D,t)dr = \frac{br}{2Dt}\exp\left(-\frac{r^2}{4Dt}\right)dr = \frac{b}{4D}\exp\left(-\frac{r^2}{4Dt}\right)\frac{2r}{t}dr$$

or,

$$p(u;D)du = \frac{b}{4D}\exp\left(-\frac{u}{4D}\right)du \tag{6}$$

## Three dimensions:

Assume, $u = \dfrac{R^2}{t}$

Then: $du = \dfrac{2R}{t}dR$ and $\sqrt{u} = \dfrac{R}{\sqrt{t}}$

From equation (3),

$$p(R;D,t)dR = \frac{bR^2}{\sqrt{4\pi}(Dt)^{\frac{3}{2}}}\exp\left(-\frac{R^2}{4Dt}\right)dR$$

$$= \frac{b}{2\sqrt{4\pi}(D)^{\frac{3}{2}}}\left(\frac{R}{\sqrt{t}}\right)\exp\left(-\frac{R^2}{4Dt}\right)\frac{2R}{t}dR$$

or,

$$p(u;D)du = \frac{b\sqrt{u}}{4\sqrt{\pi}(D)^{\frac{3}{2}}}\exp\left(-\frac{u}{4D}\right)du \tag{7}$$

If there are distinct molecular populations with different diffusion coefficients, a weighted sum can be generated. For two species in 3D:

$$p(u;D_1,D_2,A)du = [Ap(u;D_1) + (1-A)p(u;D_2)]du$$

or,

$$p(u;D_1,D_2,A)du = \frac{b\sqrt{u}}{4\sqrt{\pi}}\left[\frac{A}{(D_1)^{\frac{3}{2}}}\exp\left(-\frac{u}{4D_1}\right) + \frac{1-A}{(D_2)^{\frac{3}{2}}}\exp\left(-\frac{u}{4D_2}\right)\right]du \tag{8}$$

where $A$ is a weighting factor for the two distributions. In this case, there are three fitting parameters, $D_1$, $D_2$, and $A$.

**Simulations.** Simulations were performed using the Jump Step Histogram Simulator[60] to demonstrate the time independence of equation (7). Three-dimensional translational steps were randomly selected from a normal distribution centred around the starting position with a step variance (Var) of $Var(R) = 6Dt$ [with $Var(x) = 2D_xt$, $Var(y) = 2D_yt$, and $Var(z) = 2D_zt$]. The simulations yielded diffusional trajectories unconstrained by boundary conditions (such as the NPC scaffold) with a defined number of localizations per trajectory. In the case of Nb$^{GFP}$–HMSiR localizations, diffusional trajectories correspond to sample drift. For the Imp α–JF549 localizations, the diffusional trajectories approximate the jump steps of the protein interacting with the NPC. The simulation program allows for the inclusion of both sample drift and particle movement simultaneously, modelled identically. When the jump ($R$) histograms shown in Extended Data Fig. 5a were replotted as $R^2/t$ histograms, the time dependence vanished (Extended Data Fig. 5b), as predicted by equation (7). The Jump Step Histogram Simulator program[60] includes additional features to approximate potentially relevant motions of the dye. To approximate a 'jiggle', that is, dye movement around a central location, the position of the dye was selected from a normal distribution around a centroid with σ = jiggle. Jiggle was used to approximate the movement of the HMSiR dye around its attachment point to the NPC scaffold, that is, the linkage error, or the localized, constrained movement of Imp α–JF549 within the NPC permeability barrier network. Particle movement was modelled on the basis of its centroid. Hence, rotational error (similar to the linkage error) results from the distance between the dye on the surface of the protein and the particle centroid, for example, rotation of the import and export complexes within the NPC permeability barrier. This rotational error (sphere rotation) was modelled by randomly selecting a position on the surface of a sphere with a defined radius. Both jiggle and rotational error were assumed to equilibrate rapidly such that randomization occurred between timesteps, but nonetheless that distinct positions were populated during measurements.

The probability distributions in equations (1)–(8) do not account for the precision of the measurements. This precision was included in the simulation program (Jump Step Histogram Simulator[60]) by assuming a normally distributed measurement precision (σ) for all localizations. For MINFLUX, $\sigma_z$ is typically less than $\sigma_x \approx \sigma_y$ (refs. 2,17), so these were independently adjustable in the simulations; nonetheless, on the basis of Fig. 1i, $\sigma_x/\sigma_y = 0.93$ and $\sigma_x/\sigma_z = 1.55$ were assumed for all simulations. The effects of localization precision on the jump and $R^2/t$ histograms are shown in Extended Data Fig. 5c,d. Of note, in the presence of localization precision, the $R^2/t$ histograms were no longer independent of $t$ (Extended Data Fig. 5d). This finding led to a wider conclusion that illustrates an important feature of $R^2/t$ histograms, namely, that $R^2/t$ histograms have different sensitivities to parameters influencing jump step histograms. The magnitude of jump steps squared ($R^2$) for sample drift or particle movements depends on $t$ and, hence, dividing by $t$ makes the values time independent (Extended Data Fig. 4b). However, localization precision, jiggle (linkage error) and rotational error do not depend on time (as modelled here): they yield similar effects on the measurements of $R$ irrespective of the length of the time step. Consequently, they yield effects on $R^2/t$ histograms that, as a group, are distinct from the effects of $D$. This dichotomy is dramatically illustrated in Fig. 3o,p. Although jump histograms could be fit with multiple combinations of parameters, $R^2/t$ histograms were substantially more difficult to fit. Consequently, reproducing both histograms with the same parameters significantly increased confidence in the goodness of fit. Critically, the time steps and precision must be accounted for when using simulations to approximate the experimental data.

Simulations that include localization precision, diffusional drift and variable time steps were used to approximate the data in Fig. 1j. As the MINFLUX measurements yielded variable timesteps, the simulations included timesteps distributed according to the observed experimental frequencies (Extended Data Fig. 1). As these data in Fig. 1j reflect the distances between successive localizations of HMSiR fluorophores affixed to or tethered to static objects (NPCs), inclusion of a non-zero diffusion coefficient to improve the fit was not expected but turned out to be the most straightforward way to fit the data. We expect that this 'diffusional drift' is not sample drift that would generate large displacements over the total imaging time, as this would make the assembly of the NPC scaffold from the data impossible. Rather, the diffusional drift is confined and models the movement of the dye centroid on the NPC scaffold, for example, to favourable positions enabled by linkage error, or conformational shifts or jiggles of the NPC scaffold. This diffusional drift was instrumental to reproduce the experimental centroid deviations from simulated data (Fig. 1i and Extended Data Fig. 5i).

The data in Fig. 3o,p were fit using simulations that included localization precision, a diffusion coefficient and variable time steps. The localization precisions were independently adjustable for the three distinct molecular species, to account for the fact that the precision of static and moving particles are expected to be different. The effect of localization precision is clearly identified in an $R^2/t$ histogram (Extended Data Fig. 5d), and this was the primary initial clue that suggested to us that the precision estimates obtained from centroid deviations were

too high. The Imp α–JF549 tracking data revealed that $R$ and $t$ were largely independent (Extended Data Fig. 5l), indicating that the jump histogram (Fig. 3o) largely reflected localization error rather than diffusive motion. This interpretation was borne out by the more complex model used to fit the data and reproduce the $R^2/t$ histogram (Fig. 3p).

## Reporting summary

Further information on research design is available in the Nature Portfolio Reporting Summary linked to this article.

## Data availability

The NPC scaffolds shown in Fig. 1a are from ref. 40, Fig. 2a,b are from refs. 45,47 and Fig. 4f are from ref. 48. All data described in the article are shown in the figures and provided in the Supplementary Information. Source data for the main figures and Extended Data figures are provided as Supplementary Information. Supplementary Data contains the coordinates for all 225 Imp α trajectories transiting an NPC. Owing to size, raw MINFLUX imaging data are not provided as part of the article, but are available from the authors on request. Source data are provided with this paper.

## Code availability

Data were curated and analysed with custom code written in MATLAB R2018z and MATLAB 2022b. This custom code was essential to the analysis and central for extracting conclusions from the data. All MATLAB scripts used for data analysis, along with default parameters and model data, are available in the GitHub repository (https://github.com/npctat2021/MINFLUX_NPC_Tracking). A user guide for the scripts is provided in the repository's 'README' section. The following commercial software packages were used for data analysis, data presentation and simulations: Imspector 16.3.15620-m2205-MINFLUX_BASE, ParaView 5.8.1, Image J (Fiji 1.52P), Origin 8.5, Kaleidagraph 5.01 and Microsoft Excel 16.76 (23081101). The Microsoft Excel spreadsheet used for simulating jump steps (Jump Step Histogram Simulator.xlsx) is provided in figshare[60].

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

**Acknowledgements** The plasmid pRSET–mEosEM was a gift from P. Xu (Addgene ID #132708). Plasmids pQE9-β1, pQE60-RanGAP and pQE60-RanBP1 were gifts from D. Görlich. The plasmid pGEX4T3–CAS was a gift from Y. Chook. This research was supported by a grant from the US National Institutes of Health National Institutes of General Medical Sciences (GM126190 to S.M.M.) and a Texas A&M Health Science Center Seedling Grant (to S.M.M.). The authors acknowledge the assistance of the Joint Microscopy Laboratory at the Texas A&M University College of Medicine and the National Institutes of Health for instrument funding support of the Luminosa confocal microscope (S10 OD032208). The authors also acknowledge the access and services provided by the Imaging Centre at the European Molecular Biology Laboratory, which was supported by the Boehringer Ingelheim Foundation.

**Author contributions** S.M.M., A. Sau and S.S. conceptualized the project and interpreted the experimental results, with contributions from other authors. A. Sau, S.S. and C.-M.G. conducted the MINFLUX data acquisitions. Z.H. and A. Sau developed and automated the MATLAB analysis scripts. A. Sau analysed the MINFLUX data with support from S.S. and C.-M.G. S.M.M. performed the error analysis and simulations. A. Sau and D.B. purified all proteins used and fluorescently tagged them as needed. A. Sau, D.B. and A. Sharma acquired and analysed the astigmatism microscopy data. S.D. generated the pET28a–Imp β1–mEosEM and pTrcHisA–NLS–BFP–mEosEM plasmids. The manuscript was drafted by A. Sau and S.M.M., with contributions from S.S. All authors contributed to the revision process and approved the final manuscript. S.M.M. supervised and administered the project, and garnered the funding to support the work.

**Competing interests** C.-M.G. is employed by Abberior Instruments, the developer and manufacturer of the MINFLUX system used in this study. The other authors declare no competing interests.

## Additional information
**Correspondence and requests for materials** should be addressed to Siegfried M. Musser.

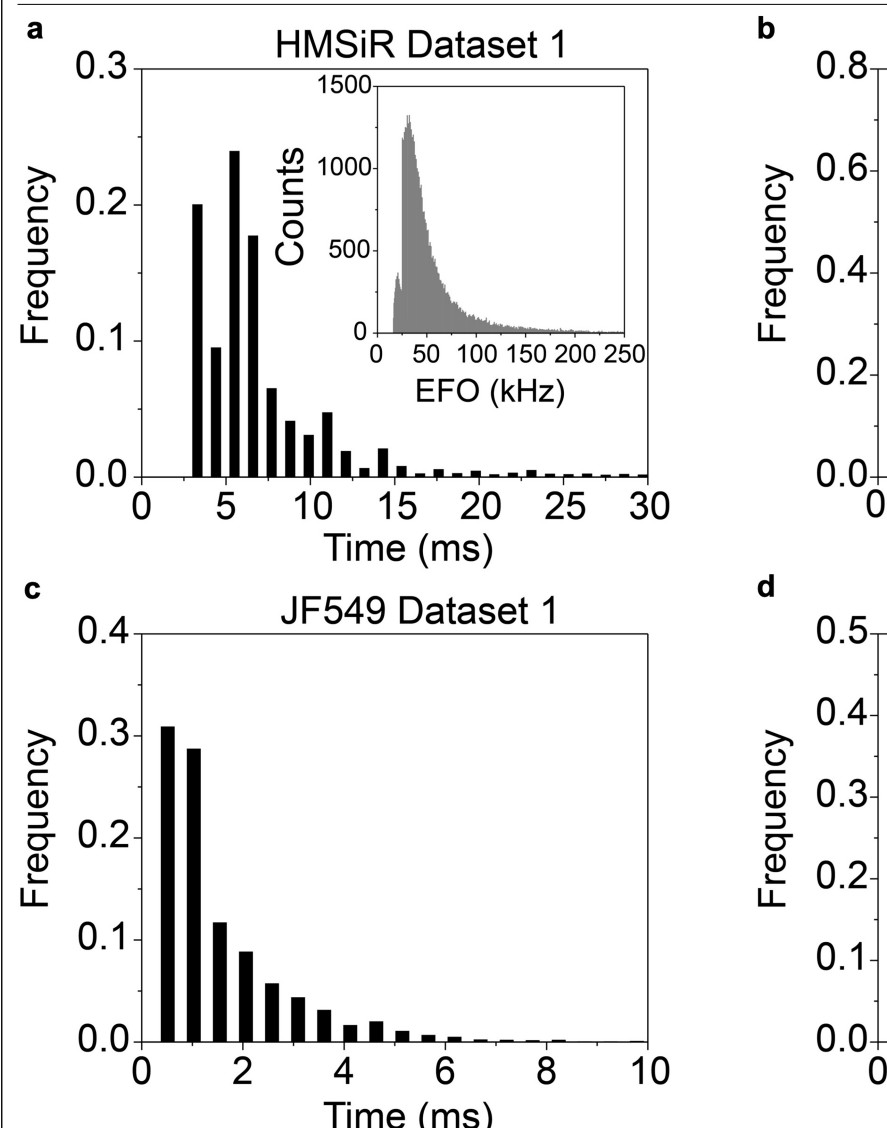

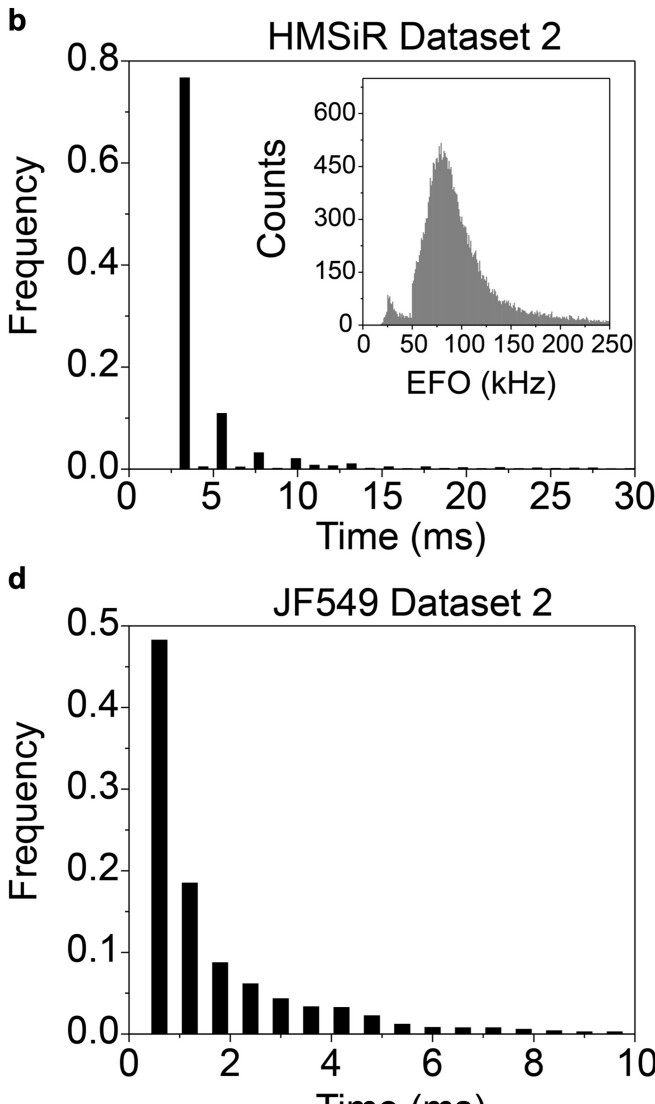

**Extended Data Fig. 1 | Time step histograms. a,b,** The time between two successive MINFLUX localizations for Nb$^{GFP}$-HMSiR in datasets 1 and 2. The insets are EFO (emission frequency at offset) histograms. The EFO is a measure of the emission intensity. With a higher average EFO, the localization is less likely to fail since sufficient photons are collected within the dwell time of the localization. Longer localization times occur when the dwell time needs to be extended to collect enough photons. These EFO histograms reveal a significantly higher EFO peak within dataset 2 compared to dataset 1. This difference explains why the average localization time of dataset 1 was longer than dataset 2. We note that the different distributions of HMSiR localization times for the two datasets should have no material effect on the NPC reconstructions since the data were collected over 15–20 min and the NPC scaffolds have been shown to be stable over this time period[7]. Further explanation and discussion of the EFO can be found in the Methods and Extended Data Fig. 6. **c,d,** The time between two successive MINFLUX localizations for Imp α-JF549 in datasets 1 and 2. See Supplementary Tables 1 and 3 for acquisition parameters.

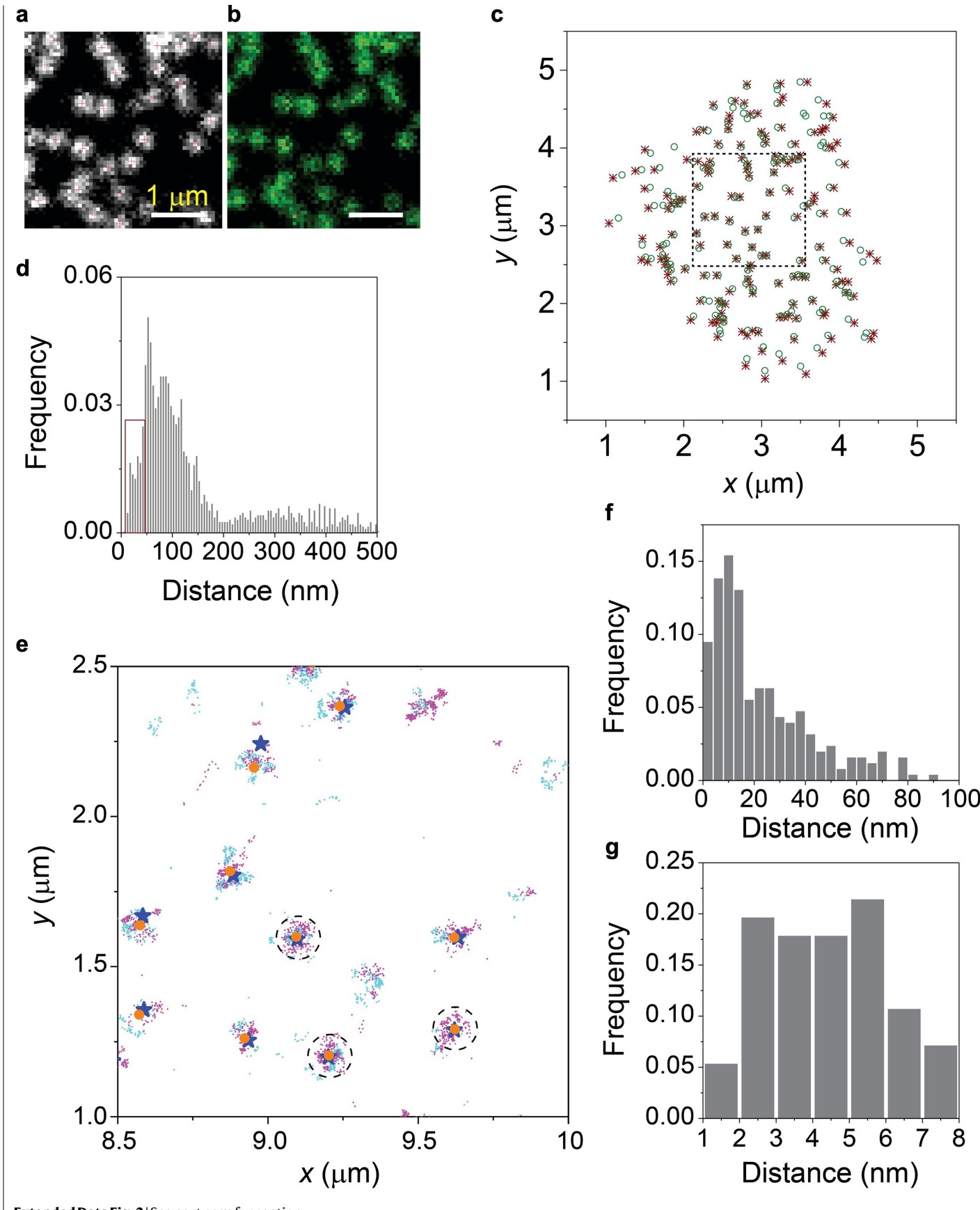

**Extended Data Fig. 2** | See next page for caption.

**Extended Data Fig. 2 | NPC positional stability. a,b**, Confocal images of the EGFP emission from NPCs in a permeabilized U2OS NUP96-EGFP cell (EX = 488 nm) before (**a**) and after (**b**) MINFLUX localizations of Nb$^{GFP}$-HMSiR and Imp α-JF549. Typical results from $N$ = 18 cells. **c**, NPC positions before and after MINFLUX imaging. Confocal spots were fit with a 2D Gaussian distribution to determine the approximate center of the NPC before (*red x's*) and after (*green circles*) the MINFLUX data acquisition. This process was used to identify scaffolds that were sufficiently stable during image acquisition to justify further analysis of the HMSiR localizations. Only well-formed scaffolds identified by HMSiR localizations (see Methods) were analyzed further. The boxed area approximates the image areas of **a** and **b**. **d**, NPC movements during imaging. The distance moved by confocal spot centroids during MINFLUX imaging indicates that ~10% of NPCs (data from 11 cells) were sufficiently stable (shift <40 nm; considered within the error range of the precision for the conditions) for potential analysis. These stable NPCs were invariably near the center of the bottom of the nuclear envelope. **e-g**, Drift estimated from MINFLUX measurements. To assess NPC scaffold drift using higher precision data, the estimated centers of clusters acquired during the 1$^{st}$ and 2$^{nd}$ half of a MINFLUX HMSiR acquisition were compared. Data were fit using a double-circle algorithm (Extended Data Fig. 3c). **e**, Scatter plot of HMSiR localizations (*cyan* for 1$^{st}$ half; *magenta* for 2$^{nd}$ half) overlapped with the fitted centers (*orange circle* for 1$^{st}$ half; *blue star* for 2$^{nd}$ half). *Black circles* identify sufficiently stable scaffolds [deviations ≤ 8 nm; see **g**]. **f**, Distances between early (1$^{st}$ half) and late (2$^{nd}$ half) centroids for all clusters ($N$ = 254) identified in 4 nuclei. **g**, Early/late centroid distances for those clusters identified in **e** that were selected for Imp α-JF549 track analysis. Note that the centroid positional accuracy for the two estimates using split datasets is reduced from the single estimate from complete datasets for each pore.

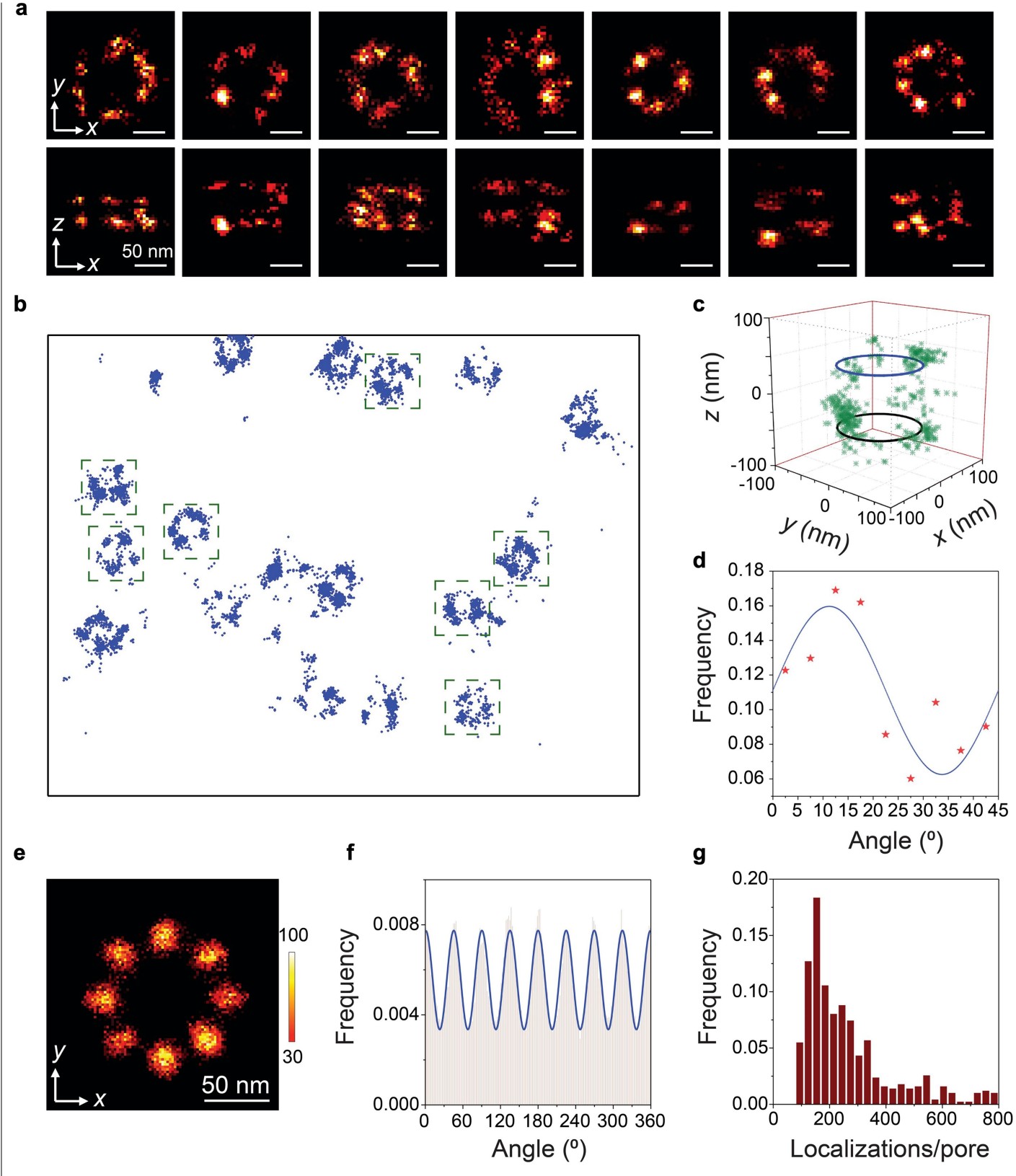

**Extended Data Fig. 3** | See next page for caption.

**Extended Data Fig. 3 | Alignment of NPC scaffolds. a**, MINFLUX images of single NPCs. Images are 2D histograms of localizations for individual NPCs from permeabilized U2OS cells containing NUP96-mEGFP labeled with Nb[GFP]-HMSiR. **b**, Scatter plot of HMSiR localizations. Only high-density circular clusters were analyzed further (*squares*); deformed or incomplete clusters were rejected. **c**, Double-circle fitting. High quality localization clusters were fit to a double-circle model[7], reflecting the double-ring structure of the NPC. **d**, Rotation phase angle. The angles in the *xy* plane of the individual localizations in **c** relative to the centroid of the double-circle fit were binned (0–45°; assumes an eightfold periodicity), normalized, and fit to $y = 1/9 + (1/20.6)*\sin(8(x\text{-}\phi))$, as described previously[7]. The 1/9 term reflects the average frequency expected for the 9 bins (5° each), and the sine scaling factor is a reasonable average based on simulations. Note that improving the chi-square of the fit by allowing for an adjustable scaling factor does not change the estimated phase angle due to the orthogonality of frequency and angle. **e**, 2D histogram of aligned NPC scaffolds. Localization clusters were rotated by a phase angle, as determined in **d**, and aligned based on their centroids, as determined in **c** (37 cells, 541 NPCs, $N = 82,331$ localizations). This is the same image as in Fig. 1g. **f**, Angular distribution of rotationally corrected localizations. The angle distribution for the individual localizations in **e** was fit to $y = 1/180 + c\sin(8(x\text{-}\phi))$, where $c$ and $\phi$ are fit parameters, and the 1/180 term reflects the average frequency expected for the 180 bins (2° each). **g**, The number of HMSiR localizations obtained per NPC scaffold. As few as ten localizations were used earlier to identify an NPC scaffold[7], but here more than 10-fold more localizations were obtained, on average.

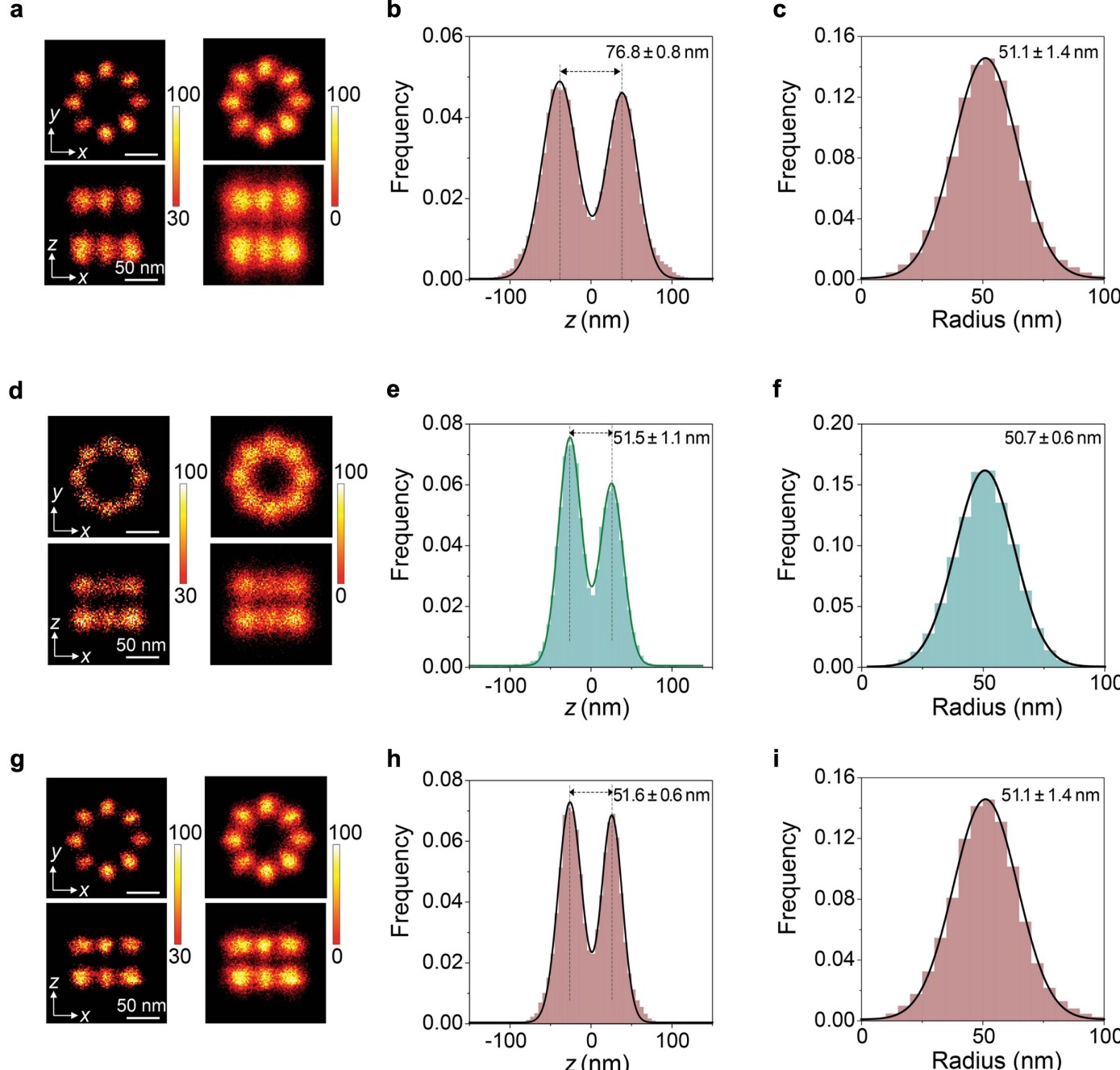

**Extended Data Fig. 4 | Determining the *z*-scaling factor for MINFLUX measurements. a**, Averaged structure for NPCs from uncorrected MINFLUX measurements (37 cells, 541 NPCs, 82,331 HMSiR localizations). **b,c**, The *z* (**b**) and radial (**c**) distributions for the data in **a**. The distance between the two peaks in **b** yields the ring separation (76.8 ± 0.8 nm), and the peak in **c** is considered the radius of the pore (51.1 ± 1.4 nm). The data/results for **a-c** are summarized in Supplementary Table 2 (row 8). **d**, Averaged structure for NPCs from astigmatism measurements of the pore scaffolds used for the alignment of mEosEM data in Fig. 4 and Extended Data Fig. 9 (129 cells, 1453 NPCs, 17,234 HMSiR localizations).

**e,f**, The *z* (**e**) and radial (**f**) distributions for the data in **d**. The data/results for **d-f** are summarized in Supplementary Table 2 (rows 2–5). **g**, Averaged structure for NPCs from corrected MINFLUX measurements. The data in **a** were corrected by multiplying all *z* values by 0.67. This factor was determined as indicated in Supplementary Table 2 (note 'e'). The images on the left are shown in Fig. 1g,h. **h,i**, The *z* (**h**) and radial (**i**) distributions for the data in **g**. Note that the radial distribution for the corrected MINFLUX data (**i**) is identical to **c**. The data/results for **g-i** are summarized in Supplementary Table 2 (row 9).

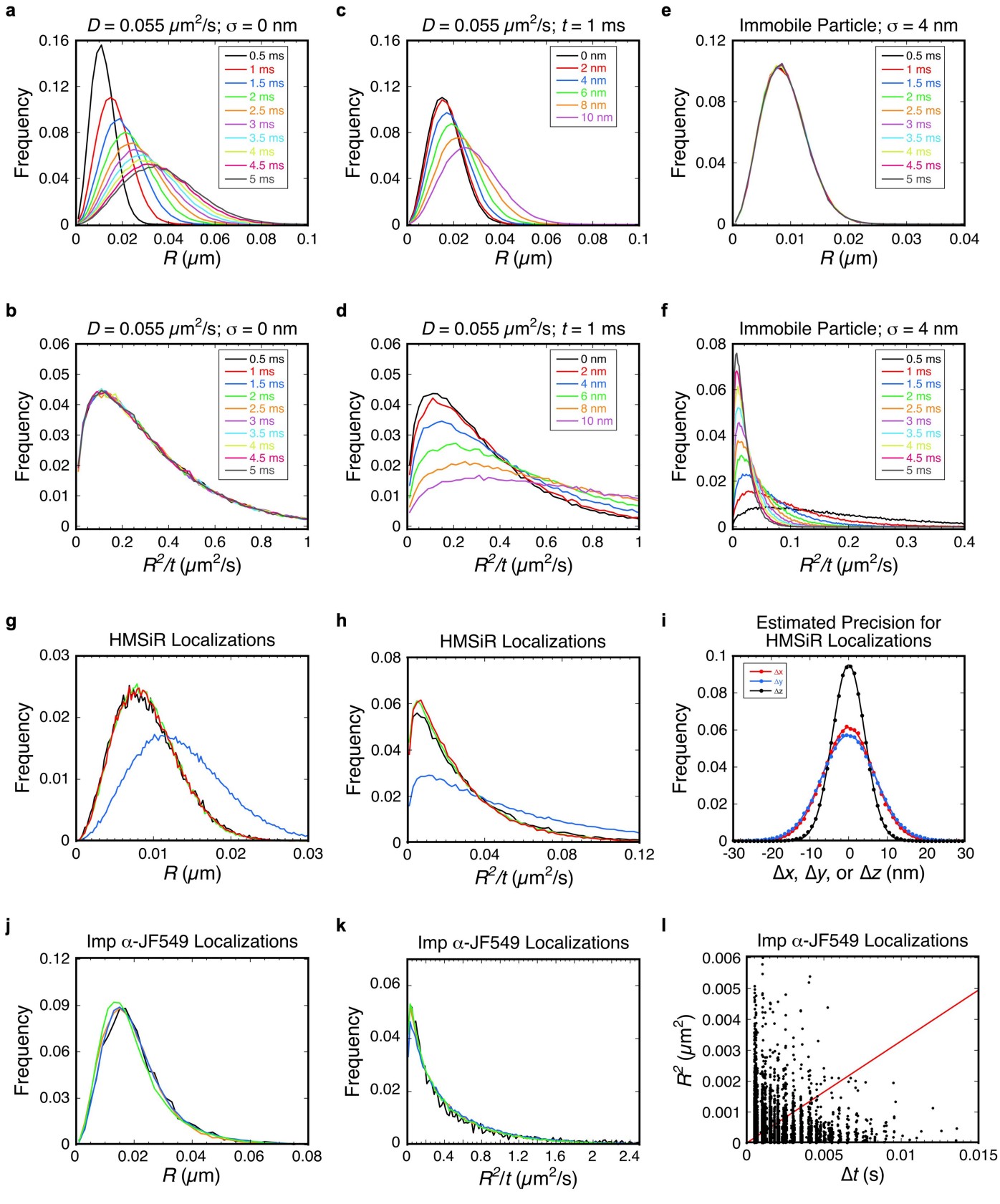

**Extended Data Fig. 5** | See next page for caption.

**Extended Data Fig. 5 | Jump and $R^2/t$ histograms under various conditions.**
**a-f**, Simulated jump and $R^2/t$ histograms illustrating the effects of particle
movement, time step duration, and precision. Conditions are indicated in the
various panels. For **a-d**, isotropic diffusion ($D_x = D_y = D_z$) was assumed; the value
$D = 0.055\,\mu m^2/s$ corresponds to the value obtained for species 3 of the best-fit
for the localizations of the JF549 dye on Imp α (see Fig. 3o,p). The range of $t$ values
in **a** and **b** approximate the first ten values from the Imp α-JF549 time step
histograms (Extended Data Fig. 1c,d). The precision (σ) values used in **c-d**
correspond to the range used to simulate fits to experimental data. **e-f** show
the effect of $t$ for an immobilized particle. Jump step histograms are unaffected
by $t$ (**e**) since the measured jump step is determined entirely by the precision
of the two localizations needed; in contrast, larger $t$ values promote a sharp
peak near zero in the $R^2/t$ histogram (**f**), which is the key observation that
promoted the inclusion of species 1 for the fit in Fig. 3o,p. **g-i**, Analysis of HMSiR
localizations. **g**, The *blue* and *red* simulated fits and the *black* experimental
data curve are identical datasets and parameters for those of the same color in
Fig. 1j. The *green* curve corresponds to simulations where $\sigma_x = 4.45\,nm = 0.93\sigma_y =$
$1.55\sigma_z$ with no diffusional drift; thus, the centroid (molecular position) is
constant throughout the trajectory and jump steps in $x$, $y$, and $z$ are distributed
according to the localization precision values, which do not agree with the
localization precisions as estimated from experimental centroid deviations
(Fig. 1i). So, the *green* fit cannot be considered consistent with the data despite
the agreement with the experimental jump step and $R^2/t$ [see **h**] histograms.
The *green* and *red* curves substantially overlap and are therefore difficult to
distinguish. **h**, $R^2/t$ histograms corresponding to the curves in **g**. **i**, To determine
the effect of diffusional motion on the estimated precision, centroid deviations
were calculated from the data for the *red* curve in **g** (same curve as in Fig. 1j)

yielding estimated precision values of $\sigma_x = 6.5\,nm$, $\sigma_y = 7.0\,nm$, and $\sigma_z = 4.2\,nm$,
which match those from the experimental centroid-based precision calculation
(Fig. 1i). Note that the $D_x$, $D_y$, and $D_z$ values were adjusted to reproduce the
experimental centroid deviations while keeping $\sigma_x = 0.93\sigma_y = 1.55\sigma_z$. Due to
particle movement during the acquisition of repeated localizations, the
precision estimated from centroid deviations therefore overestimates the
localization precision (simulation input: $\sigma_x = 4.1\,nm$, $\sigma_y = 4.4\,nm$, and $\sigma_z = 2.6$).
**j-l**, Analysis of Imp α-JF549 localizations. **j,k**, The *black* curves (experimental
data) are identical to those in Fig. 3o,p. The *orange* curves have parameters
identical to the *red* curves in Fig. 3o,p, except that particle rotation around a
centroid with $r = 6\,nm$ was included for species 2, and the precision for this
species was reduced to $\sigma_x = 7.2\,nm = 0.93\sigma_y = 1.55\sigma_z$. This approximates slow
rotation within a confined region for a transport complex with the dye on the
surface. A two species model did not allow for simultaneous good fits to both
the jump step and the $R^2/t$ histograms. Two examples are provided. The *blue*
curves correspond to simulations where $\sigma_x = 7.8\,nm = 0.93\sigma_y = 1.55\sigma_z$ with
60% stuck particles and 40% of particles have $D = 0.07\,\mu m^2/s$. The *green* curves
correspond to simulations with 58% stuck particles ($\sigma_x = 7.1\,nm = 0.93\sigma_y = 1.55\sigma_z$)
and 42% of particles have $D = 0.065\,\mu m^2/s$ ($\sigma_x = 7.8\,nm = 0.93\sigma_y = 1.55\sigma_z$). Note
that for the jump step histograms the *orange* and *blue* curves largely overlap,
and for the $R^2/t$ histograms the *orange* and *green* curves almost exactly overlap.
**l**, A plot of $R^2$ vs $\Delta t$ for Imp α-JF549 localizations supports the interpretation
that the transiting particles are 'stuck' most of the time since the displacements
do not follow an expected $<R^2> = 6Dt$ profile for a diffusing particle. The red line
is the expected slope ($6D$) for $D = 0.055\,\mu m^2/s$. For all simulations, the number
of jump distances was $N = 96{,}000$ with 25 localizations per trajectory.

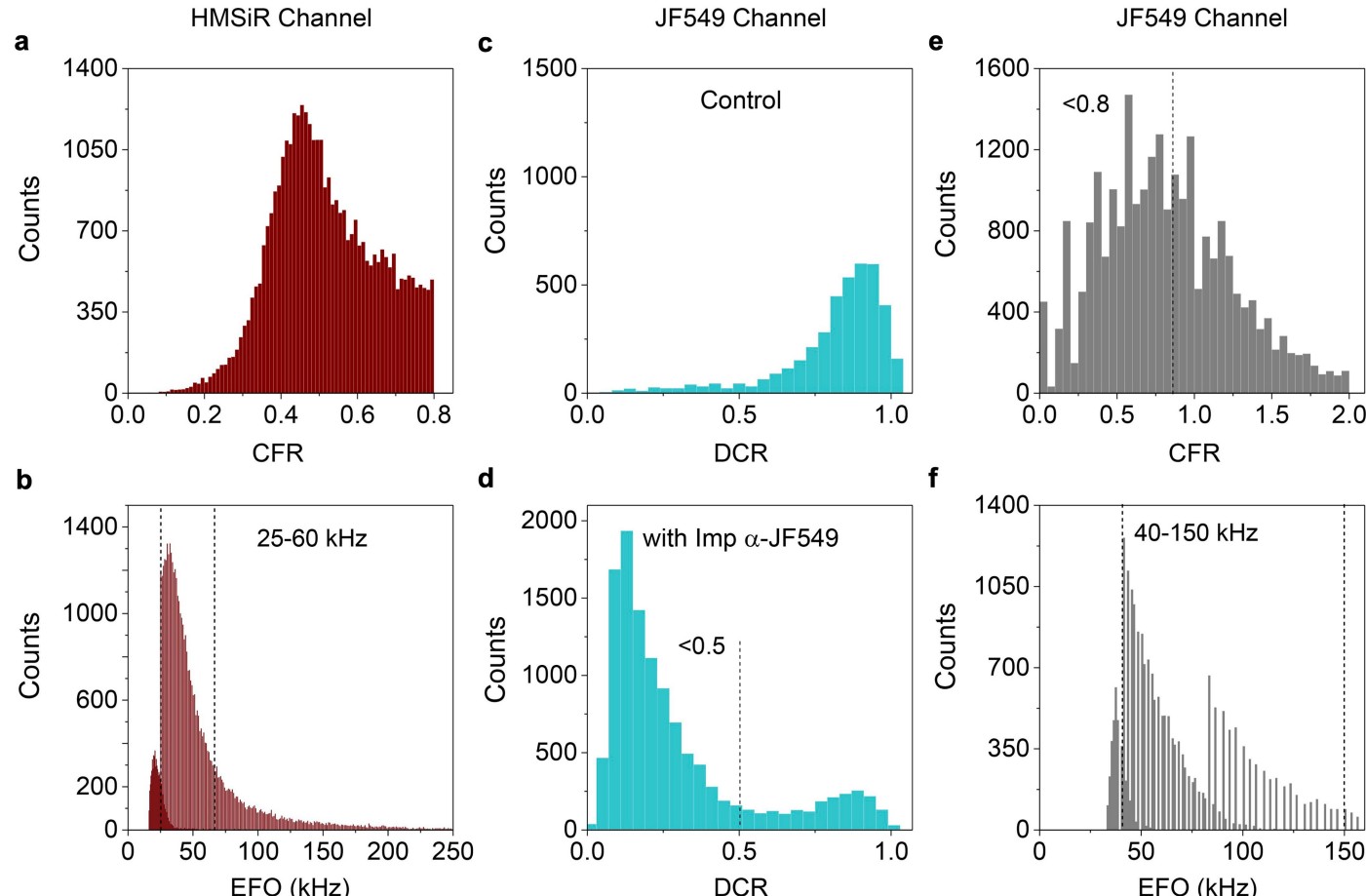

**Extended Data Fig. 6 | MINFLUX filtration criteria used for Nb$^{GFP}$-HMSiR localization and Imp α-JF549 tracking.** A description of the MINFLUX parameters reported in this figure and what they mean are described in the Methods. **a**, The center frequency ratio (CFR) values obtained for the HMSiR channel (EX = 642 nm). The CFR is a measure of localization quality. The primary reason for a high CFR in these measurements is the contribution from a second fluorophore, which leads to inaccurate localization values. Thus, an upper bound cutoff of 0.8 was used for HMSiR localizations during acquisition (see Methods). **b**, The effective frequency at offset (EFO) for the HMSiR channel. Background, i.e., low-level emission from the permeabilized cells, was partially eliminated during acquisition of HMSiR dataset 1 by using an EFO > 25 kHz so as not to spend acquisition time on weak signals. A lower threshold of 15 kHz was used during the iteration sequence (Supplementary Table 1), which still captures background (peak at ~20 kHz), but the 25 kHz threshold during pattern repeats (25 photons/ms) eliminated most of these weak signals. Post-acquisition, a trajectory length of ≥ 5 localizations reduced background contributions further. Also post-acquisition, an upper threshold of 60 kHz was used to eliminate localizations contaminated by on-switching of a second fluorophore. Due to the higher EFO values for HMSiR dataset 2 (see Extended Data Fig. 1b, *inset*), an EFO range of 50–100 kHz was used for this dataset. **c,d**, Detector channel ratio (DCR) for 561 nm excitation in the absence (**c**) and presence (**d**) of Imp α-JF549. Some background fluorescence in the JF549 channel was detectable within permeabilized U2OS NUP96-mEGFP cells that had been treated with Nb$^{GFP}$-HMSiR but without addition of 'transport mix' (see Methods). This background signal

(**c**) was identified and filtered out based on its DCR, which was generally higher than that of JF549 fluorescence. DCR is defined as the emission frequency in detector 1 divided by the emission frequencies measured for detector 1 + detector 2. Here, detector 1 = 650–685 nm and detector 2 = 580–630 nm. For cells without the transport mix (**c**), the DCR was mostly > 0.5 (40 min acquisition), whereas in the presence of the transport mix (**d**), most DCR values were <0.5, with the values above 0.5 likely reflecting the background. Thus, the data filtration criterion for tracking Imp α-JF549 was DCR < 0.5. **e**, CFR for the JF549 channel (EX = 561 nm). The data filtration criterion for JF549 was CFR < 0.8 and implemented post-acquisition. Unlike for HMSiR localizations where the CFR check during imaging was set to a low value to select for high quality localizations at the time of acquisition, for tracking Imp α-JF549 the CFR ratio was set to a large cut-off (> 2.0) to avoid rejecting tracks that were temporarily interrupted. **f**, EFO for the JF549 channel (EX = 561 nm). This EFO distribution has peaks at ~37, 42, and 83 kHz. The distribution underlying the first peak represents background noise, and the latter two peaks result from the photon emission streams generated by one or two JF549 dyes. Imp α has four reactive cysteine residues and, while the protein was under-labeled with dye, some dual labeling could not be avoided. Since trafficking behavior was not expected to be influenced by the number of dyes on Imp α, the data filtration criterion was an EFO of 40–150 kHz. Of the 225 tracks used in the analysis, 19 had an EFO > 80 kHz (two JF549 dyes). Note that the clear third peak observed here is not present in **b**, indicating that simultaneous detection of more than one HMSiR dye was infrequent.

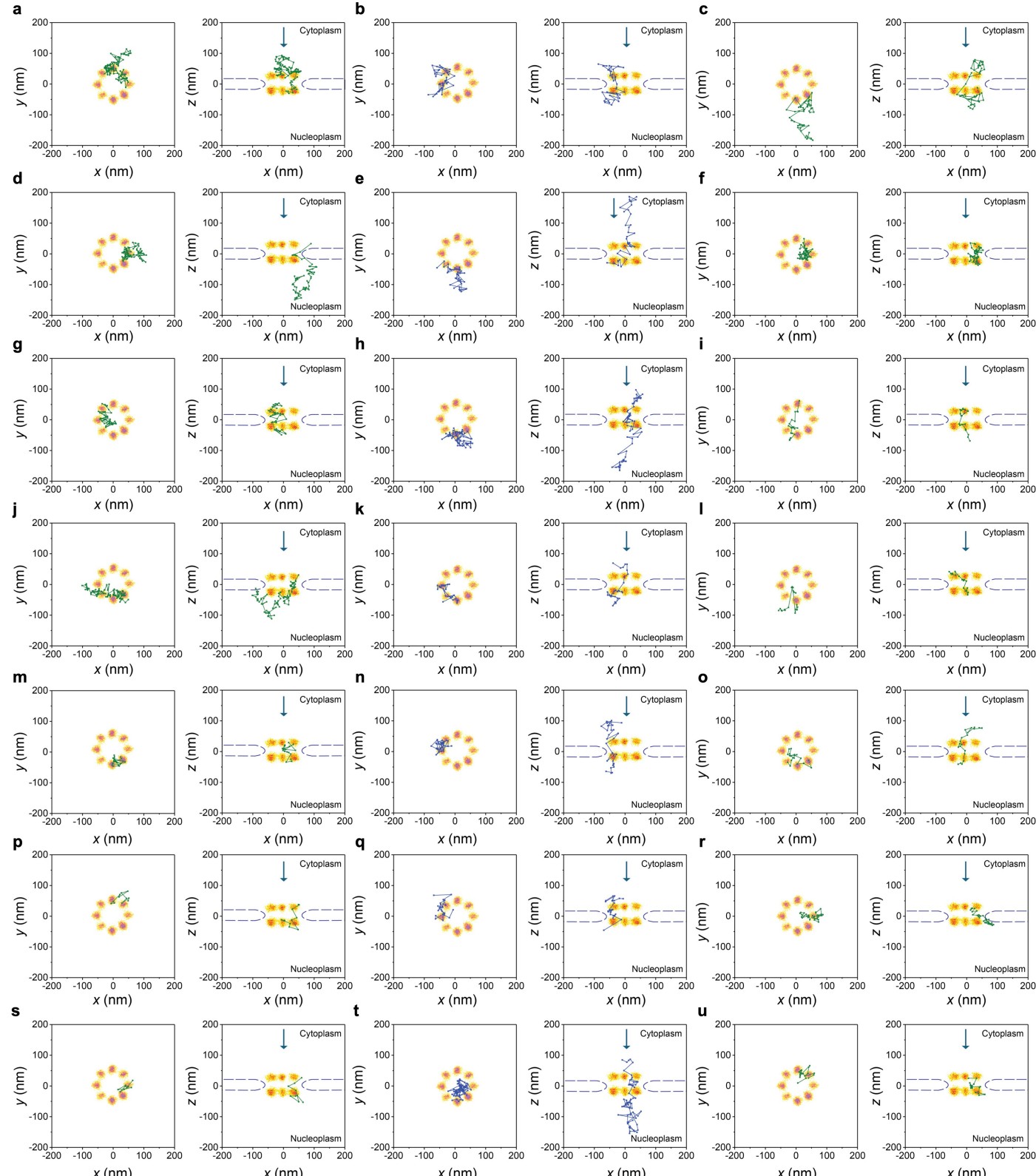

**Extended Data Fig. 7 | Individual import trajectories of Imp α-JF549.**
**a–u**, For each pair of images, the left panel shows the *xy* plane projection and the right panel shows the *xz* axial plane projection. In the right panels, the two lines around ±20 nm joined by a central curve represent the approximate position of the nuclear envelope. The blue arrow indicates the side of the midplane (*z* = 0) on which the trajectory starts. NPC localizations are shown

as a 2D histogram in fire color. The temporal length of Imp α-JF549 trajectories varied substantially, depending on when the fluorophore was first identified through the field scanning and the iterative localization sequence, and when it photobleached or became otherwise untrackable. Short and lengthy residence times near and within the NPC scaffold are suggestive of pauses or restrictive barriers.

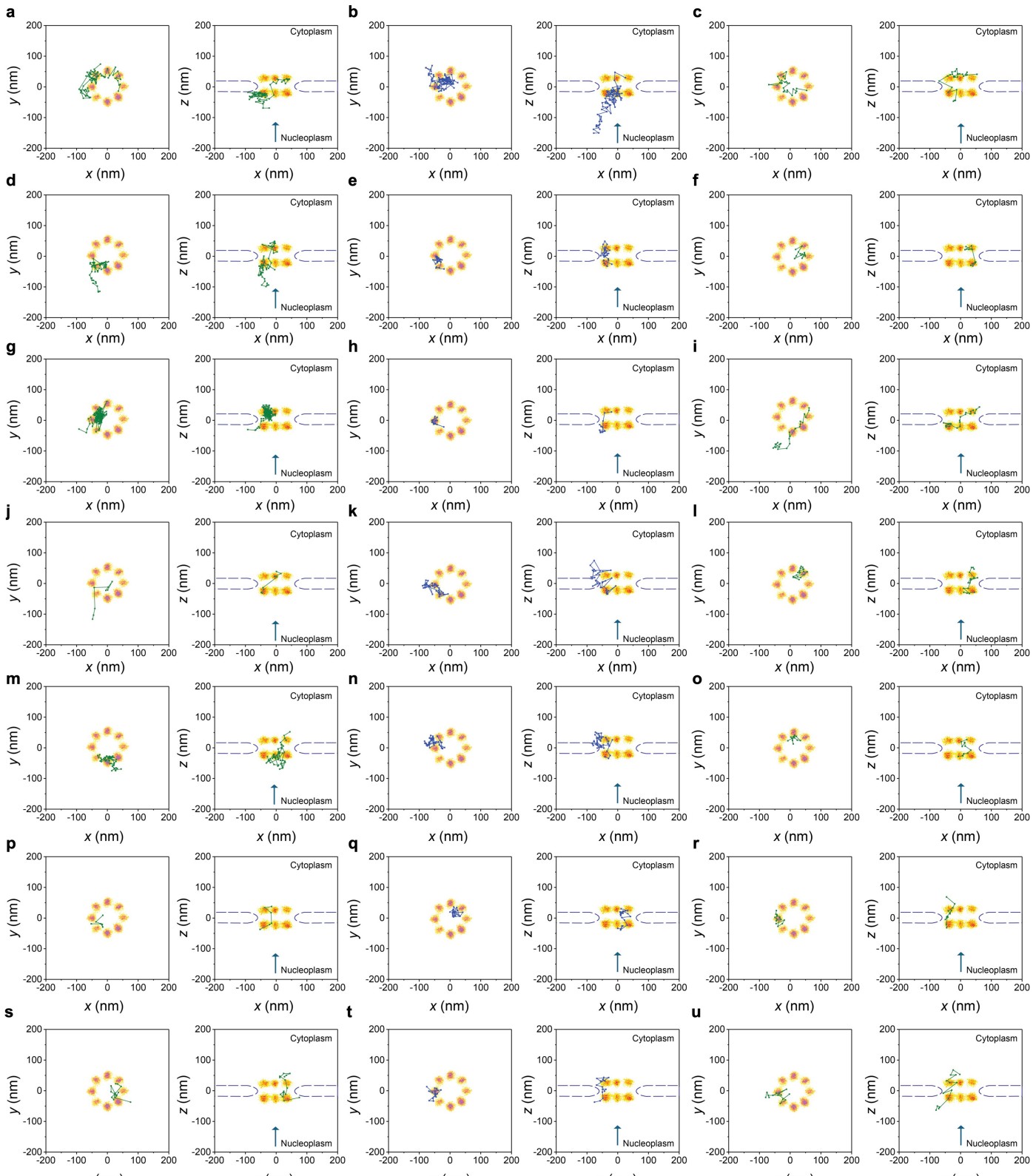

**Extended Data Fig. 8 | Individual export trajectories of Imp α-JF549.**
**a–u**, For each pair of images, the left panel shows the *xy* plane projection and the right panel shows the *xz* axial plane projection. In the right panels, the two lines around ±20 nm joined by a central curve represent the approximate position of the nuclear envelope. The blue arrow indicates the side of the midplane (*z* = 0) on which the trajectory starts. NPC localizations are shown as a 2D histogram in fire color. The temporal length of Imp α-JF549 trajectories varied substantially, depending on when the fluorophore was first identified through the field scanning and the iterative localization sequence, and when it photobleached or became otherwise untrackable. Short and lengthy residence times near and within the NPC scaffold are suggestive of pauses or restrictive barriers.

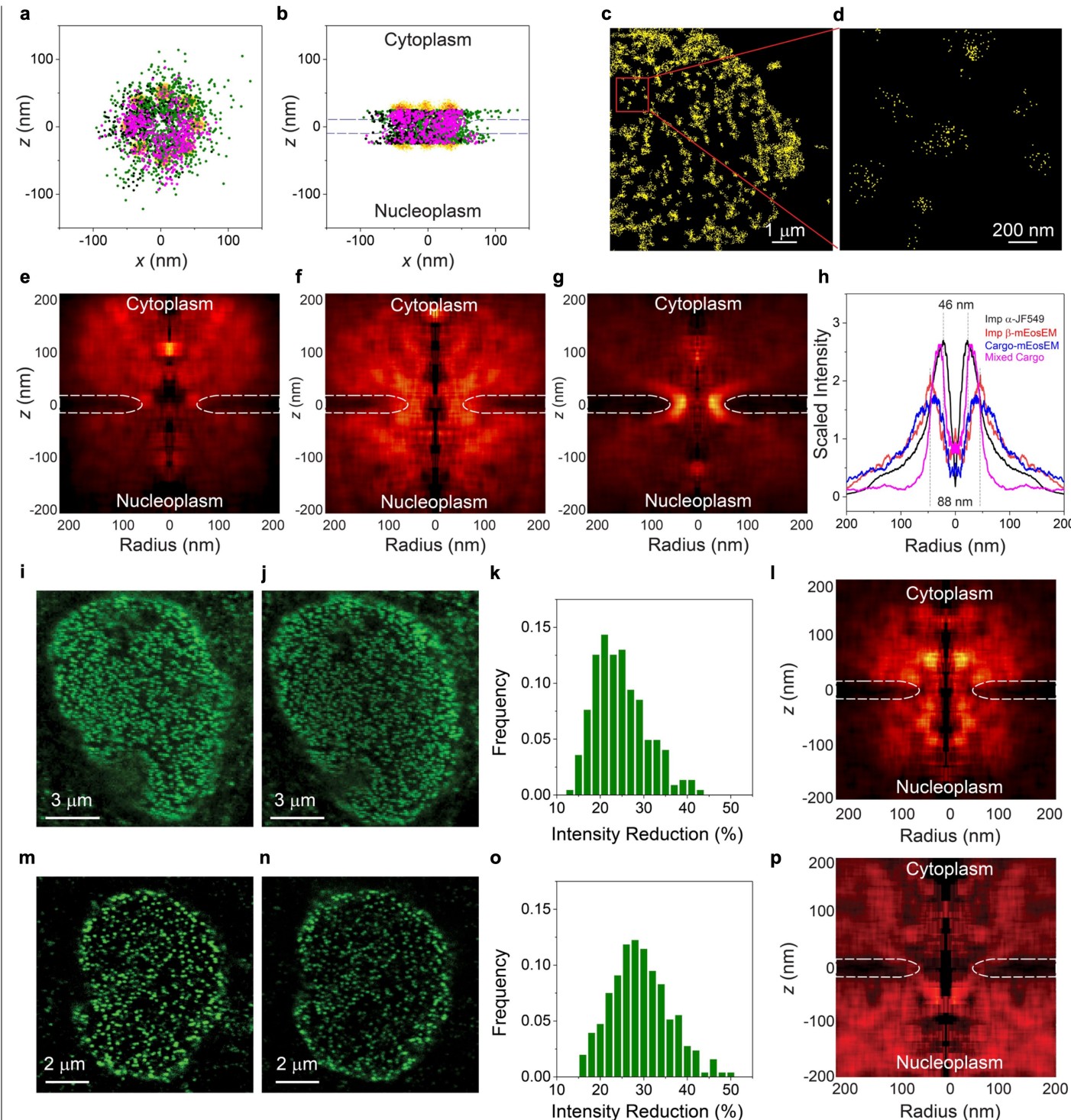

**Extended Data Fig. 9** | See next page for caption.

 a,b**, Peripheral translocation of Imp α in MINFLUX experiments. All localizations from 225 MINFLUX tracks with $|z| \leq 25$ nm ($N = 2,015$) are shown in $xy$ and $xz$ representations. Points from trajectories are identified as import (*purple*), export (*black*), and abortive and undecided (*green*). **c,d**, Scatterplots of Imp β1-mEosEM localizations at the bottom of a permeabilized U2OS cell. Under constant low-level 408 nm illumination, mEosEM was stochastically photoconverted to the 'red' form (Supplementary Video 5) and then localized by 3D astigmatism imaging (EX = 561 nm). The nuclear envelope-bound Imp β1-mEosEM was localized to distinct clusters, indicating that Imp β1 was mostly distributed within or near an NPC scaffold. **e,f**, Volume corrected radial density maps of Imp β1-mEosEM (**e**) and Imp β1/Imp α/NLS-BFP-mEosEM complexes (**f**) showing peripheral binding within the pore. These are the full scale representations of the images shown in Fig. 4c,d. **g**, Volume corrected radial density map of Imp β1/Imp α/NLS-2xBFP (dyes on Imp α) and Imp β2/M9-βGal (dyes on βGal) complexes undergoing nuclear import. Data are from Chowdhury et al.[7] ($N = 4,665$ localizations, 20 cells, 257 NPCs). **h**, Localization densities for bound Imp β1 and transport complexes (Imp β1/Imp α/NLS-BFP-mEosEM), and actively transiting Imp α and βGal. Average localization densities were calculated for $|z| \leq 15$ nm. Reproduction of Fig. 4f with the data from **g** added.

**i-l**, RanGTP wash of Imp β1-mEosEM decorated NPCs. **i**, Confocal image of a permeabilized U2OS NUP96-mEGFP cell nucleus after incubation with 0.5 μM Imp β1-mEosEM (EX = 478 nm). **j**, Confocal image of the cell nucleus in **i** after 'Ran mix' wash (see Methods). **k**, Quantification of the mEosEM fluorescence loss between **i** and **j**, corrected for -1.6% photobleaching ($N = 10$ cells, 223 NPCs). **l**, Volume corrected radial density map of Imp β1-mEosEM after 'Ran mix' wash ($N = 1,875$ localizations, 24 cells, 343 NPCs). **m-p**, 'Transport mix – high α' wash of Imp β1/Imp α/NLS-BFP-mEosEM decorated NPCs. **m**, Confocal image of a permeabilized U2OS NUP96-mEGFP cell nucleus after incubation with Imp β1/Imp α/NLS-BFP-mEosEM complexes (EX = 478 nm). **n**, Confocal image of the cell nucleus in **m** after 'transport mix – high α' wash (see Methods). Note that this 'transport mix– high α' wash is the same composition of proteins used in the simultaneous import/export MINFLUX experiments ('transport mix'), except with a higher Imp α concentration (unlabeled). **o**, Quantification of the mEosEM fluorescence loss between **m** and **n**, corrected for -1.6% photobleaching ($N = 10$ cells, 223 NPCs). **p**, Volume corrected radial density map of NLS-BFP-mEosEM after 'transport mix – high α' wash ($N = 2,731$ localizations, 18 cells, 637 NPCs). For **b, e, f, g, l**, and **p**, the two lines around ±20 nm joined by a central curve represent the approximate position of the nuclear envelope.

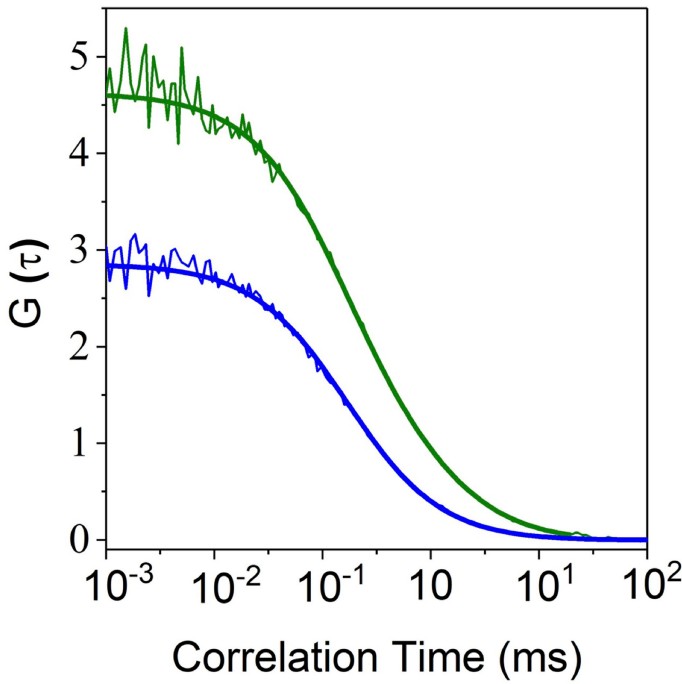

**Extended Data Fig. 10 | Diffusion coefficients from FCS experiments.**
FCS curves of Imp β1/Imp α-JF549/NLS-2xBFP import complexes in IB (*blue*) and
IB-PVP (*green*) yielded diffusion coefficients of 64.2 ± 0.4 and 14.1 ± 0.1 $\mu m^2$/s,
respectively (EX = 560 nm; pulsed mode). Compared with the rapid diffusion
observed in import buffer (IB), the slow diffusive movements within the
NPC are ~1000 times slower (0.055 $\mu m^2$/s; Fig. 3o,p). [Imp β1] = 0.5 μM,
[NLS-2xBFP] = 0.5 μM, and [Imp α-JF549] = 5 nM.

# Reporting Summary

## Statistics

For all statistical analyses, confirm that the following items are present in the figure legend, table legend, main text, or Methods section.

| n/a | Confirmed | |
|---|---|---|
| ☐ | ☒ | The exact sample size (*n*) for each experimental group/condition, given as a discrete number and unit of measurement |
| ☐ | ☒ | A statement on whether measurements were taken from distinct samples or whether the same sample was measured repeatedly |
| ☒ | ☐ | The statistical test(s) used AND whether they are one- or two-sided<br>*Only common tests should be described solely by name; describe more complex techniques in the Methods section.* |
| ☒ | ☐ | A description of all covariates tested |
| ☐ | ☒ | A description of any assumptions or corrections, such as tests of normality and adjustment for multiple comparisons |
| ☐ | ☒ | A full description of the statistical parameters including central tendency (e.g. means) or other basic estimates (e.g. regression coefficient) AND variation (e.g. standard deviation) or associated estimates of uncertainty (e.g. confidence intervals) |
| ☒ | ☐ | For null hypothesis testing, the test statistic (e.g. *F*, *t*, *r*) with confidence intervals, effect sizes, degrees of freedom and *P* value noted<br>*Give P values as exact values whenever suitable.* |
| ☒ | ☐ | For Bayesian analysis, information on the choice of priors and Markov chain Monte Carlo settings |
| ☒ | ☐ | For hierarchical and complex designs, identification of the appropriate level for tests and full reporting of outcomes |
| ☒ | ☐ | Estimates of effect sizes (e.g. Cohen's *d*, Pearson's *r*), indicating how they were calculated |

*Our web collection on statistics for biologists contains articles on many of the points above.*

## Software and code

Policy information about availability of computer code

| Data collection | Data collection was performed on microscopes as detailed in the manuscript using:<br>CasAO 1.0, MiCAO 1.3, LabVIEW 2015, Micro-Manger 2.0 (Zeiss 200M; astigmatism microscope for 3D localization of single molecules)<br>Abberior Imspector 16.3.13924-m2112 (MINFLUX microscope)<br>Luminosa 1.0.0.4067 (Luminosa microscope) |
|---|---|
| Data analysis | Data were curated and analyzed with custom code written in MATLAB R2018z and MATLAB 2022b. This custom code was essential to the analysis and central for extracting conclusions from the data. All MATLAB scripts used for data analysis, along with default parameters and model data, are available in the GitHub repository (https://github.com/npctat2021/MINFLUX_NPC_Tracking). A user guide for the scripts is provided in the repository's 'README' section.<br>The following commercial software packages were used for data analysis, data presentation, and simulations:<br>Imspector 16.3.15620-m2205-MINFLUX_BASE<br>ParaView 5.8.1<br>Image J (Fiji 1.52P)<br>Origin 8.5<br>Kaleidagraph 5.01<br>Microsoft Excel 16.76 (23081101)<br>The Microsoft Excel Spreadsheet used for simulating jump steps (Jump Step Histogram Simulator.xlsx) is provided in the Supplementary Information. |

For manuscripts utilizing custom algorithms or software that are central to the research but not yet described in published literature, software must be made available to editors and reviewers. We strongly encourage code deposition in a community repository (e.g. GitHub). See the Nature Portfolio guidelines for submitting code & software for further information.

## Data

Policy information about availability of data

All manuscripts must include a data availability statement. This statement should provide the following information, where applicable:

- Accession codes, unique identifiers, or web links for publicly available datasets
- A description of any restrictions on data availability
- For clinical datasets or third party data, please ensure that the statement adheres to our policy

The NPC scaffolds shown in Fig. 1a, Fig. 2a, Fig. 2b, and Fig. 4f are from https://www.nature.com/articles/nsmb.3244, https://doi.org/10.1126/science.add2210, https://pdb101.rcsb.org/motm/205, and https://www.sciencedirect.com/science/article/pii/S0092867421000684, respectively. All data described in the manuscript are shown in the figures and provided in Supplementary Information. Source data for the main figures and extended data figures are provided as Supplementary Information. SI All Transport contains the coordinates for all 225 Imp alpha trajectories transiting an NPC. Due to size, raw MINFLUX imaging data is not provided as part of the manuscript, but is available from the authors upon request.

## Research involving human participants, their data, or biological material

Policy information about studies with human participants or human data. See also policy information about sex, gender (identity/presentation), and sexual orientation and race, ethnicity and racism.

| | |
|---|---|
| Reporting on sex and gender | N/A |
| Reporting on race, ethnicity, or other socially relevant groupings | N/A |
| Population characteristics | N/A |
| Recruitment | N/A |
| Ethics oversight | N/A |

Note that full information on the approval of the study protocol must also be provided in the manuscript.

# Field-specific reporting

Please select the one below that is the best fit for your research. If you are not sure, read the appropriate sections before making your selection.

☒ Life sciences  ☐ Behavioural & social sciences  ☐ Ecological, evolutionary & environmental sciences

For a reference copy of the document with all sections, see nature.com/documents/nr-reporting-summary-flat.pdf

# Life sciences study design

All studies must disclose on these points even when the disclosure is negative.

| | |
|---|---|
| Sample size | No statistical methods were used to predetermine sample size. Single molecule localizations numbered in the thousands to tens of thousands and were collected for hundreds of nuclear pores, tens of cells and multiple days. The nuclear pore scaffold structures matched expectations from previous studies, indicating sufficient localizations. Transport trajectory behaviors were reproduced between two datasets, indicating reproducibility. |
| Data exclusions | Acceptable criteria for HMSiR localizations were CFR < 0.8 and 60 kHz > EFO > 25 kHz (dataset 1) or 100 kHz > EFO > 50 kHz (dataset 2). Acceptable criteria for trajectories were ≥ 5 localizations, CFR < 0.8 and DCR < 0.5. Data exclusion criteria are discussed in the Methods and main text. |
| Replication | All experimental results were successfully reproduced using different samples and measured on different days. The number of all experimental replicates are specified in the legends of all figures. |
| Randomization | The single molecule data collected were, by their very nature, random. This study did not allocate experimental groups, thus no randomization of data between groups was necessary. |
| Blinding | Blinding was not performed. Data was not allocated into different groups for comparison, and no a priori was assumed about the present observations, so blinding was unnecessary. |

# Reporting for specific materials, systems and methods

We require information from authors about some types of materials, experimental systems and methods used in many studies. Here, indicate whether each material, system or method listed is relevant to your study. If you are not sure if a list item applies to your research, read the appropriate section before selecting a response.

## Materials & experimental systems

| n/a | Involved in the study |
|---|---|
| ☐ | ☒ Antibodies |
| ☐ | ☒ Eukaryotic cell lines |
| ☒ | ☐ Palaeontology and archaeology |
| ☒ | ☐ Animals and other organisms |
| ☒ | ☐ Clinical data |
| ☒ | ☐ Dual use research of concern |
| ☒ | ☐ Plants |

## Methods

| n/a | Involved in the study |
|---|---|
| ☒ | ☐ ChIP-seq |
| ☒ | ☐ Flow cytometry |
| ☒ | ☐ MRI-based neuroimaging |

## Antibodies

| Antibodies used | The anti-GFP nanobody LaG-9 was reported previously (Fridy et al., 2014, Nat. Meth. 11:1253). This is not a commercially available nanobody but was created by the authors of the indicated paper. |
|---|---|
| Validation | The LaG-9 nanobody was validated by comparing binding to a GFP-tagged cell line relative to its untagged control. We reported this earlier (Chowdhury, Sau, & Musser. 2022 Nat. Cell Biol. 24:112). |

## Eukaryotic cell lines

Policy information about cell lines and Sex and Gender in Research

| Cell line source(s) | U-2 OS-CRISPR-NUP96-mEGFP clone #195 (300174, CLS GmbH) and U-2 OS (300364, CLS GmbH) |
|---|---|
| Authentication | Used without authentification. |
| Mycoplasma contamination | Cell lines were not tested for mycoplasma contamination since they were used within one month after thawing. |
| Commonly misidentified lines (See ICLAC register) | None. |

## Plants

| Seed stocks | N/A |
|---|---|
| Novel plant genotypes | N/A |
| Authentication | N/A |

