## [Peer Review File · Nature]

Overlapping nuclear import and export paths unveiled by two-color MINFLUX

Corresponding Author: Dr Siegfried Musser

Version 0:

Reviewer comments:

Referee #1

(Remarks to the Author)

The paper from the Musser lab shows a beautiful application of 3D Minflux tracking to the transport through the NPC. Overall I like the manuscript. The main conclusions on the empty transport region at the NPC center seem plausible given the data and the very low diffusion constants found too.

The authors base their findings on the very long tracks compared to earlier work for the export and import of cargo made possible by the reduced photobleaching of Minflux. The alignment of the different channels seems good enough to claim that the central region is not used for translocation.

The authors mainly build on their own work (ref 6), which is conceptually close. On the image processing part ref 6 as well as the methods used here are not state-of-the-art and need improvement.

Details:

* abstract: Is it really diffusional motion in the pore?

Figure 1:

* 1e) This needs images of single NPCs at least in the SI for about 10 NPC. The quality of Minflux images is up to now not as good as from e.g. PAINT or STORM in terms of degree of labelling (see ref 18) - for unknown reasons. This results in worse structural data than from standard SMLM while the precision is higher with Minflux. Here it is impossible to judge the quality in the terms of degree of labelling. If the authors can do better than state-of-the-art Minflux imaging then their imaging protocol or labelling could be better. Compare the discussion in e.g. Parkash Nature Methods 2022 and the reply. This is not the main point of the paper but of interest to the community.

* 1f,g) You cannot use scatter plot to show localizations. This must be a 2D histogram where the local density is visible. Also in the movies showing the motion over the NPC looks saturated. This is a form of non-linear density display and must be avoided. The same applies to images later like exfig3i)

* Can the authors see the phase shift of the two nuclear rings with respect to each other?

* 1h) misses legend in the figure. Here the diffusion coefficient was assumed to be known, but if that is incorrect the estimates for the localization errors scale. That seems a strange approach as now the localization error is based on an arbitrary assumption. How does this scale as an error in D? From the methods it is also clear that this approach can never disentangle xy vs z precision but the ratio but again be assumed. In conclusion, I cannot understand the rationale of this subpanel i) what is the message and ii) how is that sound given that 2 hard assumptions need to be made? In addition it is unclear why this should be a good estimation to assess minflux or any other SMLM based precision on fixed samples. I suggest to use the repeated localization of the same emitter which is the standard in the SMLM community.

g) what are the two dashed lines at around $z = \pm 20$ nm?

Figure 2:

figure 2d is not isotropic. The axial dimension is upscaled. That is not based on the imaging system. Why is that done?

* page 5 | 32. 131 tracks have been analyzed eventually. From how many cells and different NPCs are these tracks originating?

* z-distance correction factor of 0.7 and implication on the distance between the nuclear and cytoplasmic rings.
Page 16 Channel alignment. Here the z-positions have been scaled by an ad-hoc correction factor of 0.7 based on ref 18 and ExFig3i) is cited. How is it obvious from 3i) that 0.7 is the correct factor and from which physical or optical phenomena does it arise? The histogram of the localization along z in Ex3i) shows a separation distance of the two rings of about 50 nm (from the two peaks). That is consistent with recent SMLM data analysis Wang et al. Scientific Reports, 13:13327, 2023. Here they also find a similar distance, but that is inconsistent with the EM data (from the Martin Beck group). It could be that the same correction factor lead to the same z-separation of the two rings, but it is still inconsistent with the EM data. This must be commented. In addition the factor 0.7 must be explicitly made clear from the figure.

* poor NPC structural reconstructions & model fitting

The NPC is reconstructed by a fit to a 2 ring model based on ref 6. I think this is not up to standards anymore in the community. The data processing community has abandoned model based fitting already a long time ago as this leads to template bias. Methods that avoid this problem are available, e.g. Heydarian et al. Nature Communications, 12:2847, 2021. Here the plane of the rings is even assumed to be in line with the coordinate system of the imaging. The use of 8-fold symmetry for the fitting is again not needed. Overall the method used here for data processing has been appropriated 10+ years ago in the first studies towards NPC Loschberger et al. Journal of Cell Sciences, 125(3):570, 2012; Szyborska et al., Science, 341(6146):655-658, 2013 but is nowadays not acceptable anymore.

The here reconstructed NPC has large deformations. 1) The 8 blobs are elongated in radial direction. This should not be the case. Given the localization uncertainty even the 2 binding sites per blob should be reconstructable. In that case the blobs should be elongated with a tilt towards the rings (as has been shown Wang et al. Scientific Reports, 13:13327, 2023). 2) In addition the lower ring has strange deformations. This is best visible in the SI movies of the tracks, but present in all the visualization. The alignment and reconstruction of the NPC must be done decently without a model and state-of-the-art software.

All this cannot be just attributed to non-fixed cells. Moreover the reconstruction quality is not even discussed, but just applied from ref 6.

* The code: I downloaded the code from git. There is no clear manual on how to use the software. This is not usable for any other than the authors. Here a demo.m file or similar needs to be made to show case the use of the software. The software is present but not in a useful state.

(Remarks on code availability)

The current version of the code is not usable other than by the authors. This needs a second round to make it accessible to others.

Referee #2

(Remarks to the Author)

In their article 'Overlapping nuclear import and export paths unveiled by two-color MINFLUX', Sau et al investigate transport pathways through nuclear pore complexes with two-color 3D MINFLUX. They visualize the positions and structures of nuclear pore complexes in unfixed permeabilized cells using 3D MINFLUX and determine their positional stability. Following this, they track transport complexes containing Imp α as they transition through the nuclear pore complex with ~1 ms temporal resolution in a second color with 3D MINFLUX, unveiling both import and export pathways of these complexes in unprecedented details. With a median length of 17 localizations, these tracks are considerably longer than previously reported and give insight into several novel and very exciting key findings. First, both import and export paths (mainly) remain restricted to a single octant of the nuclear pore complex and circumferential movement is minimal, creating overlapping pathways for import and export. Second, the transitions occur with a very low apparent diffusion coefficient of ~0.06 $\mu\text{m}^2/\text{s}$ and are characterized by transient pauses.

While there are individual preprints on dual-color MINFLUX in the context of particle tracking (bioRxiv 2024, 2024.03.05.583551v1; bioRxiv 2023, 2023.12.09.570565v2), which even focus on tracking two moving particles simultaneously, they are primarily limited to artificial samples or slower movement in 2D. To the best of my knowledge, this work represents the first application of 3D two-color MINFLUX in the context of particle tracking in cells. It powerfully illustrates how this - in combination with the extreme photon-efficiency of MINFLUX - gives direct access into dynamic biological processes in interplay with their environment. As such, it provides a unique mean to answer controversially discussed, previously inaccessible research questions as the transport pathways through the nuclear pore complex. The assay used in this study itself is very similar to that reported in a previous paper of the same group (reference 6 in the manuscript, Nat Cell Biol 24, 112–122 (2022)) in which they studied the same NLS-2xBFP/Imp α /Imp β import pathway (but by labelled importin beta) using astigmatism imaging. Some observations reported in the current manuscript such as that tracks exhibit abortive behavior do not cross the midplane, marking it as a 'point of no return' are not new but were already described in the previous astigmatism study with the same transport complex (label at Imp β instead of Imp α). Similarly, the fact that the transport complex is imported near the periphery of the pore was already demonstrated in the astigmatism study - albeit with lower resolution. While the authors do not claim that these are new findings, I would still recommend adapting the corresponding text passage (page 7, lines 11, 19-20) to make this more apparent to the reader. Nonetheless, the application of 3D two-color MINFLUX instead of astigmatism imaging and the additional study of export pathways significantly enhanced the level of detail with which the transport pathway is examined, enabling the discovery of exciting novel findings in the first place.

Some detailed comments:

The experimental approach, data analysis and drawn conclusions were described in a clear and simple manner, making it easy for the reader to follow. Despite the complexity inherent in the application of 3D dual-color MINFLUX in such

experiments, its execution was convincingly accomplished, resulting in the characterization of the transport pathways with respect to the octahedral subunits of the nuclear pore complexes with both high spatial and temporal resolutions and long track lengths. Raw data is shown and later-applied averaging and filtering steps are strongly applied but well-explained and justified. An overlay precision for the coordinate systems of localizations in the different excitation colors of 2 nm is achieved and the overlay procedure well described (an additional supplementary figure however may still be beneficial for the reader). As such, data quality is high and the data is presented in an excellent manner.

With only 131 tracks beginning to transition a nuclear pore complex and only 12 and 15 tracks which completely pass the complex for import and export, respectively, the manuscript remains on a more descriptive rather than a quantitative level. Given the high number of >5000 tracks initially recorded and the confocal nature of MINFLUX, I believe observing a higher number of processes represents a considerable time effort. Nonetheless, is it possible to extract additional quantitative numbers besides the diffusion coefficients from the recorded tracks? Would e.g. analysis of the kinetics of the switching between both diffusional states in the tracks give further information into what the transient pauses are?

Fig. 4E is neither linked nor discussed in the main text.

The model the authors use to determine both diffusion coefficients and localization errors describes the data visualizing the nuclear pore complexes well for which a very low diffusion coefficient is assumed. As the authors already point out in the methods section, the used model however could not approximate the bimodal behavior of the transport complexes transitioning through the nuclear pore complex equally well (see Extended Data Fig. 5D). This raises the question how accurate the extracted diffusion coefficients of $0.055 \mu\text{m}^2/\text{s}$ and $0.001 \mu\text{m}^2/\text{s}$ are. The authors attribute the discrepancies observed for the transport complexes to distinct D_{xy} and D_z diffusion coefficients caused by the highly constrained diffusional trajectories which seems a reasonable conclusion when looking at the experimental data. Could the authors elaborate on this by including a more detailed model accounting for differing diffusion coefficients for xy and z and/or giving an estimate for these coefficients?

The authors use the term localization error when describing MINFLUX data and the term localization precision when describing data from astigmatism imaging. Is there a reason for this discrepancy? The term localization error implies incorrect rather than imprecise localizations.

Page 2, line 9: Typo in introduction: 'both import and export and import' mentions import twice.

Overall, this is a sound manuscript using state-of-the-art technology applied to long-standing challenges. New insight about transport through nuclear pore complexes is shown. Still, the number of events studied is limited and the disruptive nature of new information has to be evaluated with a more specialized biological background.

Referee #3

(Remarks to the Author)

In this manuscript, the authors describe an innovative application of MINFLUX imaging to study the transport through Nuclear Pore Complexes (NPCs). They employ a correlative approach to image and track single import and export events and follow single proteins as they translocate through NPCs, achieving nanometer precision and millisecond time resolution. This method allows them to observe several key aspects of nucleocytoplasmic transport:

1. The duration of translocation events through the NPC typically ranges from approximately 10 to 20 milliseconds.
2. Import and export processes occur within overlapping spatial regions, indicating the absence of dedicated channels specific to each process.
3. Translocation predominantly occurs at the periphery of the NPC, with a notable exclusion zone in the center.
4. Both import and export processes are largely confined to a single octant of the NPC.
5. Transient pauses are observed during the transport process.
6. An apparent diffusion coefficient of approximately $0.06 \mu\text{m}^2/\text{s}$ was recorded during translocation, akin to the diffusion behavior of proteins in a highly viscous medium such as glycerol.

From a technical standpoint, the paper appears to be of high quality, demonstrating the successful deployment of the complex MINFLUX imaging methodology for single particle tracking. In addition, the authors show that MINFLUX can be used to describe nucleocytoplasmic transport at the single molecule level revealing important and novel details of transport through the NPC. Nevertheless, there are several areas where further clarification and enhancement could strengthen the findings.

Major Comments:

- The manuscript highlights that 32 of the 33 tracks remain confined to a single local octant, suggesting highly defined translocation routes. This is probably the most unexpected finding that cannot be explained with current models of NPC function and selectivity. Also, current models of NPC structure do not provide any structural basis for such a spatial restriction of NPC translocation, and it is not clear how the interactions between importins and the nucleoporin FG repeats would be limited to one octant during translocation. This needs to be discussed in more detail. Could the authors propose a speculative mechanism for this restriction? Alternatively, could there be any image processing steps that inadvertently bias the localization of tracks toward one octant? Do the authors potentially observe stable transport receptor-FG repeat interactions and could the movement that they detect reflect FG repeat mobility? It would be important that the authors address this issue and potentially provide additional experimental evidence.
- Additionally, it is important to determine how many of the 33 tracks that translocate through the Nuclear Pore Complex (NPC) interact with two different octants—one in the Cytoplasmic Ring (CR) and another in the Nuclear Ring (NR), as depicted in Figure 3B. Alternatively, it would be useful to know whether these trajectories typically bind to just one octant, either in the CR or the NR, and then translocate without further binding events. Furthermore, it is necessary to explore whether the speed of the MINFLUX scanning could introduce any biases in the observations of these two scenarios. Regarding the localization of tracks within the same octant in both the Cytoplasmic Ring (CR) and Nuclear Ring (NR), how

frequently is this observed? Is there sufficient data to support or refute this occurrence, or do most trajectories simply dwell within one octant before crossing the NPC?

- The paper posits the existence of a central exclusion zone in the NPC where no import or export events are detected. Could this absence be attributed to the rapid movement of proteins through the center, potentially exceeding MINFLUX's temporal resolution? Furthermore, if these trajectories are indeed very rapid, could their photon count fall below the detection threshold, leading to their exclusion from analysis revealing only long-lived interactions at the periphery? Further justification and exploration of this point would be valuable.
- The authors are encouraged to include an XZ visualization plot of all 33 trajectories that were successfully tracked during import or export in the supplement, similar to what is presented in Figure 3B.
- A significant limitation is the low throughput of the experiment. Although the complexity of the setup is acknowledged, the collection of only 33 trajectories prevents a robust statistical basis to comprehensively characterize nucleocytoplasmic transport dynamics. Increasing the number of tracked trajectories by an order of magnitude would greatly enhance the reliability of the conclusions. The authors should at least discuss why they limit their analyses to such a low number of trajectories.

Minor Comments:

- The authors state that current models of NPC translocation propose different channels for import and export within the NPC. Is there substantial experimental support for such a model and is this indeed accepted within the field? Additional references or a more detailed discussion of this topic would be beneficial.
- The error estimation procedure assumes a Gaussian distribution of jump sizes based on the diffusion coefficient and time lag between localizations. Considering that importins interact with the NPC scaffold, which leads to anomalous diffusion, does this assumption affect the accuracy of localization precision estimates? Could the authors delve deeper into this aspect?
- The filtration criteria outlined in Extended Data Figure 6 should be more clearly explained, as understanding these criteria are crucial to evaluate the data.

Version 1:

Reviewer comments:

Referee #1

(Remarks to the Author)

I have been very much impressed by the resubmission of the authors. They went to great length in the data acquisition and analysis to address all my comments and the other two reviewers comments.

On the technical side, the idea to use the jump histogram turned out to be good. The loc. unc. analysis and the influence on the error propagation has been commented by all reviewers. Now I think it is sound and nice.

I do not have any further comments and recommend publication.

(Remarks on code availability)

The now is fine.

Referee #2

(Remarks to the Author)

The revised manuscript has addressed the points I raised during the previous round of review. I however have some remaining minor comments.

- 1) The authors measured a further dataset to strengthen the statistical relevance of their claims. While they adjusted some parameters for Imp a-JF549 3D tracking, the parameters for NbGFP-HMSiR 3D localizations remained the same (Table S1). When looking at the time between successive MINFLUX localizations, dataset 2 shows a sharp peak at 3-4 ms whereas dataset 1 peaks between 5-6 ms and has a very strong tail. Could the authors comment on where these differences arise from?
- 2) The histograms in Fig. 1G, H show no scalebar. When comparing to ED Fig. 4G, it becomes apparent that the scalebar does not start at 0 but only at 30 localizations. Please add scalebars to clarify.
- 3) In the introduction, it is stated that the tracking is limited to diffusional steps of most 100-200 nm, dependent on donut size. Does this not rather depend on the size of the TCP?
- 4) With the added tracks, Fig. 3E,F,I,J,M and N are overly crowded, making it difficult for the reader to extract information. Please reduce the number of tracks shown.
- 5) When analyzing the diffusional behavior of Imp a-JF549, the authors now directly jump from a model with one species to a model with three species. Could a model assuming two species also fit the data? The model with three species additionally allows for a lot of free fit parameters. Especially when looking at the two non-diffusive species, are there any other combinations of parameters which allow explaining the data equally well?

Referee #3

(Remarks to the Author)

The authors have done a good job addressing the comments of the reviewers. Importantly, they have increased the number

of trajectories, have addressed most technical issues and have clarified the text and the discussion. I remain excited about the results and recommend publication.

November 4, 2024

Senior Editor, Nature

Re: Nature Manuscript 2024-03-06173
"Overlapping nuclear import and export paths unveiled by two-color MINFLUX"

Dear Editor,

Thank you for shepherding the review of the above manuscript and for inviting us to resubmit a revised version.

We appreciate the helpful and insightful comments of the Reviewers. As requested, we have made significant improvements in data processing and presentation to be more consistent with the state-of-the-art. We have expanded our discussion to clarify issues and address comments raised by the Reviewers, including making the code more user-friendly. We have also almost doubled our dataset size and introduced stricter criteria for pore and track selection, thus strengthening our conclusions and demonstrating reproducibility. Due to the efforts needed to generate this revised manuscript, the author order has been changed to appropriately acknowledge the contributions to this revision.

We have included a track changes version of the manuscript, as per request. As extensive modifications have been made, we are also submitting a clean version of the manuscript for easier review.

In our point-by-point response below, comments from the Reviewers are in *italics*, and our responses follow in blue. Page and line numbers in our responses below refer to the revised clean version of the manuscript. With the exception of **ED Fig. 10**, all of the figures and tables have been edited, though these changes are not marked in the track changes version of the manuscript. **Table S2** and **ED Figs. 4, 7, & 8** are entirely new; other table and figure edits range from minor to substantial.

REVIEWER 1:

The paper from the Muser lab shows a beautiful application of 3D Minflux tracking to the transport through the NPC. Overall I like the manuscript. The main conclusions on the empty transport region at the NPC center seem plausible given the data and the very low diffusion constants found too.

The authors base their findings on the very long tracks compared to earlier work for the export and import of cargo made possible by the reduced photobleaching of Minflux. The alignment of the different channels seems good enough to claim that the central region is not used for translocation.

The authors mainly build on their on work (ref 6), which is conceptually close. On the image processing part ref 6 as well as the methods used here are not state-of-the-art and need improvement.

We thank the Reviewer for highlighting the key points of this manuscript and the positive overall evaluation. The issues with image processing and our corrections are detailed later.

Details:

** abstract: Is it really diffusional motion in the pore?*

Over the past decades, the field has debated whether there are affinity gradients, or sliding tracks that could bias motion. Motor proteins are clearly not present. The simplest model consistent with the available data – including the data in this paper – is that movement is governed by diffusive steps, but these are certainly constrained by the shape of the pore, transient binding events, and the permeability barrier network. Movement across the pore does not indicate directed movement, but simply that enough random steps in the right direction have occurred to achieve successful transport. The high number of abortive events are consistent with this picture. Nonetheless, we have deleted the word ‘diffusional’ from the abstract, as this is unnecessary.

Figure1:

** 1e) This needs images of single NPCs at least in the SI for about 10 NPC. The quality of Minflux images is up to now not as good as from e.g. PAINT or STORM in terms of degree of labelling (see ref 18) - for unknown reasons. This results in worse structural data than from standard SMLM while the precision is higher with Minflux. Here it is impossible to judge the quality in the terms of degree of labelling. If the authors can do better than state-of-the art Minflux imaging then their imaging protocol or labelling could be better. Compare the discussion in e.g. Parkash Nature Methods 2022 and the reply. This is not the main point of the paper but of interest to the community.*

Yes, the image quality for single NPCs from our labeling and imaging protocol using the spontaneously blinking dye HMSiR is not as high as from a PAINT or STORM approach. However, the purpose of our imaging protocol is to obtain sufficient structural scaffold information quickly on functional pores to provide context for transport trajectories before the NPC positions become too unstable. PAINT typically includes fixation, which is incompatible with nucleocytoplasmic transport, and the strand exchange rate leads to slower acquisition than the fast blinking HMSiR. STORM relies on GLOX-buffer with thiols, which is also incompatible with transport. Individual NPC scaffolds are now shown in **Fig. 1F** and **ED Fig. 3A**. The number of HMSiR localizations is primarily limited by two factors:

1) Limited Acquisition Time – The NPCs in unfixed, permeabilized cells are positionally stable for 0.5-1 h, and transport is most active in the first 45 minutes. We therefore typically allocate 15-20 minutes for HMSiR localizations and another 15-20 minutes for tracking. In contrast, PAINT data on NPC scaffolds is typically acquired over a few hours, at minimum.

2) Labeling Efficiency and the Effect of Wash Steps – Complete labeling of all GFP domains by the nanobody was never achieved, for unknown reasons. In addition, every wash step caused some nanobody to wash off of the NPCs. Washes were therefore minimized, but the application of transport mix acts as a wash step removing some nanobody-HMSiR.

** 1f,g) You cannot use scatter plot to show localizations. This must be a 2D histogram where the local density is visible. Also in the movies showing the motion over the NPC looks saturated. This is a form of non-linear density display and must be avoided. The same applies to images later like exfig3i)*

All aligned pore images have now been converted to 2D density histogram format, including in the movies. We appreciate the reviewer's suggestion, as the NPCs indeed appear more visually appealing in this format.

* Can the authors see the phase shift of the two nuclear rings with respect to each other?

We were able to detect a minimal (1.4°) phase shift between the two rings of the NPC. To understand why we could not detect a larger phase shift (a maximum of 7.8° is expected), we performed a series of simulations and compared them with our experimental data. Using a previously developed algorithm (ref. 6), we simulated the effect of localization precision and jiggle (normally distributed positional fluctuations) on localization distributions. In **Fig. R1**, angular distributions are shown for the localizations from the top ring (*blue*) and bottom ring (*magenta*) of the NPC under various conditions. The angle distributions were fit to $y = y_0 + c \cdot \sin(8(x - \phi))$, where y_0 , c and ϕ are fit parameters (see **ED Fig. 3D,F**). The first panel identifies the 3D positions of the C-termini of Nup96 from the cryo-electron microscopy density map for the human NPC and identifies the ground-truth inter-ring angle of 7.8 degrees. As localization precision worsened, the detected phase shift decreased (panels 2 & 3). With an NPC jiggle of ~ 8 nm (as we previously estimated in unfixed cells – ref. 6) and localization precisions approximating those in our MINFLUX experiments (**Fig. 1I**), a phase shift of 1.6° was detectable (panel 4). These simulated results were comparable with the experimental data (panel 5 – data from **Fig. 1G**). This analysis does not include any centroid misalignment or axial tilt error of individual pores to yield the composite picture – considering the low number of localizations/NPC, a centroid misalignment of a few nanometers and a tilt error of low single digit degrees are not unreasonable. Low labeling densities of the two rings contribute to an inability to resolve the expected phase shift. In contrast to our conditions, the higher labeling efficiencies for Nup96-GFP and Nup107-GFP in fixed cells and a localization precision < 4 nm likely enabled the detection of more substantial phase shifts in previous reports (Heydarian et al., *Nature Commun*, 2021, 12:2847; Wu et al., *Nature Meth*, 2022, 20:139). The phase shift between the two rings does not influence our interpretation of the transport trajectories for Imp α -JF549 and is therefore not discussed in the manuscript.

Figure R1. High Resolution is Required to Determine the Phase Shift between the Cytoplasmic and Nucleoplasmic Rings of the NPC.

* 1h) misses legend in the figure. Here the diffusion coefficient was assumed to be known, but if that is incorrect the estimates for the localization errors scale. That seems a strange approach as now the localization error is based on an arbitrary assumption. How does this scale as an error in D ? From the methods it is also clear that this approach can never disentangle xy vs z precision but the ratios can again be assumed. In conclusion, I cannot understand the rationale of this subpanel i) what is the message and ii) how is that sound given that 2 hard assumptions need to be made? In addition it is unclear why this should be a good estimation to assess miniflux or any other SMLM based precision on fixed samples. I suggest to use the repeated localization of the same emitter which is the standard in the SMLM community.

We agree that we did not present our error analysis approach with sufficient depth and context. We have expanded and clarified our discussion of this analysis method in various places throughout the manuscript. We now include localization precision values estimated from repeated localizations of 'static particles', i.e., HMSiR on the NPCs in the experiment (**Fig. 1I**). These yielded localization precisions of $\sigma_x = 6.5$ nm, $\sigma_y = 7.0$ nm and $\sigma_z = 4.2$ nm; the resultant precision ratios $\sigma_x/\sigma_y = 0.93$ and $\sigma_x/\sigma_z = 1.55$ were used throughout the manuscript for all simulations.

The Reviewer's comments motivated us to explore a deeper analysis. Analysis of repeated HMSiR localizations via a jump step histogram should in principle be consistent with precision estimates obtained from the repeated localization method. However, as we now show using jump step analysis (**Fig. 1J**), the repeated localization approach yielded localization precision estimates that are too high (by ~37%, or 1.6-2.6 nm). Jump step histograms are largely insensitive to drift, as they analyze pairs of localizations closely spaced in time. In contrast, for repeated localization measurements, a centroid from the trajectory must be calculated and compared with each point in the trajectory. Drift during the trajectory acquisition will widen the localization distribution, leading to an overestimate of the precision. Our analysis indicated, therefore, that the HMSiR dye molecules were not static. While multiple fitting strategies were attempted, the simplest approach to fit the data turned out to be a simulation model including diffusional drift and 37% lower localization precisions (**Fig. 1J**, red curve). These simulated data yielded centroid deviations that matched the experimentally-determined values (**ED Fig. 5I**), and they matched the experimental data when plotted as an R^2/t histogram (**ED Fig. 5H**). This 'diffusional drift' is not expected to be sample drift that would generate large displacements over the total imaging time, as this would make the assembly of the NPC scaffold from the data impossible. Rather, the 'diffusional drift' is expected to be confined and models the movement of the dye centroid, e.g., to favorable positions enabled by linkage error, or conformational shifts or jiggles of the NPC scaffold. Considering that the nanobody-bound HMSiR fluorophore is potentially up to 6 nm away from the NUP96 attachment point of GFP, a low nanometer-scale zone in which the dye maintains distinct preferential positions is reasonable (discussed on p. 4, lines 11-31, and p. 23, line 39 to p. 24, line 5). Thus, our jump step histogram approach provided for a more informed analysis of the NPC localization data.

The previously assumed diffusion coefficient for the HMSiR localization data was indeed too high. The simulation code has been updated so that no diffusional component is needed for modeling static particles.

g) what are the two dashed lines at around $z = \pm 20$ nm?

These dashed lines marked the approximate position of the nuclear envelope. This figure panel has been revised and the lines removed. Other figure captions now indicate when the nuclear envelope position is estimated within figure panels.

Figure 2:

figure 2d is not isotropic. The axial dimension is upscaled. That is not based on the imaging system. Why is that done?

We thank the reviewer for highlighting this issue. The z scale was incorrectly scaled, and there was a color conversion error during export. These have now been corrected – the abscissa and ordinate scales are now identical. Additionally, we now present a larger area compared with the previous submission.

* page 5 | 32. 131 tracks have been analyzed eventually. From how many cells and different NPCs are these tracks originating?

In the previous version of the manuscript, these 131 tracks originated from 23 cells and 254 NPCs. Data statistics are summarized in **Table S4**, as indicated on p. 5, line 24. This table has now been updated due to the inclusion of new data (a total of 225 tracks from 37 cells and 541 NPCs).

* z-distance correction factor of 0.7 and implication on the distance between the nuclear and cytoplasmic rings.

Page 16 Channel alignment. Here the z-positions have been scaled by an ad-hoc correction factor of 0.7 based on ref 18 and ExFig3i) is cited. How is it obvious from 3i) that 0.7 is the correct factor and from which physical or optical phenomena does it arise? The histogram of the localization along z in Ex3i) shows a separation distance of the two rings of about 50 nm (from the two peaks). That is consistent with recent SMLM data analysis Wang et al. Scientific Reports, 13:13327, 2023. Here they also find a similar distance, but that is inconsistent with the EM data (from the Martin Beck group). It could be that the same correction factor lead to the same z-separation of the two rings, but it is still inconsistent with the EM data. This must be commented. In addition the factor 0.7 must be explicitly made clear from the figure.

This is an important point. The z scaling factor of 0.7 is a value recommended by the manufacturer of the MINFLUX microscope (Abberior, Göttingen, Germany). This factor was calculated by the manufacturer based on simulations from Leutenegger et al (<https://opg.optica.org/oe/fulltext.cfm?uri=oe-14-23-11277&id=117883>). The underlying optical effect is a result of spherical aberrations caused by the difference in refractive index between the immersion medium (1.51) used with the objective and the medium of the sample (~1.33), thus necessitating a correction of the measurement data. The z scaling of 3D MINFLUX data has been described and applied in previous publications (refs. 15-17); however, the only direct measurement of the z scaling factor that we are aware of is 0.69 (ref. 50).

Since the calculated z scaling factor is only an approximation provided by the manufacturer and may vary depending on the experimental conditions (imaging depth, microscope imperfections, etc.), we have now estimated the z scaling factor by comparing NPC scaffold data obtained under identical conditions using single molecule localization by astigmatism imaging and MINFLUX. Astigmatism imaging was used to obtain NPC scaffold information for aligning the mEosEM localizations in **Fig. 4** and **ED Fig. 9**. This yielded four independent datasets (**Table S2**) in addition to our previous measurements (ref. 6), yielding an average ring spacing of 51.5 ± 1.1 nm (**ED Fig. 4E**). The uncorrected MINFLUX data yielded a ring spacing of 76.8 ± 0.8 nm (**ED Fig. 4B**). From these, we calculate a z scaling factor for the MINFLUX measurements of $51.5 \text{ nm} / 76.8 \text{ nm} = 0.67$, which is very close to the value obtained through simulations by the microscope manufacturer. This experimentally-

determined value was used for all z axis scaling. These data and z scaling calculation are summarized in **Table S2**, illustrated in **ED Fig. 4**, and discussed in the 'Measuring the Z Scaling Factor' section of the Methods (p. 18).

Comparisons with EM data should be made cautiously. Different conditions can produce structural changes – structural dilation/constriction of NPCs is known to occur and the conditions that produce changes are still poorly characterized. Moreover, the linkage error between dyes and the NPC scaffold proteins to which they are attached can be significant. Thus, the determined structures from super-resolution imaging methods can yield very good approximations to EM structural data, but they do not yield absolute distances expected for the actual NUP96 proteins in both rings. Finally, noticeable differences between calculated structures obtained by distinct labeling schemes (e.g., between Halo-tags and nanobodies, or between imaging methods using distinct nanobodies) are also not unexpected at the current resolution levels. In short, our images are consistent with the expectations from EM data.

** poor NPC structural reconstructions & model fitting*

The NPC is reconstructed by a fit to a 2 ring model based on ref 6. I think this is not up to standards anymore in the community. The data processing community has abandoned model based fitting already a long time ago as this leads to template bias. Methods that avoid this problem are available, e.g. Heydarian et al. Nature Communications, 12:2847, 2021.

Here the plane of the rings is even assumed to be in line with the coordinate system of the imaging. The use of 8-fold symmetry for the fitting is again not needed. Overall the method used here for data processing has been appropriated 10+ years ago in the first studies towards NPC Loschberger et al. Journal of Cell Sciences, 125(3):570, 2012; Szyborska et al., Science, 341(6146):655-658, 2013 but is nowadays not acceptable anymore.

We appreciate the advantages of model-independent alignment, and we have explored such an approach using both experimental and simulated data under the constraints dictated by the experiments. We found that while a model-free approach has advantages when certain conditions are fulfilled, it does not provide any analytical benefit for the current data.

A prerequisite for model-free fitting is that all segmented particles are samples of the same underlying structure. With enough segmented particles as input, the first step is to construct an initial template with an all-to-all registration of all segmented particles. With this data-driven template, the model-free fitting algorithm updates the template through a minimization of a global error function of template-to-all registration. The final output of such an approach is a fused particle that best describes the underlying structure (Heydarian et al., *Nature Commun*, 2021, 12:2847; Wu et al., *Nature Meth*, 2022, 20:139).

We expended significant effort to implement the protocol of Heydarian et al, as suggested by the Reviewer. Unfortunately, the available code was not compiled, it required different software versions, and despite contact with the original authors, we were not able to fully implement the published protocol. Instead, we employed a similar and recent model-free approach using 'SMAP' and 'LocMoFit' as described by Wu et al. (*Nature Meth*, 2022, 20:139). Using PAINT data that we acquired on fixed cells, the Wu protocol performed well (**Fig. R2A**). The reconstructions obtained using Nb^{GFP}-HMSiR data acquired for unfixed transport-active NPCs were poor, however, due to insufficient labeling density (**Fig. R3**). This limitation of low labeling density was identified earlier (Heydarian et al., *Nature Meth*, 2018, 15:781; Heydarian et al., *Nature Commun*, 2021, 12:2847; Wang et al., *Sci Rep*, 2023, 13:13327; Wu et al., *Nature Meth*, 2022, 20:139), and confirmed by our

own efforts. A model-free fitting algorithm requires that localizations are obtained for a considerable portion of the target structure to generate a model template from the individual structures. This indicates a need for a relatively high labeling density. As each input particle must share recognizable similarity to each other – and eventually to the underlying structure model – sufficient localizations per fluorophore to yield low error in the fluorophores' centroid positions are needed. In quantitative terms, more than 50% of all subunits of the NPC scaffold should be labelled (> 8 of 16 lobes). This condition was not met by Nb^{GFP}-HMSiR data, which yielded $< 5\%$ of NPCs that had localizations for > 8 of the 16 lobes. The total localizations per NPC averaged 259 for HMSiR data (**ED Fig. 3G**) and 1572 for PAINT data, indicating poorer sampling for the input structures for the former. Given the limitations on pore stability, we were unable to increase the localization density per pore for HMSiR data by increasing acquisition time.

The quality of the NPC scaffold structure obtained by the model-based reconstruction of HMSiR localization data (**ED Fig. 4G**) clearly provides a reasonable template for interpreting Imp α -JF549 tracking data. Considering this, the time constraints for imaging permeabilized cells due to pore instability (see our reply to this Reviewer's third comment), and the infeasibility of the model-free approach for Nb^{GFP}-HMSiR data (**Fig. R3**), we retained our model-based approach for generating all of our NPC reconstructions. We emphasize that the purpose of this study is not to develop an improved NPC imaging method or to learn something about the NPC scaffold. Rather, our goal is to use the NPC scaffold as a reference for trajectories of particles moving through the pore. In this regard we have been successful, and the NPC scaffold structures for functional pores that are now displayed in 2D density histogram format are consistent with the current literature.

Figure R2. Model-free and Model-based Reconstructions of NPCs from PAINT Data. U2OS cells with NUP96-EGFP were fixed with 4% paraformaldehyde, permeabilized with digitonin, decorated with MASSIVE-TAG-X2 anti-GFP nanobodies (Massive Photonics, Germany), and imaged via MINFLUX using DNA PAINT with nanobody-specific imager strands. The input MINFLUX data were not z scaled. (A) Model-free fitting result. (B) Inter-ring distance and radius measurements from the model-free fitting. (C) Model-based fitting result (as outlined in **ED Fig. 3**). (D) Inter-ring distance and radius measurements from the model-based fitting.

Figure R3. Model-free and Model-based Reconstructions of NPCs from HMSiR Data. (A,B) MINIFLUX images. These are approximately the fields shown in Fig. 1D,E. (C,D) Model-free fitting result of the data in A,B. (E,F) Model-based fitting result of the data in A,B.

Technical note: The model-free approach is sensitive to the axial tilt of the input clusters. A larger number of clusters are required to obtain reasonable reconstructions from simulated data with an increased range of axial tilt angles. The influence of axial tilt was already recognized in our previous work (ref. 6), and we therefore limited the useful NPCs to those with an axial tilt of $< \sim 10^\circ$ in our workflow. Simulations confirm that this constraint leads to a minimal increase in alignment errors considering the localization precision of the experiments.

The here reconstructed NPC has large deformations. 1) The 8 blobs are elongated in radial direction. This should not be the case. Given the localization uncertainty even the 2 binding sites per blob should be reconstructable. In that case the blobs should be elongated with a tilt towards the rings (as has been shown Wang et al. Scientific Reports, 13:13327, 2023). 2) in addition the lower ring has strange deformations. This is best visible in the SI movies of the tracks, but present in all the visualization. The alignment and reconstruction of the NPC must be done decently without a model and state-of-the-art software.

All this cannot be just attributed to non-fixed cells. Moreover the reconstruction quality is not even discussed, but just applied from ref 6.

In the previous version of the manuscript, the reference structures included in the figures and movies were unfortunately generated from a subset of the total pores imaged. With the inclusion of reference structures generated from all NPC clusters and the increased size of our dataset, the 'deformation' issue has been resolved.

Comparisons with previous super-resolved NPC images should be tempered by the lower number of localizations/NPC obtained from HMSiR labels compared with, for example, the number of localizations that can be obtained by PAINT (see answer to previous comment). Increased sample instability due to the unfixed cells is certainly an issue (see Fig. 1I,J). The labeling location on the GFP fusion protein also significantly influences the reconstruction. As mentioned in the response to

the previous comment, noticeable differences can be observed for the same GFP-tagged pores with different imaging schemes involving different nanobodies. These data suggest that the labeling scheme can produce a more substantial influence on the averaged structural scaffold than our reconstruction method. Finally, while localization precision is in the mid-nanometer range, reconstruction quality will be influenced by pore alignment precision, both in angular rotation (in-plane and tilt) and centroid localization. These are simply not good enough due to the issues discussed earlier to identify two binding sites per blob.

** The code: I downloaded the code from git. There is no clear manual on how to use the software. This is not usable for any other than the authors. Here a demo.m file or similar needs to be made to show case the use of the software. The software is present but not in a useful state.*

Referee #1 (Remarks on code availability):

The current version of the code is not usable other than by the authors. This needs a second round to make it accessible to others.

We acknowledge that the original commit of code on the GitHub repository was not user-friendly. We have updated the code to improve readability and reproducibility and made it more intuitive for use by a novice user. A description and explanation for the steps and parameters is now provided via documentation, comments inside the code, figure captions, and explicit workflow step reporting to the MATLAB console. New functions were added to facilitate semi-automated NPC clustering, checking the fitted result, and result visualization. Intermediate steps were combined to make the whole workflow more compact and easier to follow. A demo script has been included that runs through the workflow with default parameters and model data, and then displays a 3D visualization of individual tracks relative to the NPC scaffold. This sample data is also locally stored to the GitHub repository (github.com/npctat2021/MINFLUX_NPC_Tracking). The README section on GitHub has also been revised, providing step-by-step instructions on how to use the codes.

Technical note: In the repository, the file "Nuclear Pore Model Data.txt/.mat" includes authentic experimental measurements for Nb^{GFP}-HMSiR from a permeabilized cell. The file "Tracks Model Data.txt" consists of example tracks derived from multiple image fields, artificially aligned to the nuclear pore scaffolds for illustrative purposes, demonstrating the functionality of the fitting and alignment routine. "Bead NPC/Track" provides synthetic coordinates from two channels, based on the average positional differences obtained in bead measurements. While efforts were made to preserve experimental integrity of the model dataset, the number of tracks recovered per image field was low (typically 1-2). Therefore, these model tracks are for demonstrating code functionality and should not be used for drawing biological conclusions.

REVIEWER 2:

In their article 'Overlapping nuclear import and export paths unveiled by two-color MINFLUX', Sau et al investigate transport pathways through nuclear pore complexes with two-color 3D MINFLUX. They visualize the positions and structures of nuclear pore complexes in unfixed permeabilized cells using 3D MINFLUX and determine their positional stability. Following this, they track transport complexes containing Imp α as they transition through the nuclear pore complex with ~ 1 ms temporal resolution in a second color with 3D MINFLUX, unveiling both import and export pathways of these complexes in unprecedented details. With a median length of 17 localizations, these tracks are considerably

longer than previously reported and give insight into several novel and very exciting key findings. First, both import and export paths (mainly) remain restricted to a single octant of the nuclear pore complex and circumferential movement is minimal, creating overlapping pathways for import and export. Second, the transitions occur with a very low apparent diffusion coefficient of $\sim 0.06 \mu\text{m}^2/\text{s}$ and are characterized by transient pauses.

While there are individual preprints on dual-color MINFLUX in the context of particle tracking (bioRxiv 2024, 2024.03.05.583551v1; bioRxiv 2023, 2023.12.09.570565v2), which even focus on tracking two moving particles simultaneously, they are primarily limited to artificial samples or slower movement in 2D. To the best of my knowledge, this work represents the first application of 3D two-color MINFLUX in the context of particle tracking in cells. It powerfully illustrates how this - in combination with the extreme photon-efficiency of MINFLUX – gives direct access into dynamic biological processes in interplay with their environment. As such, it provides a unique mean to answer controversially discussed, previously inaccessible research questions as the transport pathways through the nuclear pore complex.

We thank the Reviewer for this succinct summary of our findings.

The assay used in this study itself is very similar to that reported in a previous paper of the same group (reference 6 in the manuscript, Nat Cell Biol 24, 112–122 (2022)) in which they studied the same NLS-2×BFP/Imp α /Imp β import pathway (but by labelled importin beta) using astigmatism imaging. Some observations reported in the current manuscript such as that tracks exhibit abortive behavior do not cross the midplane, marking it as a ‘point of no return’ are not new but were already described in the previous astigmatism study with the same transport complex (label at Imp β instead of Imp α). Similarly, the fact that the transport complex is imported near the periphery of the pore was already demonstrated in the astigmatism study – albeit with lower resolution. While the authors do not claim that these are new findings, I would still recommend adapting the corresponding text passage (page 7, lines 11, 19-20) to make this more apparent to the reader. Nonetheless, the application of 3D two-color MINFLUX instead of astigmatism imaging and the additional study of export pathways significantly enhanced the level of detail with which the transport pathway is examined, enabling the discovery of exciting novel findings in the first place.

A more careful delineation of what part of our findings are confirmatory and those that are truly new is now explained throughout the text. For the ‘point of no return’ issue, our previous work is explicitly noted (p. 7, lines 11-12). For the peripheral transport, the data from our previous work are identified on p. 9 line 10 and in the caption for **ED Fig. 9G**.

For clarification, in both the previous and current work, the labeled protein was importin alpha (not importin beta). However, the current study included proteins required for export, thus allowing nuclear export to be simultaneously examined.

Some detailed comments:

The experimental approach, data analysis and drawn conclusions were described in a clear and simple manner, making it easy for the reader to follow. Despite the complexity inherent in the application of 3D dual-color MINFLUX in such experiments, its execution was convincingly accomplished, resulting in the characterization of the transport pathways with respect to the octahedral subunits of the nuclear pore complexes with both high spatial and temporal resolutions and long track lengths. Raw data is shown and later-applied averaging and filtering steps are strongly applied but well-explained and justified. An overlay precision for the coordinate systems of

localizations in the different excitation colors of 2 nm is achieved and the overlay procedure well described (an additional supplementary figure however may still be beneficial for the reader). As such, data quality is high and the data is presented in an excellent manner.

We thank the Reviewer for this strong support of our approach.

With only 131 tracks beginning to transition a nuclear pore complex and only 12 and 15 tracks which completely pass the complex for import and export, respectively, the manuscript remains on a more descriptive rather than a quantitative level. Given the high number of >5000 tracks initially recorded and the confocal nature of MINFLUX, I believe observing a higher number of processes represents a considerate time effort. Nonetheless, is it possible to extract additional quantitative numbers besides the diffusion coefficients from the recorded tracks? Would e.g. analysis of the kinetics of the switching between both diffusional states in the tracks give further information into what the transient pauses are?

The Reviewer is indeed correct that it is highly challenging to obtain a large number of high-quality tracks. The stochastic nature of the confocal scouting pattern of MINFLUX makes capturing complete import and export events rare. Some tracks started from the mid-region of NPC scaffolds and were rejected from the analysis since they could not be definitively classified as import or export events. We have increased the dataset size (32 and 23 complete import and export events, and a total of 225 trajectories that interacted with the central pore of the NPC), which bolsters support for our initial conclusions. While we intend to improve throughput in future work, we do not believe that the current dataset is of sufficient size to assess switching frequency or to probe the reason for the transient pauses at this time, both of which could be location dependent. However, the new jump step histogram analysis (**Fig. 3O,P**) indicates that transiting particles are immobile (i.e., not diffusing) over half the time, which is a new conclusion. In fitting these data, it is clear that a substantial immobile fraction is required, though there is uncertainty as to the appropriate details for the most acceptable model (see **ED Fig. 5J,K**).

Fig. 4E is neither linked nor discussed in the main text.

This was an unfortunate error that resulted from final editing before submission. All panels in **Fig. 4** (which has been revised) are now linked and discussed in the text.

The model the authors use to determine both diffusion coefficients and localization errors describes the data visualizing the nuclear pore complexes well for which a very low diffusion coefficient is assumed. As the authors already point out in the methods section, the used model however could not approximate the bimodal behavior of the transport complexes transitioning through the nuclear pore complex equally well (see Extended Data Fig. 5D). This raises the question how accurate the extracted diffusion coefficients of $0.055 \mu\text{m}^2/\text{s}$ and $0.001 \mu\text{m}^2/\text{s}$ are. The authors attribute the discrepancies observed for the transport complexes to distinct D_{xy} and D_z diffusion coefficients caused by the highly constrained diffusional trajectories which seems a reasonable conclusion when looking at the experimental data. Could the authors elaborate on this by including a more detailed model accounting for differing diffusion coefficients for xy and z and/ or giving an estimate for these coefficients?

As described in our response to Reviewer 1 regarding concerns about the jump step analysis model, we have re-examined our analysis and investigated multiple potential models to explain the distributions obtained from the scaffold and tracking localizations. Based on localization precisions estimated from centroid deviations, we fixed the localization precision ratios as $\sigma_x/\sigma_y = 0.93$ and $\sigma_x/\sigma_z = 1.55$, which reduced the number of fitting parameters. Importantly, we also recognized that R^2/t histograms have different sensitivities to the parameters that influence jump step histograms (discussed on p. 23, lines 20-38). This realization led us to parameter sets that simultaneously reproduced both jump step and R^2/t histograms for the tracking localizations (**Fig. 3O,P** and **ED Fig. 5J,K**). Thus, there is no longer any reason to invoke distinct D_{xy} and D_z diffusion coefficients.

The authors use the term localization error when describing MINFLUX data and the term localization precision when describing data from astigmatism imaging. Is there a reason for this discrepancy? The term localization error implies incorrect rather than imprecise localizations.

This was an inadvertent use of two terms to mean the same thing. We now exclusively use the term 'localization precision'. We thank the reviewer for pointing this out.

Page 2, line 9: Typo in introduction: 'both import and export and import' mentions import twice.

Fixed.

Overall, this is a sound manuscript using state-of-the-art technology applied to long-standing challenges. New insight about transport through nuclear pore complexes is shown. Still, the number of events studied is limited and the disruptive nature of new information has to be evaluated with a more specialized biological background.

At various points within the manuscript, we now more clearly point out the impact of the new findings in an appropriate biological context. Points 2-6 identified by Reviewer 3 in the paragraph that follows are novel findings of this study, and these are outlined in the first paragraph of the Discussion. The three-zone annular model (**Fig. 4F**) is novel.

REVIEWER 3:

In this manuscript, the authors describe an innovative application of MINFLUX imaging to study the transport through Nuclear Pore Complexes (NPCs). They employ a correlative approach to image and track single import and export events and follow single proteins as they translocate through NPCs, achieving nanometer precision and millisecond time resolution. This method allows them to observe several key aspects of nucleocytoplasmic transport:

- 1. The duration of translocation events through the NPC typically ranges from approximately 10 to 20 milliseconds.*
- 2. Import and export processes occur within overlapping spatial regions, indicating the absence of dedicated channels specific to each process.*
- 3. Translocation predominantly occurs at the periphery of the NPC, with a notable exclusion zone in the center.*
- 4. Both import and export processes are largely confined to a single octant of the NPC.*

5. Transient pauses are observed during the transport process.
6. An apparent diffusion coefficient of approximately $0.06 \mu\text{m}^2/\text{s}$ was recorded during translocation, akin to the diffusion behavior of proteins in a highly viscous medium such as glycerol.

From a technical standpoint, the paper appears to be of high quality, demonstrating the successful deployment of the complex MINFLUX imaging methodology for single particle tracking. In addition, the authors show that MINFLUX can be used to describe nucleocytoplasmic transport at the single molecule level revealing important and novel details of transport through the NPC. Nevertheless, there are several areas where further clarification and enhancement could strengthen the findings.

We thank the Reviewer for specifically outlining the findings of our study.

Major Comments:

- The manuscript highlights that 32 of the 33 tracks remain confined to a single local octant, suggesting highly defined translocation routes. This is probably the most unexpected finding that cannot be explained with current models of NPC function and selectivity. Also, current models of NPC structure do not provide any structural basis for such a spatial restriction of NPC translocation, and it is not clear how the interactions between importins and the nucleoporin FG repeats would be limited to one octant during translocation. This needs to be discussed in more detail. Could the authors propose a speculative mechanism for this restriction? Alternatively, could there be any image processing steps that inadvertently bias the localization of tracks toward one octant? Do the authors potentially observe stable transport receptor-FG repeat interactions and could the movement that they detect reflect FG repeat mobility? It would be important that the authors address this issue and potentially provide additional experimental evidence.

We agree that the most unexpected finding of this study is that the tracks are largely confined to a single octant during the translocation step. While current computational models do not predict such behavior, they have not, until now, ever had to seek conditions that give rise to such constraints. The most likely explanation is that there are defined pathways or conduits through the permeability barrier. These do not need to be rigid channels, but rather high probability regions of transit, which could, for example, be regions of lower FG-polypeptide density. Multiple groups have suggested the presence of such channels, initially in cartoon form based on structural propensities (ref. 37) and later within EM images (ref. 38). Most likely, any such constrained behavior would be a result of both the FG-Nups and bound NTRs, suggesting a complex set of interactions. However, we cannot rule out any direct interactions of transiting complexes with the NPC scaffold. These issues are now discussed on p. 10, lines 2-11.

We have no evidence that the confined tracks are an imaging artifact. In the complete dataset, the entire range of expected angles (-180° to 180°) was observed, and migration between octants was observed outside the pore (e.g., see **ED Figs. 7,8**). Using the criteria that entrance/exit occurs at $z = \pm 25$ nm, there is a clear bias to remain within an octant during transport (**Fig. 3H,L**). While localizations during transit are not strictly confined to an octant, which can be observed in individual tracks (**ED Figs. 7,8**) and the $\Delta\theta$ summaries (**Fig. 3H,L**), the latter suggests that any such deviations from an octant are likely statistical noise.

It is unclear how we could be expected to establish the presence of a stable NTR-FG repeat interaction from the current data. This would require simultaneously but independently monitoring the position of the NTR and the FG-polypeptide – a significant experimental challenge. We therefore cannot establish at this time on whether such interactions could, for example, be maintained for

significant periods or for the duration of transport, e.g., as an FG-polypeptide flips from one side of the pore to another. This is a formally possible model, however. In our revised jump step analysis (**Fig. 3O,P**), the transiting particles appear stuck over half the time, which could indicate a stable NTR-FG interaction, but which could also arise from confinement within the permeability barrier.

- *Additionally, it is important to determine how many of the 33 tracks that translocate through the Nuclear Pore Complex (NPC) interact with two different octants—one in the Cytoplasmic Ring (CR) and another in the Nuclear Ring (NR), as depicted in Figure 3B. Alternatively, it would be useful to know whether these trajectories typically bind to just one octant, either in the CR or the NR, and then translocate without further binding events. Furthermore, it is necessary to explore whether the speed of the MINFLUX scanning could introduce any biases in the observations of these two scenarios. Regarding the localization of tracks within the same octant in both the Cytoplasmic Ring (CR) and Nuclear Ring (NR), how frequently is this observed? Is there sufficient data to support or refute this occurrence, or do most trajectories simply dwell within one octant before crossing the NPC*

The extent of angular confinement within the pore was initially assessed in the previous **ED Fig. 7I,J**. In this analysis, the trajectory point nearest the pore center was used as a reference and the angle from this reference point was plotted for all points on the trajectories between $z = -20$ nm and $z = +20$ nm. This covered most of the distance between the two rings (separated by ~ 50 nm).

We have now adjusted our analysis slightly. For all import and export trajectories that completely crossed the pore (at least one point > 25 nm and one point < -25 nm), the trajectory point nearest the pore center was used as a reference and the $\Delta\theta$ was calculated for all points within the pore during the crossing event. This extends the analysis to cover the entire distance between the two rings. Importantly, this selection rule restricting $\Delta\theta$ values to those during the crossing event eliminates larger angles that had been identified previously (where z value was the only constraint). In some long trajectories, it appears that some complexes can re-enter a pore at a different octant after escaping, or if a transport event is unsuccessful, a complex can enter and cross within a different octant. Such behavior was infrequent, and therefore considered insufficient for reporting a statistical analysis at this time. As the confinement to an octant during transport is recognized as a key point of the paper, this analysis has now been moved into the main text (**Fig. 3H,L**).

The MINFLUX scanning speed does not appear to be biasing localization to the octants since outside the central pore some trajectories moved to different octants (**ED Figs. 7,8**).

As we pointed out in our response to the previous comment, localizations during transit are not strictly confined to an octant, which can be observed in individual tracks (e.g., **ED Figs. 7,8**) and the $\Delta\theta$ summaries (**Fig. 3H,L**), the latter suggests that any such deviations from an octant are likely statistical noise.

- *The paper posits the existence of a central exclusion zone in the NPC where no import or export events are detected. Could this absence be attributed to the rapid movement of proteins through the center, potentially exceeding MINFLUX's temporal resolution? Furthermore, if these trajectories are indeed very rapid, could their photon count fall below the detection threshold, leading to their exclusion from analysis revealing only long-lived interactions at the periphery? Further justification and exploration of this point would be valuable.*

We cannot rule out that the imaging method missed very rapid translocation events. In this and previous work, however, we have repeatedly observed large ‘clouds’ of slow mobility regions extending ~100 nm from both the nucleoplasmic and cytoplasmic exits of the pore (the ‘vestibule’ regions). While it is possible that there are conduits through these regions and the pore that allows for very rapid undetectable transport (i.e., sequential localizations on opposite sides of the pore), we feel that this is unlikely. The fact that molecules were observed in an annular region and did not cross to the other side, e.g., by fast transit through the central region, supports a model in which access to the central channel is indeed blocked. This is explicitly considered on p. 7, line 45 to p. 8, line 22.

- *The authors are encouraged to include an XZ visualization plot of all 33 trajectories that were successfully tracked during import or export in the supplement, similar to what is presented in Figure 3B.*

Individual import and export events are now shown in **ED Figs. 7 & 8**.

- *A significant limitation is the low throughput of the experiment. Although the complexity of the setup is acknowledged, the collection of only 33 trajectories prevents a robust statistical basis to comprehensively characterize nucleocytoplasmic transport dynamics. Increasing the number of tracked trajectories by an order of magnitude would greatly enhance the reliability of the conclusions. The authors should at least discuss why they limit their analyses to such a low number of trajectories.*

At the dilution needed to unambiguously identify single cargo transport events, transport occurs infrequently. Coupled with the facts that transport events are fast (tens of milliseconds), that not all NPCs yield well-resolved scaffold structures, that the realistic observation window before the permeabilized cell structure becomes unstable is ~45 minutes, that some cells could not be used due to cell instability or insufficient usable fiducial markers, that not all tracks enter or completely transit a pore, and that the MINFLUX scan pattern must fall on an NPC at the exact time a molecule is transiting through, it quickly becomes apparent that these experiments are quite challenging, and throughput is currently low. On average, we obtained < 2 import/export tracks per cell and could maximally examine ~3-4 cells/day (imaging statistics in **Table S4**). Nonetheless, with this revision, we have increased the total number of import and export tracks to 55.

As a technical sidenote, to perform these experiments, personnel were sent from the home lab in Texas, USA with all reagents to a MINFLUX instrument in Heidelberg, Germany because that is where the appropriate MINFLUX technical expertise on NPCs was to perform the experiments. Substantially increasing dataset size without a local MINFLUX machine is therefore quite resource intensive.

Minor Comments:

- *The authors state that current models of NPC translocation propose different channels for import and export within the NPC. Is there substantial experimental support for such a model and is this indeed accepted within the field? Additional references or a more detailed discussion of this topic would be beneficial.*

This has been a controversial topic for decades, and strong direct experimental evidence for or against this idea has been difficult to obtain. We have toned down our statement by eliminating the phrase ‘widely held view’ to now indicate that separate import and export paths is an option that

would help alleviate congestion (p. 2, lines 6-8). We have also modified the abstract, removing the indication that separate import and export channels is “a common assumption”.

- *The error estimation procedure assumes a Gaussian distribution of jump sizes based on the diffusion coefficient and time lag between localizations. Considering that importins interact with the NPC scaffold, which leads to anomalous diffusion, does this assumption affect the accuracy of localization precision estimates? Could the authors delve deeper into this aspect?*

Please see our responses to Reviewers 1 and 2 regarding revisions to the error analysis model.

We have now identified models that are sufficient to explain the experimental data. While mean-squared displacement curves indicate anomalous diffusion by their curvature at longer time intervals, jump step histograms are intrinsically less sensitive since they typically compare distances between pairs of successive points rather than points with increased temporal separation. The most anomalous diffusion (a static particle) yields a jump histogram that is very similar to that of a diffusing particle (**ED Fig. 5E**), indicating that the anomalous nature of movement is difficult, if not impossible, to identify through jump histogram analysis. So yes, the accuracy of the localization precision estimate could be influenced by anomalous diffusion behavior, but the characteristics of the diffusional volume are insufficiently known at this time to build a reasonable more-refined model to decipher an anomalous contribution within trajectories with complex movement behaviors.

- *The filtration criteria outlined in Extended Data Figure 6 should be more clearly explained, as understanding these criteria are crucial to evaluate the data.*

We have clarified our discussion of these filtration criteria in the figure caption and in a new ‘Data Filtering Parameters’ section of the Methods (p. 17).

Overall, the manuscript has been improved significantly by directly addressing the questions and comments of the Reviewers. We are grateful for the thoughtfulness and time expended by the Reviewers in critiquing this manuscript. Lastly, we thank you for your efforts in handling this manuscript. We hope that the Reviewers and you find the current version of this manuscript suitable for publication in *Nature*.

December 20, 2024

Senior Editor, *Nature*

Re: Nature Manuscript 2024-03-06173A
"Overlapping nuclear import and export paths unveiled by two-color MINFLUX"

Dear Editor,

Thank you for shepherding the review of the above manuscript and for inviting us to submit a second revision.

We appreciate the comments of the Reviewers. In our point-by-point response, comments from the Reviewers are in *italics*, and our responses follow in blue.

For ease in re-review, we have modified the previously submitted manuscript with track changes. This version includes the figure modifications described as follows in our response to the Reviewers, but these changes are un-marked in the manuscript file. The page and line numbers refer to the text with tracked changes visible. This version of the manuscript is submitted as a pdf as a 'Related Manuscript File'.

We have also included a version of the manuscript that more closely aligns with *Nature* guidelines, as requested in the letter reporting the result of the last review. This manuscript version is submitted as an MS Word file and is available for review as a pdf merged with the main figures by the submission portal. In the Word version of the manuscript, edits can be identified with the track changes feature. Some additional minor text edits have been added to enhance clarity. All main and extended data figures are submitted as individual files in the appropriate format. We have somewhat reduced the length of the main figure captions, but to us the information currently included is essential.

REVIEWER 1:

I have been very much impressed by the resubmission of the authors. They went to great length in the data acquisition and analysis to address all my comments and the other two reviewers comments.

On the technical side, the idea to use the jump histogram turned out to be good. The loc. unc. analysis and the influence on the error propagation has been commented by all reviewers. Now I think it is sound and nice.

I do not have any further comments and recommend publication.

Referee #1 (Remarks on code availability):

The now is fine.

We thank the Reviewer for these very kind comments.

No changes were made in response to these comments.

REVIEWER 2:

The revised manuscript has addressed the points I raised during the previous round of review. I however have some remaining minor comments.

1) The authors measured a further dataset to strengthen the statistical relevance of their claims. While they adjusted some parameters for Imp a-JF549 3D tracking, the parameters for NbGFP-HMSiR 3D localizations remained the same (Table S1). When looking at the time between successive MINFLUX localizations, dataset 2 shows a sharp peak at 3-4 ms whereas dataset 1 peaks between 5-6 ms and has a very strong tail. Could the authors comment on where these differences arise from?

We have not been able to identify the source of these differences with 100% certainty as these two datasets were collected about a year apart and the instrument was realigned several times between data acquisition sessions. The most likely explanation, however, is the larger EFO values (EFO = emission frequency at offset) within the HMSiR Dataset 2. The EFO is a measure of the emission intensity. With a higher average EFO, the localization is less likely to fail since sufficient photons are collected within the dwell time of the localization. Longer localization times occur when the dwell time needs to be extended to collect enough photons. EFO histograms for the two HMSiR Datasets are now included as insets into **ED Fig. 1A,B**. These EFO histograms reveal a significantly higher EFO peak within Dataset 2 for HMSiR. We note that the different distributions of HMSiR localization times for the two datasets should have no material effect on the NPC reconstructions since the total datasets were collected over 15-20 minutes and the NPC scaffolds have been shown to be stable over this time period. We now indicate this potential explanation in the caption for **ED Fig. 1A,B** and indicate the higher EFO range of 50-100 kHz for filtering data for HMSiR Dataset 2 in the caption for **ED Fig. 6B**.

2) The histograms in Fig. 1G, H show no scalebar. When comparing to ED Fig. 4G, it becomes apparent that the scalebar does not start at 0 but only at 30 localizations. Please add scalebars to clarify.

We thank the Reviewer for pointing out that this valuable information was missing. We have now added a scalebar for **Fig. 1G,H**. The scalebar is in units of percent of maximum, not counts. This is now noted in the caption for **Fig. 1G,H**.

3) In the introduction, it is stated that the tracking is limited to diffusional steps of most 100-200 nm, dependent on donut size. Does this not rather depend on the size of the TCP?

Both the donut size and scan pattern (TCP) will affect the ability to measure step sizes over the entire range of the target region with high precision. Only molecules within the TCP that are excited with the donut can be localized; thus, a very large TCP would require a larger donut. However, the Reviewer

is correct that it is the TCP that is the primary determinant of what step sizes are actually measurable. This was a phrasing error. To avoid confusion within this very general discussion of the MINFLUX technique, we have kept it simple and modified the parenthetical comment within this sentence to 'depending on acquisition parameters', and we have further clarified this sentence (p. 2, lines 28-29). To the following sentence, we have added 'in excess' (p. 2, line 31) to clarify that it is the larger steps beyond ~200 nm that impede MINFLUX tracking ability.

4) *With the added tracks, Fig. 3E,F,I,J,M and N are overly crowded, making it difficult for the reader to extract information. Please reduce the number of tracks shown.*

We have removed tracks ($N = 22$ reduced to 13 or 17) from **Fig. 3E,F,I,J** so that individual tracks are more visible. Since we now show individual import and export tracks in **ED Figs. 7 and 8**, these no longer need to be combined into single panels in **Fig. 3** in the main text. The abortive tracks in **Fig. 3M,N**, however, emphasize an important point when combined, namely that none of them penetrate far across the midplane of the pore ($z = 0$). It is for this reason that we can conclude that the decision between transport and no transport is made near the pore midplane. While we have deleted some tracks from **Fig. 3M,N** to clean up the image a bit ($N = 42$ and 45 reduced to 32 and 35 , respectively), we feel that the number of tracks should remain high in these images to drive home the point about the decision point for transport. Unlike the import and export trajectories shown in **ED Figs. 7 and 8**, abortive import and abortive export trajectories are not shown elsewhere in the manuscript.

5) *When analyzing the diffusional behavior of Imp a-JF549, the authors now directly jump from a model with one species to a model with three species. Could a model assuming two species also fit the data? The model with three species additionally allows for a lot of free fit parameters. Especially when looking at the two non-diffusive species, are there any other combinations of parameters which allow explaining the data equally well?*

We have added two examples of two-species models in **ED Fig. 5J,K** that include both stuck (58-60%) and diffusive (40-42%) species. We were not able to obtain a two-species model that yields a good fit to both the jump step and R^2/t histograms simultaneously. In general, parameter sets that increase the peak in the R^2/t histogram near zero also shift the peak in the jump step histogram to the left. The only way that we found to resolve this was to add a third species – notably, only 7% of a third species is required in our best fit model (**Fig. 3O,P**, red curves). Textual changes accompanying the revised **ED Fig. 5J,K** are found in the figure caption (p. 39, lines 26-33) and the main text (p. 7, lines 23-27).

REVIEWER 3:

The authors have done a good job addressing the comments of the reviewers. Importantly, they have increased the number of trajectories, have addressed most technical issues and have clarified the text and the discussion.

I remain excited about the results and recommend publication.

We thank the Reviewer for these kind comments.

No changes were made in response to these comments.

Once again, we express our thanks for the thoughtfulness and time expended by the Reviewers in critiquing this manuscript, which has been significantly improved as a result. Lastly, we thank you for your efforts in handling this manuscript. We hope that the Reviewers and you find the current version of this manuscript suitable for publication in *Nature*.